# Dropout-Based Rashomon Set Exploration for Efficient Predictive Multiplicity Estimation

**Hsiang Hsu[1], Guihong Li[2]\*, Shaohan Hu[1], and Chun-Fu (Richard) Chen[1]**
[1]JPMorgan Chase Bank, N.A., USA
[2]The University of Texas at Austin, TX, USA
{hsiang.hsu, shaohan.hu, richard.cf.chen}@jpmchase.com
lgh@utexas.edu

## Abstract

Predictive multiplicity refers to the phenomenon in which classification tasks may admit multiple competing models that achieve almost-equally-optimal performance, yet generate conflicting outputs for individual samples. This presents significant concerns, as it can potentially result in systemic exclusion, inexplicable discrimination, and unfairness in practical applications. Measuring and mitigating predictive multiplicity, however, is computationally challenging due to the need to explore all such almost-equally-optimal models, known as the Rashomon set, in potentially huge hypothesis spaces. To address this challenge, we propose a novel framework that utilizes dropout techniques for exploring models in the Rashomon set. We provide rigorous theoretical derivations to connect the dropout parameters to properties of the Rashomon set, and empirically evaluate our framework through extensive experimentation. Numerical results show that our technique consistently outperforms baselines in terms of the effectiveness of predictive multiplicity metric estimation, with runtime speedup up to $20\times \sim 5000\times$. With efficient Rashomon set exploration and metric estimation, mitigation of predictive multiplicity is then achieved through dropout ensemble and model selection.

## 1 Introduction

The Rashomon effect, first introduced in Breiman [2001], describes the phenomenon that there is not just one "best" explanation for the data, but many almost-equally-optimal models. The deliberation of Rashomon effect has recently prevailed due to the increasingly complicated learning architectures that exacerbate the under-specification[1] of optimization problems [D'Amour et al., 2022]. The collection of these almost-equally-optimal models is termed the Rashomon set [Semenova et al., 2019; Marx et al., 2020]. The existence of Rashomon sets facilitates finding almost-equally-optimal models that further carry practically-desired properties for practitioners, such as interpretability [Rudin, 2019], epistemic patterns (e.g. causality) [Hancox-Li, 2020], fairness [Coston et al., 2021], or compliance with domain knowledge [Hasan & Talbert, 2022].

Despite the newfound potential, the Rashomon effect introduces a burgeoning challenge in classification problems, known as *predictive multiplicity*—the phenomenon where almost-equally-optimal classification models assign conflicting predictions to individual samples, i.e., the predictions given to an individual could be determined by an arbitrary model in the Rashomon set. When predictive multiplicity is left out of account, an arbitrary choice of a single model in the Rashomon set may lead to systemic exclusion from critical opportunities, unexplainable discrimination, and unfairness, to individuals [Creel & Hellman, 2021]. Moreover, recent studies suggest predictive multiplicity as an additional dimension to evaluate the trade-offs of privacy-assuring mechanisms such as differential privacy [Kulynych et al., 2023] and fairness interventions [Long et al., 2023]. Therefore, measuring predictive multiplicity has been increasingly recognized as a key aspect when reporting model performance aside from its accuracy, for example, in Model Cards [Mitchell et al., 2019].

Several metrics have been proposed to measure predictive multiplicity by quantifying either the spread of output scores from models in the Rashomon set such as viable prediction range [Watson-

---

\*Work done during internship at JPMorgan Chase Bank, N.A.

[1]Under-specification means there is no unique solution to an optimization problem, e.g. the empirical risk minimization that is widely used in modern machine learning [Teney et al., 2022].

Daniels et al., 2023] and Rashomon Capacity [Hsu & Calmon, 2022], or the inconsistency of decision such as ambiguity/discrepancy [Marx et al., 2020] and disagreement [Kulynych et al., 2023], to name a few. The exact computation of these metrics relies on full access to models in the Rashomon set, which is computationally infeasible and memory inefficient when the hypothesis space of candidate models is large and complex. Therefore, current strategies to compute predictive multiplicity metrics either focus on special hypothesis spaces (e.g., linear classifiers [Marx et al., 2020], sparse decision trees [Xin et al., 2022], and generalized additive models [Chen et al., 2023]), or by repeatedly re-training models with different random seeds to explore the Rashomon set when there is no special structures to relieve the computational overheads. However, re-training is still time-consuming when learning with large datasets. Thus, how to efficiently estimate predictive multiplicity metrics for highly complicated models, esp. neural networks, still remains an open problem.

In this paper, we propose an efficient dropout-based Rashomon set exploration framework that is easily applicable to neural networks without strenuously-repeated model re-training. The dropout technique was first found useful to reduce over-fitting [Srivastava et al., 2014], and is later connected with Bayesian approximation of neural networks to measure prediction uncertainty [Gal & Ghahramani, 2016]. However, Gal & Ghahramani [2016] do not consider prediction uncertainty with the notion of the Rashomon effect—they consider prediction uncertainty among all possible models in the hypothesis space instead of models in the Rashomon set (see Appendix C for further comparisons). To the best of our knowledge, this work is the first in using dropout techniques for model exploration in Rashomon sets and efficient estimation of predictive multiplicity metrics. We provide a rigorous theory on how hyper-parameters of dropout and network architectures control the loss deviations of Rashomon sets for feed-forward neural networks (FFNNs). Through meticulous parameter selection, a dropout model is *highly likely* to belong to a Rashomon set as the number of model parameters approaches infinity. Besides theoretical discussions, we conduct empirical studies of measuring predictive multiplicity metrics using dropout with multiple datasets from diverse domains, including financial analytics, medical prediction, large-scale images classification, and human detection. Exploring models in Rashomon sets with our framework attains $20\times \sim 5000\times$ speedup compared with the baseline methods such as re-training. Finally, we demonstrate the mitigation of predictive multiplicity using the proposed dropout-based Rashomon set exploration with (i) ensemble methods and (ii) model selection with smaller predictive multiplicity.

Omitted proofs, additional explanations and discussions, details on experiment setups and training, and additional experiments are included in the Appendix. Code to reproduce our experiments can be accessed at https://github.com/jpmorganchase/dropout-rashomon-set-exploration.

## 2 BACKGROUND AND RELATED WORK

We consider a dataset $\mathcal{D} = \{(\mathbf{x}_i, \mathbf{y}_i)\}_{i=1}^n$, where each pair $(\mathbf{x}_i, \mathbf{y}_i)$ consists of a feature vector $\mathbf{x}_i = [\mathbf{x}_{i1}, \cdots, \mathbf{x}_{id}]^\top \in \mathcal{X} \subseteq \mathbb{R}^d$ and a target $\mathbf{y}_i \in \mathcal{Y}$. We denote by $\mathcal{H}$ a hypothesis space, a set of candidate models parameterized by $\mathbf{w} \in \mathcal{W}$, i.e., $\mathcal{H} \triangleq \{h_{\mathbf{w}} : \mathcal{X} \to \hat{\mathcal{Y}} : \mathbf{w} \in \mathcal{W}\}$. The loss function used to evaluate model performance is denoted by $\ell : \hat{\mathcal{Y}} \times \mathcal{Y} \to \mathbb{R}^+$ and $L(\mathbf{w}, \mathcal{D}) \triangleq \frac{1}{|\mathcal{D}|} \sum_{(\mathbf{x}_i, \mathbf{y}_i) \in \mathcal{D}} \ell(h_{\mathbf{w}}(\mathbf{x}_i), \mathbf{y}_i)$ denotes the empirical risk evaluated with dataset $\mathcal{D}$. When the context is clear, we simply write $L(\mathbf{w})$ instead of $L(\mathbf{w}, \mathcal{D})$ for the risk, and use $\mathbf{w}$ to denote a model. For $c$-class classification problems, i.e., $\mathcal{Y} = [c]$ and $\hat{\mathcal{Y}}$ is the $c$-dimensional probability simplex $\Delta_c \triangleq \{(r_1, \cdots, r_c) \in [0, 1]^c; \sum_{i=1}^c r_i = 1\}$, the loss function could be the cross-entropy loss or the Brier score loss [Brier, 1950] $L_{\mathsf{BS}}(\mathbf{w}) \triangleq \frac{1}{n} \sum_{i=1}^n \|h_{\mathbf{w}}(\mathbf{x}_i) - \mathbf{y}_i\|_2$. For regression problems, $\mathcal{Y} = \hat{\mathcal{Y}} = \mathbb{R}$, and the loss function is mean square error (MSE) loss $L_{\mathsf{MSE}}(\mathbf{w}) \triangleq \frac{1}{n} \sum_{i=1}^n (h_{\mathbf{w}}(\mathbf{x}_i) - \mathbf{y}_i)^2$. In this paper, we mainly focus on predictive multiplicity for classification problems. Finally, for a vector $\mathbf{v}$, $\mathrm{diag}(\mathbf{v})$ is a matrix with $\mathbf{v}$ on diagonals and 0 on off-diagonals, and for a matrix $\mathbf{V}$, $\mathrm{diag}(\mathbf{V})$ is a matrix that only keeps the diagonal entries.

**The Rashomon set.** We define the Rashomon set as the set of all models in the hypothesis space $\mathcal{H}$ whose empirical risks are *similar* to that of a given empirical risk minimizer $\mathbf{w}^* \in \mathrm{argmin}_{\mathbf{w} \in \mathcal{W}} L(\mathbf{w})$. Formally, given a Rashomon parameter $\epsilon \geq 0$, the *Rashomon set*[2] is defined as [Semenova et al., 2019]:

$$\mathcal{R}(\mathcal{H}, \mathcal{D}, \mathbf{w}^*, \epsilon) \triangleq \{h_{\mathbf{w}} \in \mathcal{H}; L(h_{\mathbf{w}}, \mathcal{D}) \leq L(h_{\mathbf{w}^*}, \mathcal{D}) + \epsilon\}. \tag{1}$$

---

[2]The Rashomon set is defined regarding any given dataset $\mathcal{D}$, even for out-of-distribution data.

Here, $\epsilon$ determines the size of the set. We omit the arguments of $\mathcal{R}(\mathcal{H}, \mathcal{D}, \mathbf{w}^*, \epsilon)$ later in this paper when they are clearly implied from context. The Rashomon set is at the core of measuring predictive multiplicity, as introduced next.

**Measuring predictive multiplicity.** There are various metrics to quantify predictive multiplicity across models in $\mathcal{R}$ by either considering their decisions (thresholded predictions/scores after $\mathrm{argmax}$) or output scores. Due to space limit, we summarize the mathematical formulations of existing predictive multiplicity metrics in Appendix B.

Decision-based metrics include *ambiguity* and *discrepancy*, which measure the proportion of samples in a dataset that can cause models in the Rashomon set to make conflicting decisions [Marx et al., 2020], and *disagreement*, which is the probability that any two models in the Rashomon set output different decisions for a given sample [Kulynych et al., 2023]. On the other hand, a finer-grained characterization of predictive multiplicity examines the output scores (i.e., vectors in $\Delta_c$). For example, Long et al. [2023], Cooper et al. [2023] and Watson-Daniels et al. [2023] quantify predictive multiplicity by the standard deviation, variance and the largest possible difference of the scores (termed *viable prediction range (VPR)* therein) respectively. The *Rashomon Capacity (RC)* measures score variations in the probability simplex with information-theoretic quantities that can be generalized beyond binary classification. Despite score conflicts can not be directly translated to decision conflicts, score-based metrics avoid potential over-estimation of predictive multiplicity compared to decision-based metrics (see [Hsu & Calmon, 2022, Fig. 2] for a detailed discussion).

**Exploring models in the Rashomon set.** The computation of predictive multiplicity metrics (cf. Appendix B) requires an exact characterization[3] of a Rashomon set, i.e., having full access to all of its models. When $\mathcal{H}$ is large (e.g., a neural network architecture), exhaustively finding all models in the Rashomon set is computationally infeasible. Therefore, to approximate a full Rashomon set $\mathcal{R}(\mathcal{H}, \mathcal{D}, \mathbf{w}^*, \epsilon)$, we define an *empirical* Rashomon set with $m$ models, as follows,

$$\mathcal{R}^m(\mathcal{H}, \mathcal{D}, \mathbf{w}^*, \epsilon) \triangleq \{h_{\mathbf{w}_1}, \cdots, h_{\mathbf{w}_m} \in \mathcal{H}; L(h_{\mathbf{w}_i}, \mathcal{D}) \leq L(h_{\mathbf{w}^*}, \mathcal{D}) + \epsilon, \; \forall i \in [m]\}. \quad (2)$$

The empirical Rashomon set $\mathcal{R}^m(\mathcal{H}, \mathcal{D}, \mathbf{w}^*, \epsilon)$ is a subset of the Rashomon set defined in (1). Predictive multiplicity metrics can then be evaluated with the empirical Rashomon set instead of the Rashomon set. Note that any estimation based on the empirical Rashomon set is an *under-estimate* of the true predictive multiplicity. In consequence, an empirical Rashomon set is a "better" approximate of the true Rashomon set than another empirical Rashomon set if it leads to higher estimates of predictive multiplicity metrics. As $m$ increases, the empirical Rashomon set better recovers the Rashomon set, leading to a more precise estimation of predictive multiplicity metrics. Therefore, how to *efficiently* acquire a vast amount of models in a Rashomon set becomes the core problem of measuring predictive multiplicity.

In practice, the $m$ models in the empirical Rashomon set can be obtained either by *re-training* with different initializations (e.g., different random seeds to initialize the model weights, different data shuffling for stochastic gradient descent, etc.), or by *adversarial weight perturbation (AWP)* [Hsu & Calmon, 2022, Section 4]. The re-training strategy views a training procedure $\mathcal{T}$ to be randomized. Accordingly, we can denote a random variable $\mathcal{T}(\mathcal{D})$ that outputs all possible models trained with procedure $\mathcal{T}$ on $\mathcal{D}$. Different models in $\mathcal{T}(\mathcal{D})$ can be induced by using different random initialization[4]. By re-training models and rejecting those that disobey the loss deviation constraint in (2), we are able to collect $m$ models for the empirical Rashomon set. The AWP strategy, on the other hand, aims to perturb the weights of a pre-trained model such that the output scores of a sample are thrust toward all possible classes, again under the loss deviation constraint (cf. Appendix B for more details). Both re-training and AWP require repeatedly training models, which is time-consuming and hinders practitioners from efficiently estimating predictive multiplicity metrics.

Another strategy besides re-training and AWP is to explore *sparse* models in the Rashomon set. For example, Xin et al. [2022] prove that large portions of the decision tree in the hypothesis space do not contain any members of the Rashomon set and can safely be excluded, and provide an algorithm and data structure to completely enumerate and store all models in the Rashomon set for sparse decision trees. More recently, Chen et al. [2023] study an interpretable, predictive model called generalized

---

[3]It is possible for simple models such as an exact characterization of the Rashomon set for ridge regression [Semenova et al., 2019, Section 5.1] or an exact computation of ambiguity/discrepancy for linear classifiers by mixed integer programming [Marx et al., 2020, Section 3].

[4]See Kulynych et al. [2023] and Semenova et al. [2019] for more details about the re-training strategy.

additive models (GAMs), and use high-dimensional ellipsoids to approximate the Rashomon sets of sparse GAMs. Despite these progress, how to explore Rashomon sets and measure predictive multiplicity for more complicated models, especially neural networks, still remain unclear. This paper intends to bridge this gap using dropout techniques, as discussed next.

## 3 EXPLORING THE RASHOMON SET WITH DROPOUT

The dropout technique, originated from the concept of dilution[5] [Hertz et al., 1991], is a family of stochastic techniques for regularization to prevent over-fitting [Hinton et al., 2012]. As implied by its name, the dropout technique randomly removes neurons in a neural network at each training time, and can be interpreted as implicitly averaging over an ensemble of *sparse* neural networks with different configurations during training [Srivastava et al., 2014].

In addition to applying dropout in training, Gal & Ghahramani [2016] apply dropout at inference time to estimate output uncertainty of a model. They interpret dropout as a variational approximation of a deep Gaussian process, which is a Bayesian framework that produces a probability distribution of model outputs. The underlying distribution can then be used to estimate the variance for a certain input, indicating the uncertainty of the model. However, the dropout inference studied in Gal & Ghahramani [2016] has not taken the Rashomon effect into consideration, i.e., they consider the output uncertainty of all possible models in the hypothesis space instead of all almost-equally-optimal models in the Rashomon set; see Appendix C for a more thorough discussion. In this case, predictive multiplicity and output uncertainty are over-estimated, since models without comparable accuracy are also counted in, and therefore dropout inference is not directly applicable to measuring predictive multiplicity. To address this issue, here, we illustrate how dropout parameters control the loss deviations of Rashomon sets for linear models, and eventually for FFNNs.

### 3.1 FORMULATION OF DROPOUT TECHNIQUES

We start with the mathematical formulation. Consider model weights $\mathcal{W} \in \mathbb{R}^d$, and define dropout random variables as $\mathbf{z} = [Z_1, Z_2, \cdots, Z_d]$. Let $\mathbf{D_z} = \text{diag}(\mathbf{z}) \in \mathbb{R}^{d \times d}$, the weights after dropout can be denoted as $\mathbf{w}_D = \mathbf{D_z}\mathbf{w}$. There are two common choices of the random variables $Z_i$. Bernoulli dropout (also called standard dropout), adopts $Z_i$ to be i.i.d. Bernoulli$(1 - p)$ random variables, where $p \in [0, 1]$ is the probability of dropping out a weight (i.e., the dropout rate). On the other hand, Gaussian dropout applies i.i.d. Gaussian multiplicative noise, Gaussian$(1, \alpha)$, on the weights [Wang & Manning, 2013; Kingma et al., 2015]. Gaussian dropout can also be interpreted as a variational information bottleneck layer [Rey & Mnih, 2021].

Suppose we have $m$ realizations of the dropout matrix, $\mathbf{D_{z_1}}, \cdots, \mathbf{D_{z_m}}$. At the inference time, we may apply each dropout realization $\mathbf{D_{z_i}}$ to a fixed model weight $\mathbf{w}^*$, where each realization is a new, sparse model $h_{\mathbf{D_{z_i}}\mathbf{w}^*}$. We would like to note that even though applying dropout is much more computationally efficient than re-training and AWP to obtain a vast amount of models, if, however, the dropout method leads to a huge loss deviation $L(h_{\mathbf{D_{z_i}}\mathbf{w}^*}) - L(h_{\mathbf{w}^*})$, dropout models are not likely to be in the Rashomon set unless the Rashomon parameter $\epsilon$ is set to be large. Therefore, in the next section, we investigate the connection among dropout parameters (i.e., rate $p$ for Bernoulli dropout and variance $\alpha$ for Gaussian dropout), loss deviations, and the Rashomon set.

### 3.2 THE RASHOMON SET ON RIDGE REGRESSION WITH DROPOUT

We start the analysis with a special case of Bernoulli dropout on ridge regression and its connection to the notion of Rashomon sets. Consider a parametric hypothesis space of linear models $\mathcal{H} = \{h_\mathbf{w}(\mathbf{x}) = \mathbf{w}^\top\mathbf{x}; \mathbf{w} \in \mathbb{R}^d\}$, ridge regression aims to minimizes the penalized sum of squared error (SSE) loss, $L_{\mathsf{SSE}}(\mathbf{w}) \triangleq \|\mathbf{X}\mathbf{w} - \mathbf{y}\|_2^2$ for a dataset $(\mathbf{X}, \mathbf{y}) \in \mathbb{R}^{n \times d} \times \mathbb{R}^n$, i.e., $\min_{\mathbf{w} \in \mathcal{W}} L_{\mathsf{SSE}}(\mathbf{w}) + \lambda\|\mathbf{w}\|_2^2$, where $\lambda$ is the regularization strength. The solution of ridge regression is given by $\mathbf{w}^* = (\mathbf{X}^\top\mathbf{X} + \lambda\mathbf{I}_d)^{-1}\mathbf{X}^\top\mathbf{y}$ [Hastie et al., 2009]. Denoting the weight solution after dropout to be $\mathbf{w}_D^* = \mathbf{D_z}\mathbf{w}^*$, the loss $L_{\mathsf{SSE}}(\mathbf{w}_D^*)$ becomes a random variable. Choosing $Z_i \sim \text{bernoulli}(1 - p)$, the goal is to characterize the deviation between $L_{\mathsf{SSE}}(\mathbf{w}_D^*)$ and $L_{\mathsf{SSE}}(\mathbf{w}^*)$. In fact, if we define the loss deviation as $\epsilon = L_{\mathsf{SSE}}(\mathbf{w}_D^*) - L_{\mathsf{SSE}}(\mathbf{w}'^*)$, where $\mathbf{w}'^* \triangleq (1 - p)\mathbf{w}^*$, then $\epsilon$ is a random variable as well, and its expectation can be computed, as shown in the following proposition.

---

[5]The difference is that dilution randomly removes a connect of a neuron while dropout randomly removes the entire neuron (and all its connections to other neurons). For variants of dilution and dropout, see Hertz [2018, Chapter 3] and Labach et al. [2019].

**Proposition 1.** *Consider Bernoulli dropout with rate $p$, and denote the dropout weights as $\mathbf{w}_D^* = \mathbf{D_z}\mathbf{w}^*$. The loss deviation $\epsilon = L_{SSE}(\mathbf{w}_D^*) - L_{SSE}(\mathbf{w}'^*)$ satisfies*

$$\mathbb{E}_{\mathbf{z}\sim\text{bernoulli}(1-p)}[\epsilon] = \mathbb{E}_{\mathbf{z}\sim\text{bernoulli}(1-p)}\left[L_{SSE}(\mathbf{w}_D^*) - L_{SSE}(\mathbf{w}'^*)\right] = p(1-p)\mathbf{w}^{*\top}\text{diag}(\mathbf{X}^\top\mathbf{X})\mathbf{w}^*. \tag{3}$$

*Moreover, if the features of data matrix $\mathbf{X}$ are linearly independent and normalized, i.e., $\mathbf{X}^\top\mathbf{X} = \mathbf{I}_d$, we have $\mathbb{E}_{\mathbf{z}\sim\text{bernoulli}(1-p)}[\epsilon] = \frac{p(1-p)}{(1+\lambda)^2}\|\mathbf{y}\|_2^2$.*

Proposition 1 illustrates that a dropout model $h_{\mathbf{w}_D^*}$, in expectation, belongs to the Rashomon set $\mathcal{R}(\mathbf{w}'^*, p(1-p)\mathbf{w}^{*\top}\text{diag}(\mathbf{X}^\top\mathbf{X})\mathbf{w}^*)$. We provide Monte Carlo simulation with a Gaussian synthetic dataset with different dimension $d$ in Figure 1. It is clear that the expected simulated losses follow the theory, and sheds light on the possibility to control loss deviations of Rashomon sets using dropout. Proposition 1 can be further improved. First, (3) could be defined regarding the original weights $\mathbf{w}^*$ instead of $\mathbf{w}'^*$. Second, (3) only specifies that dropout models belongs to a Rashomon set in expectation. In the rest of this section, we aim to improve this two limitations, and show that the probability is controlled by the feature dimension $d$.

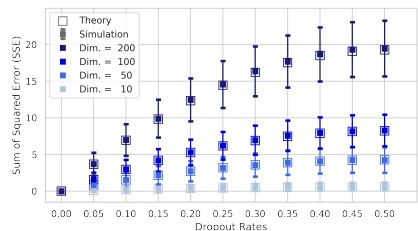

Figure 1: Proposition 1 with 20k models and dimension $d = \{10, 50, 100, 200\}$. As $d$ increases, the variance of loss enlarges, while the mean still matches the theory in (3).

### 3.3  THE RASHOMON SET OF LINEAR MODELS WITH DROPOUT

We start with ridge regression, which Rashomon set $\mathcal{R}_{\text{ridge}}(\mathbf{w}^*, \epsilon)$ has an analytical form; in fact, it is an $d$-dimensional ellipsoid centered at $\mathbf{w}^*$ [Semenova et al., 2019, Theorem 16], i.e.,

$$\mathcal{R}_{\text{ridge}}(\mathbf{w}^*, \epsilon) = \{\mathbf{w} \in \mathcal{W}; (\mathbf{w} - \mathbf{w}^*)^\top(\mathbf{X}^\top\mathbf{X} + \lambda\mathbf{I}_d)(\mathbf{w} - \mathbf{w}^*) \le \epsilon\}. \tag{4}$$

With the exact form of the Rashomon set, we can further compute the probability that a dropout model $\mathbf{D_z}\mathbf{w}^*$ is in the Rashomon set, i.e., $\Pr\{\mathbf{D_z}\mathbf{w}^* \in \mathcal{R}_{\text{ridge}}(\mathbf{w}^*, \epsilon)\}$.

**Proposition 2.** *Without loss of generality, assume $\mathbf{X}^\top\mathbf{X} = \mathbf{I}_d$ (otherwise we can always "whiten" the data matrix $\mathbf{X}$) and the weights are bounded, i.e., $\|\mathbf{w}\|_2^2 \le M$, then the probability that the model $\mathbf{w}^*$ after dropout is in the Rashomon set is lower bounded by*

$$\Pr\{\mathbf{D_z}\mathbf{w}^* \in \mathcal{R}_{ridge}(\mathbf{w}^*, \epsilon)\} \ge \begin{cases} 1 - (1+\lambda)\frac{pM}{\epsilon}, & \text{if } Z_i \overset{i.i.d.}{\sim} \text{bernoulli}(1-p); \\ 1 - (1+\lambda)\frac{\alpha M}{\epsilon}, & \text{if } Z_i \overset{i.i.d.}{\sim} \text{Gaussian}(1, \alpha). \end{cases} \tag{5}$$

Proposition 2 suggests that the dropout parameters, rate $p$ and variance $\alpha$, are important for controlling the probability. Indeed, a trivial case is when $p = \alpha = 0$ (i.e., no dropout), the probability $\Pr\{\mathbf{D_z}\mathbf{w}^* \in \mathcal{R}_{\text{ridge}}(\mathbf{w}^*, \epsilon)\}$ is always 1. Moreover, let $\delta > 0$, if $p = O(d^{-\delta})$ for Bernoulli dropout $Z_i \sim \text{Bernoulli}(1-p)$, and $\alpha = O(d^{-\delta})$ for Gaussian dropout $Z_i \sim \text{Gaussian}(1, \alpha)$, both dropout mechanisms leads to $\lim_{d\to\infty}\Pr\{\mathbf{D_z}\mathbf{w}^* \in \mathcal{R}_{\text{ridge}}(\mathbf{w}^*, \epsilon)\} = 1$ (cf. Appendix A.6 for details).

Similar results also hold if we pass the outputs $h_{\mathbf{w}}(\mathbf{x}_i) = \mathbf{w}^\top\mathbf{x}_i$ through a softmax function $\text{softmax}(t) = 1/(1 + \exp(-t))$ to get output scores, and use the Brier score loss $L_{\text{BS}}(\mathbf{w})$ to minimize the average difference between labels and scores for a classification problem. Using the 1-Lipschitzness of the softmax function, the next proposition shows the probability that dropout models belong to a Rashomon set $\mathcal{R}_{\text{Brier}}(\mathbf{w}^*, \epsilon) = \{\mathbf{w} \in \mathcal{W}; L_{\text{BS}}(\mathbf{w}) - L_{\text{BS}}(\mathbf{w}^*) \le \epsilon\}$.

**Proposition 3.** *Let $\overline{\|\mathbf{x}\|_2^2} = \frac{1}{n}\|\mathbf{x}_i\|_2^2$ and suppose the weights are bounded, i.e., $\|\mathbf{w}\|_2^2 \le M$, for both Bernoulli dropout $Z_i \sim \text{Bernoulli}(1-p)$, and Gaussian dropout $Z_i \sim \text{Gaussian}(1, \alpha)$, we have*

$$\Pr\{\mathbf{D_z}\mathbf{w}^* \in \mathcal{R}_{Brier}(\mathbf{w}^*, \epsilon)\} \ge \begin{cases} 1 - (1+\lambda)\frac{pMd\overline{\|\mathbf{x}\|_2^2}}{\epsilon}, & \text{if } Z_i \overset{i.i.d.}{\sim} \text{bernoulli}(1-p); \\ 1 - (1+\lambda)\frac{\alpha Md\overline{\|\mathbf{x}\|_2^2}}{\epsilon}, & \text{if } Z_i \overset{i.i.d.}{\sim} \text{Gaussian}(1, \alpha). \end{cases} \tag{6}$$

The bound in Proposition 3 is different from that in Proposition 2 with merely a factor of $d$. Therefore, the asymptotic behaviors of the probability hold again as $d$ approaches infinity. Precisely,

as long as $p$ and $\alpha$ are of order $O(d^{-(1+\delta)})$ and the norm of inputs $\mathbf{x}_i$ is finite, $\delta > 0$, we have $\lim_{d \to \infty} \Pr\{\mathbf{D_z}\mathbf{w}^* \in \mathcal{R}_{\mathsf{Brier}}(\mathbf{w}^*, \epsilon)\} = 1$ for both Bernoulli and Gaussian dropout mechanisms.

Proposition 2 and 3 together show that for regression and classification problems, as long as the dropout parameters are carefully controlled with respect to the feature dimension $d$, dropout leads to models in the Rashomon set *with high probability*.

### 3.4 EXTENSION TO FEED-FORWARD NEURAL NETWORKS

In the previous section, we focus on simple linear models. Here, we include the discussion between dropout and the Rashomon set for FFNNs with hidden layers[6]. We use the fact that activation functions $\sigma(\cdot)$ commonly used in neural networks such as ReLU, softmax, and tanh, and non-linear functions such as max-pooling, are Lipschitz with constant 1 (cf. Virmaux & Scaman [2018]).

Consider a $K$-hidden-layer neural network as follows:

$$h_{\mathbf{W}}^K(\mathbf{x}_i) = \mathbf{W}_K^\top \sigma\left(\mathbf{W}_{K-1}^\top \cdots \sigma\left(\mathbf{W}_1^\top \mathbf{x}_i\right)\cdots\right), \tag{7}$$

where $\mathbf{W}_k \in \mathbb{R}^{m_{k-1} \times m_k}$, $k \in [K]$ are the weight matrices, where $m_0 = d$ and $m_K = c$. Let $\mathbf{W} = \{\mathbf{W}_1, \cdots, \mathbf{W}_K\}$ be the collection of all the weights and $\mathbf{W}^* = \{\mathbf{W}_k^*\}_{k=1}^K$ is an empirical minimizer. Now, consider i.i.d. dropout matrices $\mathbf{D}_k$ for each weight matrix $\mathbf{W}_k$ respectively, the output after dropout is then

$$h_{\mathbf{W}_D^*}^K(\mathbf{x}_i) = (\mathbf{D}_K\mathbf{W}_K^*)^\top \sigma\left((\mathbf{D}_{K-1}\mathbf{W}_{K-1}^*)^\top \cdots \sigma\left((\mathbf{D}_1\mathbf{W}_1^*)^\top \mathbf{x}_i\right)\cdots\right), \tag{8}$$

where $\mathbf{W}_D^* = \{\mathbf{D}_k\mathbf{W}_k^*\}_{k=1}^K$ is the collection of all dropout weights. Next, we prove that by carefully controlling variance of Gaussian dropout, the deviation of losses before and after dropout can also be controlled.

**Proposition 4.** *Consider a $K$-hidden-layer neural network $h_{\mathbf{W}^*}^K(\mathbf{x}_i)$ defined in (7) with $\|\mathbf{W}_k\|_F^2 \leq M$ for all $k \in [K]$, and Gaussian dropout, i.e., all diagonal entries of $\mathbf{D}_k \overset{i.i.d.}{\sim} \mathsf{Gaussian}(1, \alpha)$. For both MSE loss $L(\cdot) = L_{\mathsf{MSE}}(\cdot)$ and Brier score loss $L(\cdot) = L_{\mathsf{BS}}(\cdot)$, with probability at least $1 - \rho$,*

$$L(\mathbf{W}_D^*) - L(\mathbf{W}^*) \leq \rho^{-1}M^K\overline{\|\mathbf{X}\|_F^2}\left((m\alpha+1)^K - 1\right), \tag{9}$$

*where $m = \max_{k \in [K]} m_k$ and $\overline{\|\mathbf{X}\|_F^2} = \frac{1}{n}\sum_{i=1}^n \|\mathbf{x}_i\|_2^2$.*

Proposition 4 indicates that both Bernoulli and Gaussian dropouts on neural networks lead to models in the Rashomon set $\mathcal{R}\left(\mathbf{W}^*, \rho^{-1}M^K\overline{\|\mathbf{X}\|_F^2}\left((m\alpha+1)^K - 1\right)\right)$ with probability $1 - \rho$.

## 4 MEASURING PREDICTIVE MULTIPLICITY WITH DROPOUT MODELS

In previous sections, we have shown that the probability of dropout models in a Rashomon set could be controlled by the hyper-parameters of neural network architectures and dropout. These dropout models can then be used to construct the empirical Rashomon set defined in (2), and to estimate predictive multiplicity metrics. Note that not all estimators of predictive multiplicity metrics carry a theoretical analysis of its statistical properties such as consistency and sample complexity (cf. Appendix B for details). In Appendix A.7, we provide additional theoretical analysis regarding a sample complexity bound of estimating (a surrogate metric of) score variance for a single and multiple samples, using Hoeffding's inequality [Hoeffding, 1994] and Gaussian annulus theorem [Blum et al., 2020, Theorem 2.9].

Next, we show empirical results of estimating predictive multiplicity metrics with the empirical Rashomon set obtained from the dropout techniques. We estimate the predictive multiplicity metrics with 6 UCI datasets [Asuncion & Newman, 2007], including three from the financial domain (Adult Income, Bank Marketing, and Credit Approval), and the rest from the medical domain (Dermatology, Mammography, and Contraception). These domains are selected since predictive multiplicity therein could cause critical consequences of injustice. See Appendix D.1 for detailed descriptions and pre-processing of the datasets, and Appendix D.2 for detailed training setups. The baseline methods we adopt to compare with dropout are re-training with different seeds [Semenova et al.,

---

[6]The analysis can also be applied to convolutional neural networks as they are FFNNs that use filters and pooling layers [Goodfellow et al., 2016].

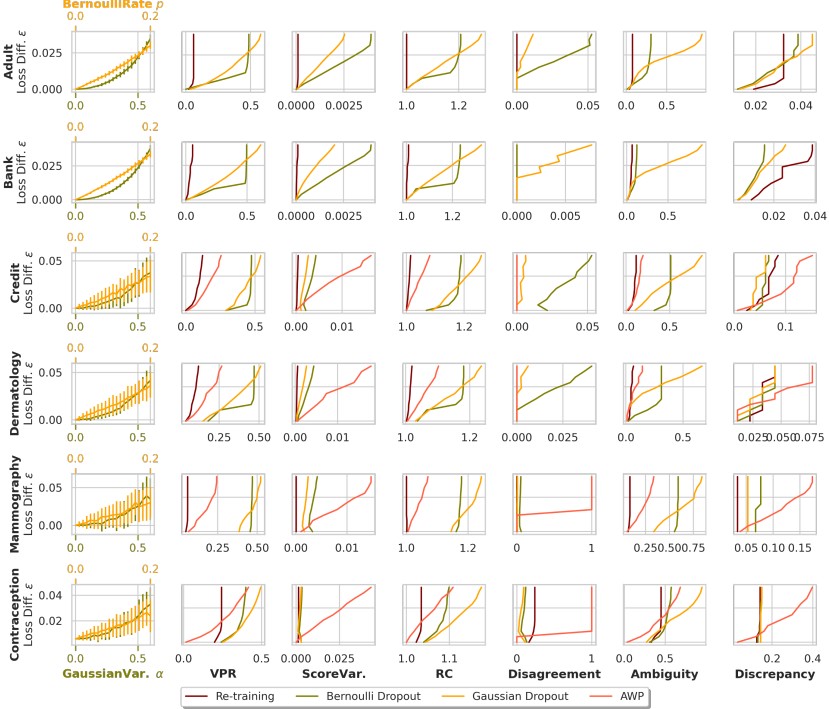

Figure 2: Loss vs. dropout parameters and the corresponding predictive multiplicity metrics of the baselines with UCI datasets. The figures in a row share the same y-axis for the loss difference $\epsilon$, i.e., the Rashomon parameter in (2). Both Bernoulli and Gaussian dropouts give higher multiplicity estimates than re-training under the same loss deviation constraints. In other words, dropout is much more effective than re-training. Despite AWP outperforming all the other methods, it is the most computationally expensive.

2019], and the adversarial weight perturbation algorithm [Hsu & Calmon, 2022]. All the models here have the same architecture—a neural network with a single hidden layer of 1k neurons.

Figure 2 summarizes the estimation of the 6 predictive multiplicity metrics, introduced in Section 2, using the empirical Rashomon set obtained with dropout models. For Bernoulli dropout, we pick the dropout rates $p \in [0.0, 0.2]$, and for Gaussian dropout, the variance $\alpha$ is set to be in $[0.0, 0.6]$; we obtain 100 models per dropout parameters. The leftmost column shows how the loss deviations (Loss diff. $\epsilon$ in the figure) are controlled by varying the dropout parameters, as suggested by the analysis in Section 3. Note that since the Adult Income and Bank Marketing datasets have much more samples than the rest of the datasets, their variances of loss difference are much smaller compared to the other datasets. Moreover, for all datasets, the accuracy deviations caused by the loss differences are all within 1% (see Appendix E.1).

For predictive multiplicity metrics that are defined per sample (e.g., VPR and RC), for the sake of demonstration, we plot the values of 50% or 90% quantile, depending on which quantile value best shows the difference between dropout and the baseline methods. We report the values of those metrics for all samples in Appendix E.1. As observed in Figure 2, both Bernoulli and Gaussian dropouts mostly outperform the re-training strategy in terms of exploring models in the Rashomon set for estimating predictive multiplicity. For example, in both Bank Marketing and Mammography datasets, the VPR and Ambiguity given by retraining are close to zero, whereas dropouts provide diverse models with high multiplicity that are within the same loss deviation regime (i.e.,

Table 1: Average runtime speedup per model on UCI datasets, evaluated with the same computational platform. For the raw values of the runtime, see Table E.6.

| Dataset | Gaussian Dropout Speedup over | |
|---|---|---|
| | Re-training | AWP |
| Adult Income | 305.53× | — |
| Bank Deposit | 35.36× | — |
| Credit Approval | 267.75× | 5345.45× |
| Dermatology | 28.88× | 1121.59× |
| Mammography | 130.20× | 4238.18× |
| Contraception | 78.40× | 2952.73× |

in the same Rashomon set). On the other hand, AWP outperforms both dropouts and re-training, since it *adversarially* searches the models that mostly flip the decisions toward all possible classes for each sample. Despite that AWP best explores the Rashomon set, it comes at the cost of high time complexity, as shown in Table 1, where we record the average runtime per model for re-training, Bernoulli/Gaussian dropouts, and AWP. Note that for all methods, a pre-trained an empirical minimizer $\mathbf{w}^*$ is given and its training time is not included in Table 1. The good performance of AWP comes at the cost of it being more than 1000 times slower than dropout methods. Also, the dropout methods are tens to hundreds times faster than re-training. We refer the reader to Table E.7 for an additional comparison of the efficiency to obtain models from the Rashomon set between the re-training and dropout strategies.

We study a more complicated and practical use case by using the Microsoft COCO human detection dataset [Lin et al., 2014]. The goal of human detection is to determine, in image or video sequence that contain multiple objects, the smallest rectangular *bounding boxes* that enclose humans. Predictive multiplicity therein could lower the trustworthiness of applications involving human detection such as autonomous driving or AR/VR systems [Nguyen et al., 2016]. Figure 3 shows the predictive multiplicity of determining the human bounding boxes with the YoloV3 detector [Redmon & Farhadi, 2018]. We use the Gaussian dropout to explore models in the Rashomon set, whose average precisions (APs) are within 0.5% to the pre-trained model. Despite that the APs of the models are similar, the bounding boxes exhibit high predictive multiplicity in terms of the coverage and confidence of the regions. For the corresponding predictive multiplicity metrics with the YoloV3 and Mask R-CNN [He et al., 2017] detectors, see Appendix E.2.

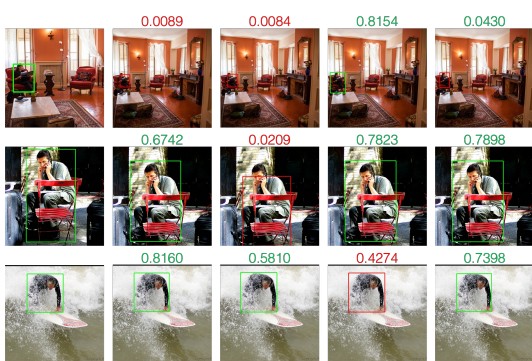

Figure 3: Human detection on MS COCO dataset. The leftest column shows the ground truth of the bounding boxes, and the rest of columns are the bounding boxes found by 4 models in the dropout-based Rashomon set. The green values denote the confidence of the bounding boxes larger than 0.5, and red otherwise. The detectors of the bounding boxes suffer from predictive multiplicity in terms of the coverage and confidence.

For more extensive experiments with CIFAR-10/-100 datasets [Krizhevsky & Hinton, 2009] using VGG16 [Simonyan & Zisserman, 2014] and ResNet50 [He et al., 2016], along with their runtime comparisons, see Appendix E.3. For ablation studies on the depth, width, and architectures of the neural network models, and model calibration [Platt et al., 1999], see Appendix E.4.

Note that either baselines such as re-training and AWP, or the proposed dropout method, can only explore *partially* of the entire (true) Rashomon set when the hypothesis space is large, and therefore all estimates obtained from the three strategies are under-estimates of the multiplicity metrics. In Figure E.13 in Appendix E.1, we demonstrate that when the hypothesis space is small (e.g., logistic regression), the re-training and dropout strategies have similar performance of exploring the Rashomon set. Despite that the proposed dropout method may only search *local* models in the Rashomon set, it provides fast lower bounds for multiplicity metric estimates.

## 5 APPLICATIONS USING EFFICIENT EXPLORATION OF THE RASHOMON SET

The efficient dropout technique makes applications regarding the exploration of the Rashomon set computationally feasible. Here, we show two applications of the Rashomon set: (i) mitigating predictive multiplicity using the ensemble method with dropout, and (ii) selecting models that have the smallest multiplicity subject to potential loss deviations. We use the UCI Adult Income dataset [Asuncion & Newman, 2007] following the same training settings in Section 4 for both applications.

**Dropout ensembles to mitigate predictive multiplicity.** Long et al. [2023] have shown that the outputs of ensemble models have less variance and hence suffer less from predictive multiplicity. The main challenge of getting an ensemble model is to obtain multiple models in the Rashomon set. Using dropout, we can efficiently obtain a vast amount of models to compose an ensemble model by averaging their outputs. As shown in Figure 4a, as the number of models in an ensemble

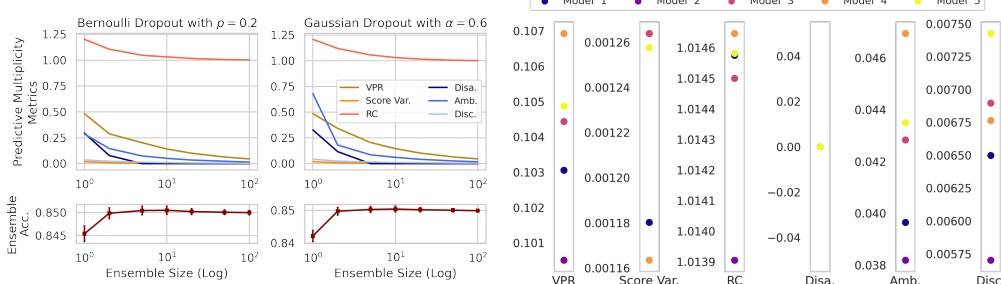

(a) Predictive multiplicity metrics vs. different number of models in an ensemble. A larger ensemble can effectively reduce multiplicity. Each value is averaged over 100 ensemble models.

(b) Predictive multiplicity metrics of 5 different pre-trained models using bernoulli$(1 - 0.01)$ dropout. Model 2 is preferred since it has smallest estimate of multiplicity metrics over the rest of the models.

Figure 4: Applications of the Rashomon set using dropout with the Adult Income dataset.

(Ensemble Size in the figure) increases, both the score and decision-based predictive multiplicity metrics shrink. Therefore, whenever given a pre-trained model, we can efficiently construct an ensemble with dropout models in order to reduce predictive multiplicity.

**Selecting models with smaller predictive multiplicity.** We can also use our efficient estimation framework to perform model selection if presented with multiple pre-trained models with similar performance. By performing dropout on each pre-trained model to construct the empirical Rashomon set, we can pick the model that yields the smallest estimate of predictive multiplicity metrics. For example, Figure 4b shows the spread of predictive multiplicity metrics for 5 models , trained with different weight initializations and same architectures. Model 2 has the smallest VPR, RC, Ambiguity, and Discrepancy compared to the rest of the models. Conversely, Model 4 has the highest estimate of predictive multiplicity metrics. Therefore, Model 2 would be a more favorable choice for deployment over Model 4.

## 6 DISCUSSION

Here we reflect on the limitations and highlight interesting avenues for future work.

**Limitations.** First, in order to apply dropout on the model weights, our proposed framework requires full access to the weights of the pre-trained model as it multiplies them with the dropout random variables, i.e., the pre-trained model has to be a white-box model. However, white-box pre-trained models are often unavailable due to practical reasons such as data security and IP protection. Second, as discussed in E.3, when the hypothesis space is extremely large, dropout method could possibly only explore models that are "close" to (and dependent on) the pre-trained model and therefore under-perform the re-training strategy. A potential solution to this issue is to combine the re-training strategy and the dropout methods—we first re-train a few models to ensure a sufficient exploration of diverse local minima, and then apply dropout on these models to further explore the Rashomon set —see Figure E.22 and E.23 in Appendix E.3 for a demonstration of such strategy.

**Future directions.** First, our analysis of the connection between dropout and the Rashomon set is primarily on FFNNs and CNNs. The analysis could be generalized to other neural network architectures with feedback loops (i.e., recurrent neural networks [Yu et al., 2019]) in order to investigate predictive multiplicity for other applications such as time series or natural languages. Second, our analysis assumes equal dropout probabilities for all weights in a neural network. However, dropout on the first few layers (or some unfrozen layers) may better explore the Rashomon set, as some layers carry more semantics information [Yosinski et al., 2014]. Third, disagreements among models in the empirical Rashomon set can be used to falsify or improve at least one (with the lowest loss) of the models by reconciling the conflicting decisions [Roth et al., 2023]. Given models with different dropout parameters, a possible procedure to reconcile the conflicting decisions could be assigning different weights to those decisions, depending on the dropout parameters and neural network properties.

**Ethics statement.** The phenomenon of the Rashomon effect and predictive multiplicity can be maliciously exploited by adversaries to make harmful decisions towards targeted users. For example, an adversary can measure predictive multiplicity across different training procedures, e.g., different neural network architectures or order of batches of the training dataset, and select a combination of settings that produces the most "unfair" decisions against a certain group of population.

**Reproducibility statement.** Our codes are based on PyTorch [Paszke et al., 2017], where the Bernoulli and Gaussian dropouts could be implemented by intermediate activations with forward hooks. Note that our implementation is different from Gal & Ghahramani [2016] in `https://github.com/yaringal/DropoutUncertaintyExps/tree/master`. The estimation of the predictive multiplicity metrics follows directly from either the corresponding mathematical definitions in the papers or the GitHub repository therein. See Appendix B for the mathematical formulations, corresponding definitions in each reference and the GitHub repositories.

**Disclaimer.** This paper was prepared for informational purposes by the Global Technology Applied Research center of JPMorgan Chase & Co. This paper is not a product of the Research Department of JPMorgan Chase & Co. or its affiliates. Neither JPMorgan Chase & Co. nor any of its affiliates makes any explicit or implied representation or warranty and none of them accept any liability in connection with this paper, including, without limitation, with respect to the completeness, accuracy, or reliability of the information contained herein and the potential legal, compliance, tax, or accounting effects thereof. This document is not intended as investment research or investment advice, or as a recommendation, offer, or solicitation for the purchase or sale of any security, financial instrument, financial product or service, or to be used in any way for evaluating the merits of participating in any transaction.

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

The appendix is divided into the following parts. Appendix A: Omitted proofs and theoretical results; Appendix B: Discussion on predictive multiplicity metrics; Appendix C: Additional discussion on prediction uncertainty; Appendix D: Details on the experimental setup; and Appendix E: Additional empirical results.

## A    OMITTED PROOFS AND THEORETICAL RESULTS

### A.1    USEFUL LEMMAS

We first introduce (and prove) the following useful lemmas to facilitate the proofs of the propositions. The first lemma is about the expectation of the dropout matrix $\mathbf{D_z}$.

**Lemma A.1.** *Let* $\mathbf{D_z} = \mathrm{diag}(Z_1, Z_2, \cdots, Z_d) \in \mathbb{R}^{d \times d}$*, where* $Z_i$ *are i.i.d. random variables, and given a vector* $\mathbf{a} \in \mathbb{R}^d$*, then*

$$\mathbb{E}\left[\mathbf{a}^\top (\mathbf{D_z} - \mathbf{I}_d)^\top (\mathbf{D_z} - \mathbf{I}_d)\mathbf{a}\right] = \begin{cases} \|\mathbf{a}\|_2^2 p, & \text{if } Z_i \overset{i.i.d.}{\sim} \text{ bernoulli}(1-p); \\ \|\mathbf{a}\|_2^2 \alpha, & \text{if } Z_i \overset{i.i.d.}{\sim} \text{ Gaussian}(1, \alpha). \end{cases} \tag{A.1}$$

*Moreover, if* $\mathbf{a}$ *is the all-one* $d$*-dimensional vector, (A.1) can be alternatively expressed as*

$$\mathbb{E}\left[\|\mathbf{D_z} - \mathbf{I}_d\|_F^2\right] = \begin{cases} dp, & \text{if } Z_i \overset{i.i.d.}{\sim} \text{ bernoulli}(1-p); \\ d\alpha, & \text{if } Z_i \overset{i.i.d.}{\sim} \text{ Gaussian}(1, \alpha). \end{cases} \tag{A.2}$$

*Proof.* By direct computation of the expectation, we have

$$\mathbb{E}\left[\mathbf{a}^\top (\mathbf{D_z} - \mathbf{I}_d)^\top (\mathbf{D_z} - \mathbf{I}_d)\mathbf{a}\right] = \mathbb{E}\left[\sum_{i=1}^{d}(\mathbf{a}_i)^2(Z_i - 1)^2\right] = \mathbb{E}\left[(Z_i - 1)^2\right]\sum_{i=1}^{d}(\mathbf{a}_i)^2$$
$$= \mathbb{E}\left[(Z_i - 1)^2\right]\|\mathbf{a}\|_2^2. \tag{A.3}$$

Now, we discuss different distributions for the random variable $Z_i$.

**Bernoulli random variable.** If $Z_i \sim \text{bernoulli}(1-p)$, then $Z_i - 1 \sim 2\text{bernoulli}(p) - 1$ and $(Z_i - 1)^2 \sim \text{bernoulli}(p)$. Therefore,

$$\mathbb{E}\left[(Z_i - 1)^2\right] = \mathbb{E}[Z_i'] = p, \text{ where } Z_i' \sim \text{bernoulli}(p). \tag{A.4}$$

**Gaussian random variable.** If $Z_i \sim \text{Gaussian}(1, \alpha)$, then $Z_i - 1 \sim \sqrt{\alpha}S_i$, where $S_i \sim \text{Gaussian}(0, 1)$ follows the standard Gaussian distribution. Therefore,

$$\mathbb{E}\left[(Z_i - 1)^2\right] = \mathbb{E}\left[\left(\sqrt{\alpha}S_i\right)^2\right] = \alpha\mathbb{E}\left[S_i^2\right] = \alpha\left(\text{var}(S_i) + \mathbb{E}[S_i]^2\right) = \alpha. \tag{A.5}$$

The desired result follows from combining (A.3), (A.4) and (A.5).

Finally, if $\mathbf{a}$ is the all-one $d$-dimensional vector,

$$\mathbb{E}\left[\mathbf{a}^\top (\mathbf{D_z} - \mathbf{I}_d)^\top (\mathbf{D_z} - \mathbf{I}_d)\mathbf{a}\right] = \mathbb{E}\left[\sum_{i=1}^{d}(Z_i - 1)^2\right] = \mathbb{E}\left[\sum_{i=1}^{d}\sum_{j=1}^{d}[\mathbf{D_z} - \mathbf{I}_d]_{i,j}^2\right]$$
$$= \mathbb{E}\left[\|\mathbf{D_z} - \mathbf{I}_d\|_F^2\right], \tag{A.6}$$

and the rest follows by directly plugging in $\|\mathbf{a}\|_2^2 = d$.  □

The second lemma is on the upper bound of the neural network outputs after dropout.

**Lemma A.2.** *Given the dropout matrices* $\mathbf{D}_1, \cdots, \mathbf{D}_K$*, weight matrices* $\mathbf{W}_1, \cdots, \mathbf{W}_K$*, input* $\mathbf{x}$ *and the activation function with Lipschitz constant 1, we have*

$$\|\sigma\left(\mathbf{D}_K\mathbf{W}_K\sigma\left(\cdots\sigma\left(\mathbf{D}_1\mathbf{W}_1\mathbf{x}\right)\cdots\right)\right)\|_2^2 \leq \left(\prod_{k=1}^{K}\|\mathbf{W}_k\|_F^2\right)\left(\prod_{k=1}^{K}\left(\|\mathbf{D}_k - \mathbf{I}_d\|_F^2 + 1\right)\right)\|\mathbf{x}\|_2^2. \tag{A.7}$$

*Proof.* We prove this lemma by induction on the number of layers $k$. First, when $k = 1$, we have

$$
\begin{aligned}
\|\sigma\left(\mathbf{D}_1\mathbf{W}_1\mathbf{x}\right)\|_2^2 &= \|\sigma\left((\mathbf{D}_1\mathbf{W}_1)^\top\mathbf{x}\right) - \sigma\left(\mathbf{W}_1^\top\mathbf{x}\right) + \sigma\left(\mathbf{W}_1^\top\mathbf{x}\right)\|_2^2 \\
&\leq \|\sigma\left((\mathbf{D}_1\mathbf{W}_1)^\top\mathbf{x}\right) - \sigma\left(\mathbf{W}_1^\top\mathbf{x}\right)\|_2^2 + \|\sigma\left(\mathbf{W}_1^\top\mathbf{x}\right)\|_2^2 \qquad\text{(A.8)} \\
&\leq \|\mathbf{W}_1\|_F^2\|\mathbf{x}\|_2^2\left(\|\mathbf{D}_1 - \mathbf{I}_d\|_F^2 + 1\right).
\end{aligned}
$$

Now, let $h_{\mathbf{W}_D}^K(\mathbf{x})$ follows the definition in (8), and suppose (A.7) holds for $k = K - 1$, i.e.,

$$
\|\sigma\left(h_{\mathbf{W}_D}^{K-1}(\mathbf{x})\right)\|_2^2 \leq \left(\prod_{k=1}^{K-1}\|\mathbf{W}_k\|_F^2\right)\left(\prod_{k=1}^{K-1}\left(\|\mathbf{D}_k - \mathbf{I}_d\|_F^2 + 1\right)\right)\|\mathbf{x}\|_2^2, \qquad\text{(A.9)}
$$

we have

$$
\begin{aligned}
\|\sigma\left(h_{\mathbf{W}_D}^K(\mathbf{x})\right)\|_2^2 &= \|\sigma\left((\mathbf{D}_K\mathbf{W}_K)^\top\sigma\left(h_{\mathbf{W}_D}^{K-1}(\mathbf{x})\right)\right)\|_2^2 \\
&\leq \|\sigma\left((\mathbf{D}_K\mathbf{W}_K)^\top\sigma\left(h_{\mathbf{W}_D}^{K-1}(\mathbf{x})\right)\right) - \sigma\left(\mathbf{W}_K^\top\sigma\left(h_{\mathbf{W}_D}^{K-1}(\mathbf{x})\right)\right)\|_2^2 \\
&\quad + \|\sigma\left(\mathbf{W}_K^\top\sigma\left(h_{\mathbf{W}_D}^{K-1}(\mathbf{x})\right)\right)\|_2^2 \\
&\leq \|\mathbf{D}_K - \mathbf{I}_d\|_F^2\|\mathbf{W}_K\|_F^2\|\sigma\left(h_{\mathbf{W}_D}^{K-1}(\mathbf{x})\right)\|_2^2 + \|\mathbf{W}_K\|_F^2\|\sigma\left(h_{\mathbf{W}_D}^{K-1}(\mathbf{x})\right)\|_2^2 \\
&\leq \left(\|\mathbf{D}_K - \mathbf{I}_d\|_F^2 + 1\right)\|\mathbf{W}_K\|_F^2\|\sigma\left(h_{\mathbf{W}_D}^{K-1}(\mathbf{x})\right)\|_2^2 \\
&\leq \left(\|\mathbf{D}_K - \mathbf{I}_d\|_F^2 + 1\right)\|\mathbf{W}_K\|_F^2\left(\prod_{k=1}^{K-1}\|\mathbf{W}_k\|_F^2\right)\left(\prod_{k=1}^{K-1}\left(\|\mathbf{D}_k - \mathbf{I}_d\|_F^2 + 1\right)\right)\|\mathbf{x}\|_2^2 \\
&= \left(\prod_{k=1}^{K}\|\mathbf{W}_k\|_F^2\right)\left(\prod_{k=1}^{K}\left(\|\mathbf{D}_k - \mathbf{I}_d\|_F^2 + 1\right)\right)\|\mathbf{x}\|_2^2.
\end{aligned}
$$

$$\text{(A.10)}$$

$\square$

The third lemma essentially says that the distance of points, sampled from a $d$-dimensional Gaussian distribution, is tightly concentrated around the distance $\sqrt{d}$. The quantity $\sqrt{d}$ is also called natural scale or radius of $d$-dimensional Gaussian distributions.

**Lemma A.3** (Gaussian annulus theorem [Blum et al., 2020, Theorem 2.9]). *For a $d$-dimensional Gaussian with mean zero and variance $\sigma^2\mathbf{I}_d$. for any $\beta \leq \sigma\sqrt{d}$, all but at most $3e^{-\frac{\beta^2}{8}}$ of the probability mass lies within the annulus $\beta - \sigma\sqrt{d} \leq |\mathbf{x}| \leq \beta + \sigma\sqrt{d}$.*

The fourth lemma is a direct application of Lemma A.3 by choosing $\beta = \sigma\sqrt{d}$.

**Lemma A.4.** *Consider a $d$-dimensional Gaussian distribution $P_{\mathcal{N}_d}(\mathbf{v})$ with mean $\mathbf{w} \in \mathbb{R}^d$ and variance $\sigma^2\mathbf{I}_d \in \mathbb{R}^{d\times d}$, and a unit ball $\mathcal{B}$ centered at $\mathbf{w}$ with radius $2\sigma\sqrt{d}$, i.e., $\mathcal{B}(\mathbf{w}, 2\sigma\sqrt{d}) = \{\mathbf{v} \in \mathbb{R}^d; 0 \leq |\mathbf{v}| \leq 2\sigma\sqrt{d}\}$, we have the following integral*

$$
\int_{\mathbf{v}\in\mathcal{B}(\mathbf{w},2\sigma\sqrt{d})} P_{\mathcal{N}_d}(\mathbf{v})d\mathbf{v} \geq 1 - 3e^{-\frac{\sigma^2 d}{8}} \text{ and } \int_{\mathbf{v}\in\mathcal{B}^c(\mathbf{w},2\sigma\sqrt{d})} P_{\mathcal{N}_d}(\mathbf{v})d\mathbf{v} \leq 3e^{-\frac{\sigma^2 d}{8}}, \qquad\text{(A.11)}
$$

*where the latter integral could be viewed as the tail probability bound on the $d$-dimensional ball with radius $2\sigma\sqrt{d}$.*

### A.2 PROOF OF PROPOSITION 1

For any weight vector $\mathbf{w} \in \mathcal{W}$, denote the dropout weights as $\mathbf{w}_D = \mathbf{D}_\mathbf{z}\mathbf{w}$, the mean square error (MSE) is

$$
\begin{aligned}
\|\mathbf{X}\mathbf{w}_D - \mathbf{y}\|_2^2 &= (\mathbf{X}\mathbf{w}_D - \mathbf{y})^\top(\mathbf{X}\mathbf{w}_D - \mathbf{y}) = \mathbf{y}^\top\mathbf{y} - 2\mathbf{w}_D^\top\mathbf{X}^\top\mathbf{y} + \mathbf{w}_D^\top\mathbf{X}^\top\mathbf{X}\mathbf{w}_D \\
&= \mathbf{y}^\top\mathbf{y} - 2\mathbf{w}^\top\mathbf{D}_\mathbf{z}^\top\mathbf{X}^\top\mathbf{y} + \mathbf{w}^\top\mathbf{D}_\mathbf{z}^\top\mathbf{X}^\top\mathbf{X}\mathbf{D}_\mathbf{z}\mathbf{w}.
\end{aligned} \qquad\text{(A.12)}
$$

Taking the expectation, we have

$$
\begin{aligned}
\mathbb{E}\left[\|\mathbf{X}\mathbf{w}_D - \mathbf{y}\|_2^2\right] &= \mathbf{y}^\top\mathbf{y} - 2\mathbf{w}^\top\mathbb{E}\left[\mathbf{X}\mathbf{D_z}\right]^\top\mathbf{y} + \mathbf{w}^\top\mathbb{E}\left[\mathbf{X}\mathbf{D_z}\right]^\top\mathbb{E}\left[\mathbf{X}\mathbf{D_z}\right]\mathbf{w} \\
&= \mathbf{y}^\top\mathbf{y} - 2(1-p)\mathbf{w}^\top\mathbf{X}^\top\mathbf{y} + \mathbf{w}^\top\mathbb{E}\left[\mathbf{X}\mathbf{D_z}\right]^\top\mathbb{E}\left[\mathbf{X}\mathbf{D_z}\right]\mathbf{w} \\
&= \|(1-p)\mathbf{X}\mathbf{w} - \mathbf{y}\|_2^2 - (1-p)^2\mathbf{w}^\top\mathbf{X}^\top\mathbf{X}\mathbf{w} + \mathbf{w}^\top\mathbb{E}\left[\mathbf{X}\mathbf{D_z}\right]^\top\mathbb{E}\left[\mathbf{X}\mathbf{D_z}\right]\mathbf{w} \\
&= \|(1-p)\mathbf{X}\mathbf{w} - \mathbf{y}\|_2^2 + \mathbf{w}^\top\left(\mathbb{E}\left[\mathbf{X}\mathbf{D_z}\right]^\top\mathbb{E}\left[\mathbf{X}\mathbf{D_z}\right] - (1-p)^2\mathbf{X}^\top\mathbf{X}\right)\mathbf{w} \\
&= \|(1-p)\mathbf{X}\mathbf{w} - \mathbf{y}\|_2^2 + p(1-p)\mathbf{w}^\top\mathsf{diag}(\mathbf{X}^\top\mathbf{X})\mathbf{w},
\end{aligned}
$$
(A.13)

where the last equality comes from the fact that the off-diagonal entries are cancelled out, and that for Bernoulli$(1-p)$ random variables, the second moment is $p(1-p) + (1-p)^2 = 1-p$, and therefore $(1-p) - (1-p)^2 = p(1-p)$.

Since the equality above holds for all weights, let $\mathbf{w} = \mathbf{w}^*$ and $\mathbf{w}_D = \mathbf{D_z}\mathbf{w}^*$, we have

$$
\begin{aligned}
\mathbb{E}_{\mathbf{z}\sim\mathsf{bernoulli}(1-p)}[\epsilon] &= \mathbb{E}_{\mathbf{z}\sim\mathsf{bernoulli}(1-p)}\left[\|\mathbf{X}\mathbf{w}_D - \mathbf{y}\|_2^2 - \|(1-p)\mathbf{X}\mathbf{w}^* - \mathbf{y}\|_2^2\right] \\
&= \mathbb{E}_{\mathbf{z}\sim\mathsf{bernoulli}(1-p)}\left[\|\mathbf{X}\mathbf{w}_D - \mathbf{y}\|_2^2\right] - \|(1-p)\mathbf{X}\mathbf{w}^* - \mathbf{y}\|_2^2 \\
&= p(1-p)\mathbf{w}^{*\top}\mathsf{diag}(\mathbf{X}^\top\mathbf{X})\mathbf{w}^*,
\end{aligned}
$$
(A.14)

which is the desired result.

If the features of data matrix $\mathbf{X}$ are linearly independent and normalized, i.e., $\mathbf{X}^\top\mathbf{X} = \mathbf{I}_d$, we have $\mathsf{diag}(\mathbf{X}^\top\mathbf{X}) = \mathbf{I}_d$ and $\mathbf{w}^* = (\mathbf{X}^\top\mathbf{X} + \lambda\mathbf{I}_d)^{-1}\mathbf{X}^\top\mathbf{y} = \frac{\mathbf{X}^\top\mathbf{y}}{1+\lambda}$. The expectation of $\epsilon$ can then be further simplified as

$$
\begin{aligned}
\mathbb{E}_{\mathbf{z}\sim\mathsf{bernoulli}(1-p)}[\epsilon] &= p(1-p)\mathbf{w}^{*\top}\mathsf{diag}(\mathbf{X}^\top\mathbf{X})\mathbf{w}^* = p(1-p)\mathbf{w}^{*\top}\mathbf{w}^* = \frac{p(1-p)}{(1+\lambda)^2}\mathbf{y}^\top\mathbf{X}\mathbf{X}^\top\mathbf{y} \\
&= \frac{p(1-p)}{(1+\lambda)^2}\|\mathbf{y}\|_2^2.
\end{aligned}
$$
(A.15)

## A.3 PROOF OF PROPOSITION 2

Following the definition of the Rashomon set for ridge regression in (4) and the assumption $\mathbf{X}^\top\mathbf{X} = \mathbf{I}_d$, we have

$$
\begin{aligned}
\Pr\left\{\mathbf{D_z}\mathbf{w}^* \in \mathcal{R}(\mathbf{w}^*, \epsilon)\right\} &= \Pr\left\{(\mathbf{D_z}\mathbf{w}^* - \mathbf{w}^*)^\top(\mathbf{X}^\top\mathbf{X} + \lambda\mathbf{I}_d)(\mathbf{D_z}\mathbf{w}^* - \mathbf{w}^*) \le \epsilon\right\} \\
&= \Pr\left\{((\mathbf{D_z} - \mathbf{I}_d)\mathbf{w}^*)^\top(1+\lambda)\mathbf{I}_d(\mathbf{D_z} - \mathbf{I}_d)\mathbf{w}^* \le \epsilon\right\} \\
&= \Pr\left\{\mathbf{w}^{*\top}(\mathbf{D_z} - \mathbf{I}_d)^\top(\mathbf{D_z} - \mathbf{I}_d)\mathbf{w}^* \le \frac{\epsilon}{1+\lambda}\right\} \\
&\ge 1 - (1+\lambda)\frac{\mathbb{E}\left[\mathbf{w}^{*\top}(\mathbf{D_z} - \mathbf{I}_d)^\top(\mathbf{D_z} - \mathbf{I}_d)\mathbf{w}^*\right]}{\epsilon}
\end{aligned}
$$
(A.16)

where the last inequality comes from Markov inequality.

Now, using Lemma A.1 for different dropout random variables $Z_i$, the assumption $\|\mathbf{w}^*\|_2^2 \le M$, and (A.16), we have

$$
\begin{aligned}
\Pr\left\{\mathbf{D_z}\mathbf{w}^* \in \mathcal{R}(\mathbf{w}^*, \epsilon)\right\} &\ge 1 - (1+\lambda)\frac{\mathbb{E}\left[\mathbf{w}^{*\top}(\mathbf{D_z} - \mathbf{I}_d)^\top(\mathbf{D_z} - \mathbf{I}_d)\mathbf{w}^*\right]}{\epsilon} \\
&= \begin{cases} 1 - (1+\lambda)\frac{pM}{\epsilon}, & \text{if } Z_i \overset{i.i.d.}{\sim} \mathsf{bernoulli}(1-p); \\ 1 - (1+\lambda)\frac{\alpha M}{\epsilon}, & \text{if } Z_i \overset{i.i.d.}{\sim} \mathsf{Gaussian}(1, \alpha). \end{cases}
\end{aligned}
$$
(A.17)

If $p = O(d^{-\delta})$ and $\alpha = O(d^{-\delta})$ with $\delta > 0$, we have

$$
\lim_{d\to\infty}(1+\lambda)\frac{pM}{\epsilon} = \lim_{d\to\infty}(1+\lambda)\frac{\alpha M}{\epsilon} = 0.
$$
(A.18)

Therefore, the desired results follow for both Bernoulli and Gaussian dropout mechanisms, i.e.,

$$
\lim_{d\to\infty}\Pr\left\{\mathbf{D_z}\mathbf{w}^* \in \mathcal{R}(\mathbf{w}^*, \epsilon)\right\} = 1.
$$
(A.19)

## A.4 PROOF OF PROPOSITION 3

Follow the definition of the Rashomon set and using tiangle and Cauchy–Schwarz inequalities, we have

$$
\begin{aligned}
L_{\mathsf{BS}}(\mathbf{D_z w}^*) - L_{\mathsf{BS}}(\mathbf{w}^*) &= \frac{1}{n} \sum_{i=1}^{n} \left( \sigma((\mathbf{D_z w}^*)^\top \mathbf{x}_i) - \mathbf{y}_i \right)^2 - \frac{1}{n} \sum_{i=1}^{n} \left( \sigma(\mathbf{w}^{*\top} \mathbf{x}_i) - \mathbf{y}_i \right)^2 \\
&\leq \frac{1}{n} \sum_{i=1}^{n} \left( \sigma((\mathbf{D_z w}^*)^\top \mathbf{x}_i) - \sigma(\mathbf{w}^{*\top} \mathbf{x}_i) \right)^2 \\
&\leq \frac{1}{n} \sum_{i=1}^{n} \left( \mathbf{w}^{*\top} (\mathbf{D_z} - \mathbf{I}_d)^\top \mathbf{x}_i \right)^2 \\
&\leq \frac{1}{n} \sum_{i=1}^{n} \|\mathbf{w}^*\|_2^2 \|\mathbf{D_z} - \mathbf{I}_d\|_F^2 \|\mathbf{x}_i\|_2^2 \\
&= M \|\mathbf{D_z} - \mathbf{I}_d\|_F^2 \overline{\|\mathbf{x}\|_2^2},
\end{aligned}
\tag{A.20}
$$

where $\overline{\|\mathbf{x}\|_2^2} \triangleq \frac{1}{n} \sum_{i=1}^{n} \|\mathbf{x}_i\|_2^2$ and $\|\mathbf{w}^*\|_2^2 \leq M$. Therefore, the probability that the dropout weights lead to models in the Rashomon set is

$$
\begin{aligned}
\Pr\left\{ L_{\mathsf{BS}}(\mathbf{D_z w}^*) - L_{\mathsf{BS}}(\mathbf{w}^*) \leq \epsilon \right\} &\geq \Pr\left\{ M \|\mathbf{D_z} - \mathbf{I}_d\|_F^2 \overline{\|\mathbf{x}\|_2^2} \leq \epsilon \right\} \\
&= \Pr\left\{ \|\mathbf{D_z} - \mathbf{I}_d\|_F^2 \leq \frac{\epsilon}{M \overline{\|\mathbf{x}\|_2^2}} \right\} \\
&\geq 1 - \frac{M \overline{\|\mathbf{x}\|_2^2} \mathbb{E}[\|\mathbf{D_z} - \mathbf{I}_d\|_F^2]}{\epsilon},
\end{aligned}
\tag{A.21}
$$

where the inequality comes from the Markov inequality.

Using Lemma A.1 and (A.21), we have

$$
\begin{aligned}
\Pr\left\{ L_{\mathsf{BS}}(\mathbf{D_z w}^*) - L_{\mathsf{BS}}(\mathbf{w}^*) \leq \epsilon \right\} &\geq 1 - \frac{M \mathbb{E}[\|\mathbf{D_z} - \mathbf{I}_d\|_F^2 \overline{\|\mathbf{x}\|_2^2}]}{\epsilon} \\
&= \begin{cases} 1 - (1+\lambda) \frac{p M d \overline{\|\mathbf{x}\|_2^2}}{\epsilon}, & \text{if } Z_i \overset{i.i.d.}{\sim} \text{bernoulli}(1-p); \\ 1 - (1+\lambda) \frac{\alpha M d \overline{\|\mathbf{x}\|_2^2}}{\epsilon}, & \text{if } Z_i \overset{i.i.d.}{\sim} \text{Gaussian}(1,\alpha). \end{cases}
\end{aligned}
\tag{A.22}
$$

Therefore, as long as $p$ and $\alpha$ are of order $O(d^{-(1+\delta)})$, $\delta > 0$, we have

$$
\lim_{d\to\infty} (1+\lambda) \frac{p M d \overline{\|\mathbf{x}\|_2^2}}{\epsilon} = \lim_{d\to\infty} (1+\lambda) \frac{\alpha M d \overline{\|\mathbf{x}\|_2^2}}{\epsilon} = 0,
\tag{A.23}
$$

and $\lim_{d\to\infty} \Pr\left\{ L_{\mathsf{BS}}(\mathbf{D_z w}^*) - L_{\mathsf{BS}}(\mathbf{w}^*) \leq \epsilon \right\} = 1$ for both Bernoulli and Gaussian dropout mechanisms.

## A.5 PROOF OF PROPOSITION 4

We prove this proposition by the induction method on the number of layers $k \in [K]$. We first consider a neural network $h_{\mathbf{W}}^2(\cdot)$, and derive a bound for the deviation of the outputs (in $\ell_2$-norm)

$\frac{1}{n}\sum_{i=1}^{n}\|h^2_{\mathbf{W}_D}(\mathbf{x}_i) - h^2_{\mathbf{W}}(\mathbf{x}_i)\|_2^2$ before and after dropout. By triangle inequality, we have

$$\frac{1}{n}\sum_{i=1}^{n}\|h^2_{\mathbf{W}_D}(\mathbf{x}_i) - h^2_{\mathbf{W}}(\mathbf{x}_i)\|_2^2 = \frac{1}{n}\sum_{i=1}^{n}\|(\mathbf{D}_2\mathbf{W}_2)^\top\sigma\left((\mathbf{D}_1\mathbf{W}_1)^\top\mathbf{x}_i\right) - \mathbf{W}_2^\top\sigma\left(\mathbf{W}_1^\top\mathbf{x}_i\right)\|_2^2$$

$$\leq \underbrace{\frac{1}{n}\sum_{i=1}^{n}\|(\mathbf{D}_2\mathbf{W}_2)^\top\sigma\left((\mathbf{D}_1\mathbf{W}_1)^\top\mathbf{x}_i\right) - \mathbf{W}_2^\top\sigma\left((\mathbf{D}_1\mathbf{W}_1)^\top\mathbf{x}_i\right)\|_2^2}_{\text{(i)}}$$

$$+ \underbrace{\frac{1}{n}\sum_{i=1}^{n}\|\mathbf{W}_2^\top\sigma\left((\mathbf{D}_1\mathbf{W}_1)^\top\mathbf{x}_i\right) - \mathbf{W}_2^\top\sigma\left(\mathbf{W}_1^\top\mathbf{x}_i\right)\|_2^2}_{\text{(ii)}}.$$

$$(A.24)$$

By Lemma A.2 and let $\overline{\|\mathbf{X}\|_F^2} = \frac{1}{n}\sum_{i=1}^{n}\|\mathbf{x}_i\|_2^2$, we can bound the first term (i) in (A.24) as

$$\text{(i)} = \frac{1}{n}\sum_{i=1}^{n}\|((\mathbf{D}_2 - \mathbf{I}_d)\mathbf{W}_2)^\top\sigma\left((\mathbf{D}_1\mathbf{W}_1)^\top\mathbf{x}_i\right)\|_2^2$$

$$\leq \frac{1}{n}\sum_{i=1}^{n}\|\mathbf{D}_2 - \mathbf{I}_d\|_F^2\|\mathbf{W}_2\|_F^2\|\sigma\left((\mathbf{D}_1\mathbf{W}_1)^\top\mathbf{x}_i\right)\|_2^2 \qquad (A.25)$$

$$\leq \|\mathbf{D}_2 - \mathbf{I}_d\|_F^2\|\mathbf{W}_2\|_F^2\|\mathbf{W}_1\|_F^2\left(\|\mathbf{D}_1 - \mathbf{I}_d\|_F^2 + 1\right)\overline{\|\mathbf{X}\|_F^2}.$$

Similarly, we have the bound on the second term (ii) in (A.24) as

$$\text{(ii)} = \frac{1}{n}\sum_{i=1}^{n}\|\mathbf{W}_2\|_F^2\|(\mathbf{D}_1\mathbf{W}_1)^\top\mathbf{x}_i - \mathbf{W}_1^\top\mathbf{x}_i\|_2^2 \leq \|\mathbf{D}_1 - \mathbf{I}_d\|_F^2\|\mathbf{W}_2\|_F^2\|\mathbf{W}_1\|_F^2\overline{\|\mathbf{X}\|_F^2}.$$

$$(A.26)$$

Combining (A.24), (A.25) and (A.26), we have

$$\frac{1}{n}\sum_{i=1}^{n}\|h^2_{\mathbf{W}_D}(\mathbf{x}_i) - h^2_{\mathbf{W}}(\mathbf{x}_i)\|_2^2 \leq \text{(i)} + \text{(ii)}$$

$$\leq \|\mathbf{D}_2 - \mathbf{I}_d\|_F^2\|\mathbf{W}_2\|_F^2\|\mathbf{W}_1\|_F^2\left(\|\mathbf{D}_1 - \mathbf{I}_d\|_F^2 + 1\right)\overline{\|\mathbf{X}\|_F^2}$$

$$+ \|\mathbf{D}_1 - \mathbf{I}_d\|_F^2\|\mathbf{W}_2\|_F^2\|\mathbf{W}_1\|_F^2\overline{\|\mathbf{X}\|_F^2}$$

$$= \|\mathbf{W}_2\|_F^2\|\mathbf{W}_1\|_F^2\left((\|\mathbf{D}_2 - \mathbf{I}_d\|_F^2 + 1)(\|\mathbf{D}_1 - \mathbf{I}_d\|_F^2 + 1) - 1\right)\overline{\|\mathbf{X}\|_F^2}.$$

$$(A.27)$$

Now, suppose the upper bound holds for a $(K-1)$-hidden-layer feed-forward neural network, i.e.,

$$\frac{1}{n}\sum_{i=1}^{n}\|h^{K-1}_{\mathbf{W}_D}(\mathbf{x}_i) - h^{K-1}_{\mathbf{W}}(\mathbf{x}_i)\|_2^2 \leq \left(\prod_{k=1}^{K-1}\|\mathbf{W}_k\|_F^2\right)\left(\prod_{k=1}^{K-1}\left(\|\mathbf{D}_k - \mathbf{I}_d\|_F^2 + 1\right) - 1\right)\overline{\|\mathbf{X}\|_F^2},$$

$$(A.28)$$

then again by applying Lemma A.2 and (A.28), for $L$ layers, the upper bound is

$$
\begin{aligned}
\frac{1}{n}\sum_{i=1}^{n}\|h_{\mathbf{W}_D}^K(\mathbf{x}_i)-h_{\mathbf{W}}^K(\mathbf{x}_i)\|_2^2 &= \frac{1}{n}\sum_{i=1}^{n}\|(\mathbf{D}_K\mathbf{W}_K)^\top\sigma\left(h_{\mathbf{W}_D}^{K-1}(\mathbf{x}_i)\right)-\mathbf{W}_K\sigma\left(h_{\mathbf{W}}^{K-1}(\mathbf{x}_i)\right)\|_2^2 \\
&\leq \frac{1}{n}\sum_{i=1}^{n}\|(\mathbf{D}_K\mathbf{W}_K)^\top\sigma\left(h_{\mathbf{W}_D}^{K-1}(\mathbf{x}_i)\right)-\mathbf{W}_K^\top\sigma\left(h_{\mathbf{W}_D}^{K-1}(\mathbf{x}_i)\right)\|_2^2 \\
&\quad + \frac{1}{n}\sum_{i=1}^{n}\|\mathbf{W}_K^\top\sigma\left(h_{\mathbf{W}_D}^{K-1}(\mathbf{x}_i)\right)-\mathbf{W}_K\sigma\left(h_{\mathbf{W}}^{K-1}(\mathbf{x}_i)\right)\|_2^2 \\
&\leq \|\mathbf{D}_K-\mathbf{I}_d\|_F^2\|\mathbf{W}_K\|_F^2\left(\frac{1}{n}\sum_{i=1}^{n}\sigma\left(h_{\mathbf{W}_D}^{K-1}(\mathbf{x}_i)\right)\|_2^2\right) \\
&\quad + \|\mathbf{W}_K\|_F^2\left(\frac{1}{n}\sum_{i=1}^{n}\|\sigma\left(h_{\mathbf{W}_D}^{K-1}(\mathbf{x}_i)\right)-\sigma\left(h_{\mathbf{W}}^{K-1}(\mathbf{x}_i)\right)\|_2^2\right) \\
&\leq \|\mathbf{D}_K-\mathbf{I}_d\|_F^2\left(\prod_{k=1}^{K-1}\left(\|\mathbf{D}_k-\mathbf{I}_d\|_F^2+1\right)\right)\left(\prod_{k=1}^{K}\|\mathbf{W}_k\|_F^2\right)\overline{\|\mathbf{X}\|_F^2} \\
&\quad + \left(\prod_{i=1}^{K}\|\mathbf{W}_i\|_F^2\right)\left(\prod_{i=1}^{K-1}\left(\|\mathbf{D}_i-\mathbf{I}_d\|_F^2+1\right)-1\right)\overline{\|\mathbf{X}\|_F^2} \\
&= \left(\prod_{i=1}^{K}\|\mathbf{W}_i\|_F^2\right)\left(\prod_{i=1}^{K}\left(\|\mathbf{D}_i-\mathbf{I}_d\|_F^2+1\right)-1\right)\overline{\|\mathbf{X}\|_F^2}.
\end{aligned}
\tag{A.29}
$$

Since the weights are bounded, i.e., $\|\mathbf{W}_k\|_F^2\leq M$ for all $k\in[K]$, the probability that the output deviation before and after dropout is smaller than $\epsilon$ is given by

$$
\begin{aligned}
\frac{1}{n}\sum_{i=1}^{n}\Pr\left\{\|h_{\mathbf{W}_D}^K(\mathbf{x}_i)-h_{\mathbf{W}}^K(\mathbf{x}_i)\|_2^2\leq\epsilon\right\} &\geq \Pr\left\{\left(\prod_{i=1}^{K}\|\mathbf{W}_i\|_F^2\right)\left(\prod_{i=1}^{K}\left(\|\mathbf{D}_i-\mathbf{I}_d\|_F^2+1\right)-1\right)\overline{\|\mathbf{X}\|_F^2}\leq\epsilon\right\} \\
&= \Pr\left\{\left(\prod_{i=1}^{K}\left(\|\mathbf{D}_i-\mathbf{I}_d\|_F^2+1\right)-1\right)\leq\frac{\epsilon}{\left(\prod_{i=1}^{K}\|\mathbf{W}_i\|_F^2\right)\overline{\|\mathbf{X}\|_F^2}}\right\} \\
&\geq \Pr\left\{\left(\prod_{i=1}^{L}\left(\|\mathbf{D}_i-\mathbf{I}_d\|_F^2+1\right)-1\right)\leq\frac{\epsilon}{M^K\overline{\|\mathbf{X}\|_F^2}}\right\} \\
&\geq 1-\mathbb{E}\left[\prod_{i=1}^{K}\left(\|\mathbf{D}_i-\mathbf{I}_d\|_F^2+1\right)-1\right]\left(\frac{\epsilon}{M^K\overline{\|\mathbf{X}\|_F^2}}\right)^{-1} \\
&= 1-\left(\prod_{i=1}^{K}\mathbb{E}\left[\left(\|\mathbf{D}_i-\mathbf{I}_d\|_F^2+1\right)\right]-1\right)\left(\frac{\epsilon}{M^K\overline{\|\mathbf{X}\|_F^2}}\right)^{-1},
\end{aligned}
\tag{A.30}
$$

where the last equation comes from the fact that the $\mathbf{D}_k$ for different layers are independent.

Let $m = \max_{i \in [L]} m_i$ and since $Z_i \sim \mathsf{Gaussian}(1, \alpha)$, the product in (A.30) becomes

$$
\begin{aligned}
\prod_{i=1}^{K} \mathbb{E}\left[ (\|\mathbf{D}_i - \mathbf{I}_d\|_F^2 + 1) \right] &= \prod_{i=1}^{K} \mathbb{E}\left[ \left( \sum_{j=1}^{m_i} (Z_j - 1)^2 + 1 \right) \right] \leq \prod_{i=1}^{K} \mathbb{E}\left[ \left( \sum_{j=1}^{m} (Z_j - 1)^2 + 1 \right) \right] \\
&= \prod_{i=1}^{K} \left( \sum_{j=1}^{m} \mathbb{E}\left[ (Z_j - 1)^2 \right] + 1 \right) \\
&= \prod_{i=1}^{K} (m\alpha + 1) \\
&= (m\alpha + 1)^K .
\end{aligned}
\tag{A.31}
$$

Combining (A.30) and (A.31), we have

$$
\begin{aligned}
\frac{1}{n} \sum_{i=1}^{n} \Pr\left\{ \|h_{\mathbf{W}_D}^K(\mathbf{x}_i) - h_{\mathbf{W}}^K(\mathbf{x}_i)\|_2^2 \leq \epsilon \right\} \\
\geq 1 - \left( \prod_{i=1}^{K} \mathbb{E}\left[ (\|\mathbf{D}_i - \mathbf{I}_d\|_F^2 + 1) \right] - 1 \right) \left( \frac{\epsilon}{M^K \overline{\|\mathbf{X}\|_F^2}} \right)^{-1} \\
\geq 1 - \left( (m\alpha + 1)^K - 1 \right) \left( \frac{\epsilon}{M^K \overline{\|\mathbf{X}\|_F^2}} \right)^{-1} .
\end{aligned}
\tag{A.32}
$$

In other words, with probability at least $1 - \rho$, the deviation is bounded by

$$
\frac{1}{n} \sum_{i=1}^{n} \|h_{\mathbf{W}_D}^K(\mathbf{x}_i) - h_{\mathbf{W}}^K(\mathbf{x}_i)\|_2^2 \leq \rho^{-1} M^K \overline{\|\mathbf{X}\|_F^2} \left( (m\alpha + 1)^K - 1 \right).
\tag{A.33}
$$

Moreover, as long as $\alpha = O(m^{-(1+\delta)}), \delta > 0$,

$$
\lim_{m \to \infty} \rho^{-1} M^K \overline{\|\mathbf{X}\|_F^2} \left( (m\alpha + 1)^K - 1 \right) = 0.
\tag{A.34}
$$

We finally connect the deviation of the outputs to the loss functions. For regression tasks, the deviation of MSE losses becomes

$$
\begin{aligned}
L_{\mathsf{MSE}}(\mathbf{W}_D) - L_{\mathsf{MSE}}(\mathbf{W}^*) &= \frac{1}{n} \sum_{i=1}^{n} \|h_{\mathbf{W}_D}^K(\mathbf{x}_i) - \mathbf{y}_i\|_2^2 - \frac{1}{n} \sum_{i=1}^{n} \|h_{\mathbf{W}}^K(\mathbf{x}_i) - \mathbf{y}_i\|_2^2 \\
&\leq \frac{1}{n} \sum_{i=1}^{n} \|h_{\mathbf{W}_D}^K(\mathbf{x}_i) - h_{\mathbf{W}}^K(\mathbf{x}_i)\|_2^2.
\end{aligned}
\tag{A.35}
$$

For classification task, the deviation of the Brier scores is

$$
\begin{aligned}
L_{\mathsf{BS}}(\mathbf{w}_D) - L_{\mathsf{BS}}(\mathbf{W}^*) &= \frac{1}{n} \sum_{i=1}^{n} \|\mathsf{softmax}(h_{\mathbf{W}_D}^K(\mathbf{x}_i)) - \mathbf{y}_i\|^2 - \frac{1}{n} \sum_{i=1}^{n} \|\mathsf{softmax}(h_{\mathbf{W}}^K(\mathbf{x}_i)) - \mathbf{y}_i\|^2 \\
&\leq \frac{1}{n} \sum_{i=1}^{n} \|\mathsf{softmax}(h_{\mathbf{W}_D}^K(\mathbf{x}_i)) - \mathsf{softmax}(h_{\mathbf{W}}^K(\mathbf{x}_i))\|^2 \\
&\leq \frac{1}{n} \sum_{i=1}^{n} \|h_{\mathbf{W}_D}^K(\mathbf{x}_i) - h_{\mathbf{W}}^K(\mathbf{x}_i)\|^2.
\end{aligned}
\tag{A.36}
$$

Therefore, combining , for both MSE loss and Brier score, with at least probability at least $1 - \rho$,

$$
\begin{aligned}
L_{\mathsf{MSE}}(\mathbf{W}_D) - L_{\mathsf{MSE}}(\mathbf{W}^*) &\leq \rho^{-1} M^K \overline{\|\mathbf{X}\|_F^2} \left( (m\alpha + 1)^K - 1 \right), \text{ and} \\
L_{\mathsf{BS}}(\mathbf{W}_D) - L_{\mathsf{BS}}(\mathbf{W}^*) &\leq \rho^{-1} M^K \overline{\|\mathbf{X}\|_F^2} \left( (m\alpha + 1)^K - 1 \right).
\end{aligned}
\tag{A.37}
$$

## A.6 Additional notes on theoretical results

Note that the condition in Proposition 2 and 3, $d \to \infty$, is a regime commonly known in the asymptotic analysis of learning models such as over-parameterization [Cao & Gu, 2020], and the universal approximator theorem [Hornik et al., 1989].

The goal to explore the Rashomon set is to find models with diverse outputs while satisfying the loss deviation constraint. If we are allowed to find models outside of the Rashomon set, the performance of the models could be much worse and are less likely to be selected in practice. Note that the phenomenon of predictive multiplicity is only discussed within almost-equally optimal models (say models that all have 99% accuracy). In this sense, the concentration bound helps to characterize the probability that a model after dropout will still be inside the Rashomon set, and is a vital mathematical tool used in our proofs. Ideally, it is more efficient to directly select models at the boundaries of the Rashomon set. It is possible when the Rashomon set could be explicitly expressed, e.g., for ridge regression. However, for general hypothesis space, how to search at the boundaries of a Rashomon set is still an open challenge, and re-training (and AWP) was the only strategy.

The Rashomon set in (2) is defined regarding the mean of the loss for a given dataset of a fixed size. In other words, the dataset is a given parameter, not a source of randomness in the definition of a Rashomon set. Note that the randomness in the convergence in (6) and (9) is with respect to the dropout matrix, not like the vanilla concentration bound that consider drawing samples from a data distribution, it makes sense that those convergence results does not rely on the number of samples. In (5), the loss we used is the sum of the loss for each samples as we w.l.o.g. assume the data matrix is whitened. In (6) and (9) the losses are defined with the mean of the loss for each sample.

## A.7 Additional theoretical results

We show the sample complexity for estimating a surrogate metric for score variance. Using score variance as a predictive multiplicity metric is adopted in Cooper et al. [2023] and Long et al. [2023]; however, they only consider the case of binary classification. Here, we generalize the notion to multi-class classification problems. Since $h_{\mathbf{W}}(\mathbf{x}_i)$ is a $c$-dimensional vector, we can model the distribution of the score $[h_{\mathbf{W}}(\mathbf{x}_i)]_{y_i}$ of the correct label $y_i \in [c]$ as a beta distribution $\mathsf{beta}(\alpha, \beta)$. In this case, the variance of the beta distribution is

$$\frac{\alpha\beta}{(\alpha+\beta)^2(\alpha+\beta+1)} = \frac{\mu(1-\mu)}{\alpha+\beta+1} \leq \mu(1-\mu), \tag{A.38}$$

where $\mu$ is the population mean.

We assume that the models around $\mathbf{W}^*$ are uniformly distributed in a $d$-dimensional ball with center $\mathbf{W}^*$ and radius $\delta$, i.e., $\mathcal{B}(\mathbf{W}^*, \delta)$. Accordingly, we may assume that the population mean $\mu$ for a sample can be expressed as

$$\mu(\mathbf{x}_i) = \mathbb{E}_{\mathbf{W} \sim \mathsf{uniform}(\mathcal{B}(\mathbf{W}^*, \delta))} \left[ [h_{\mathbf{W}}(\mathbf{x}_i)]_{y_i} \right]. \tag{A.39}$$

The assumption of the uniform distribution around $\mathbf{W}^*$ may not reflect the true underlying distributions of models in the Rashomon set (e.g., when there are exponentially many local minima), but it is an ideal assumption that facilitates mathematical analysis that was also adopted in existing literature [Kulynych et al., 2023]. How to characterize the distribution of models in the Rashomon set still remains an open and active challenge in the field. The estimator for $\mu(\mathbf{x}_i)$ by using $T$ Gaussian dropout models is given by

$$\hat{\mu}(\mathbf{x}_i) \triangleq \frac{1}{T} \sum_{t=1}^{T} \left[ h_{\mathbf{W}_{D,t}^*}(\mathbf{x}_i) \right]_{y_i}, \text{ where } \mathbf{W}_{D,t}^* = \{\mathbf{D}_{k,t}\mathbf{W}_k^*\}_{k=1}^K \text{ and } [\mathbf{D}_{k,t}]_{i,i} \sim \mathsf{Gaussian}(1, \alpha).$$
$$\tag{A.40}$$

We pick $v(\mathbf{x}_i) \triangleq \mu(\mathbf{x}_i)(1 - \mu(\mathbf{x}_i))$ as a surrogate metric for the upper bound of the score variance in (A.38), and the plug-in estimator is $\hat{v}(\mathbf{x}_i) \triangleq \hat{\mu}(\mathbf{x}_i)(1 - \hat{\mu}(\mathbf{x}_i))$.

In the following proposition, we show that $\hat{v}(\mathbf{x}_i)$ can be used to estimate $v(\mathbf{x}_i)$ reliably in terms of the number $T$ of models in the empirical Rashomon set (2) by concentration bounds and Gaussian annulus theorem (cf. Lemma A.3).

**Proposition 5.** *For $T$ Gaussian$(1, \alpha)$ dropout models, $h_{\mathbf{W}_{D,1}}, \cdots, h_{\mathbf{W}_{D,T}}$ in the empirical Rashomon set, with probability at least $1 - \rho$, $\rho \in (0, 1]$, the deviation $|\hat{v}(\mathbf{x}_i) - v(\mathbf{x}_i)|$ satisfies*

$$|\hat{v}(\mathbf{x}_i) - v(\mathbf{x}_i)| \leq \left( 6e^{-\frac{\alpha d w_{max}^2}{8}} + \sqrt{\frac{1}{2T} \ln \frac{2}{\rho}} \right) \left( 1 + 6e^{-\frac{\alpha d w_{max}^2}{8}} + \sqrt{\frac{1}{2T} \ln \frac{2}{\rho}} \right), \qquad \text{(A.41)}$$

*where $w_{max} = \max\limits_{k \in [K], i \in [m_{k-1}], j \in [m_k]} [\mathbf{W}_k^*]_{i,j}$ and $d = \sum_{k=1}^K m_{k-1} \times m_k$.*

*Proof.* Since $\hat{v}(\mathbf{x}_i)$ is a continuous transformation of $\hat{\mu}(\mathbf{x}_i)$, we could bound the deviation $|\hat{v}(\mathbf{x}_i) - v(\mathbf{x}_i)|$ by $|\hat{\mu}(\mathbf{x}_i) - \mu(\mathbf{x}_i)|$. Suppose $\hat{\mu}(\mathbf{x}_i) - \mu(\mathbf{x}_i) = \nu$ and $\nu \in [-\eta, \eta]$, we have

$$
\begin{aligned}
|\hat{v}(\mathbf{x}_i) - v(\mathbf{x}_i)| &= |\hat{\mu}(\mathbf{x}_i)(1 - \hat{\mu}(\mathbf{x}_i)) - \mu(\mathbf{x}_i)(1 - \mu(\mathbf{x}_i))| \\
&= |(\mu(\mathbf{x}_i) + \nu)(1 - \mu(\mathbf{x}_i) - \nu) - \mu(\mathbf{x}_i)(1 - \mu(\mathbf{x}_i))| \\
&= \left| \mu(\mathbf{x}_i)(1 - \mu(\mathbf{x}_i)) + \nu(1 - \mu(\mathbf{x}_i)) - \nu\mu(\mathbf{x}_i) - \nu^2 - \mu(\mathbf{x}_i)(1 - \mu(\mathbf{x}_i)) \right| \\
&\leq |\nu| \, |1 - 2\mu(\mathbf{x}_i) - \nu| \\
&\leq |\nu| \, |1 + \nu| .
\end{aligned}
$$
$$\text{(A.42)}$$

Therefore, we only need to bound the deviation $|\hat{\mu}(\mathbf{x}_i) - \mu(\mathbf{x}_i)|$. Here, we have

$$
\begin{aligned}
|\hat{\mu}(\mathbf{x}_i) - \mu(\mathbf{x}_i)| &= |\hat{\mu}(\mathbf{x}_i) - \mathbb{E}[\hat{\mu}(\mathbf{x}_i)] + \mathbb{E}[\hat{\mu}(\mathbf{x}_i)] - \mu(\mathbf{x}_i)| \\
&\leq \underbrace{|\hat{\mu}(\mathbf{x}_i) - \mathbb{E}[\hat{\mu}(\mathbf{x}_i)]|}_{(i)} + \underbrace{|\mathbb{E}[\hat{\mu}(\mathbf{x}_i)] - \mu(\mathbf{x}_i)|}_{(ii)},
\end{aligned}
$$
$$\text{(A.43)}$$

where the expectation is taken over the distribution of all dropout matrices $\mathbf{D}_k$. For (i) in (A.43), using Chernoff-Hoeffding inequality [Hoeffding, 1994], we have

$$\Pr \{ |\hat{\mu}(\mathbf{x}_i) - \mathbb{E}[\hat{\mu}(\mathbf{x}_i)]| < \eta \} \geq 1 - 2\exp(-2\eta^2 T). \qquad \text{(A.44)}$$

In order to bound (ii) in (A.43), let $w_{\max} = \max\limits_{k \in [K], i \in [m_{k-1}], j \in [m_k]} [\mathbf{W}_k^*]_{i,j}$, $d = \sum_{k=1}^K m_{k-1} \times m_k$, $\delta = w_{\max} \sqrt{\alpha d}$ and $p_{\mathcal{N}}$ be the probability density of Gaussian dropout, using Lemma A.4, we have

$$
\begin{aligned}
|\mathbb{E}[\hat{\mu}(\mathbf{x}_i)] - \mu(\mathbf{x}_i)| &= \left| \mathbb{E}[[h_{\mathbf{W}}(\mathbf{x}_i)]_{y_i}] - \mathbb{E}_{\mathbf{W} \sim \text{uniform}(\mathcal{B}(\mathbf{W}^*, \delta))}[[h_{\mathbf{W}}(\mathbf{x}_i)]_{y_i}] \right| \\
&= \left| \int_{\mathbb{R}^{\Pi_{k=1}^K m_k}} [h_{\mathbf{W}}(\mathbf{x}_i)]_{y_i} p_{\mathcal{N}}(\mathbf{W}) d\mathbf{W} - \int_{\mathcal{B}(\mathbf{w}^*, \delta)} [h_{\mathbf{W}}(\mathbf{x}_i)]_{y_i} \frac{1}{\text{vol}(\mathcal{B}(\mathbf{W}^*, \delta))} d\mathbf{W} \right| \\
&\leq \left| \int_{\mathcal{B}(\mathbf{W}^*, \delta)} [h_{\mathbf{W}}(\mathbf{x}_i)]_{y_i} \left( p_{\mathcal{N}}(\mathbf{W}) - \frac{1}{\text{vol}(\mathcal{B}(\mathbf{W}^*, \delta))} \right) d\mathbf{W} \right| \\
&\quad + \left| \int_{\mathcal{B}^c(\mathbf{W}^*, \delta)} [h_{\mathbf{W}}(\mathbf{x}_i)]_{y_i} p_{\mathcal{N}}(\mathbf{W}) d\mathbf{W} \right| \\
&\leq \left| \int_{\mathcal{B}(\mathbf{W}^*, \delta)} \left( p_{\mathcal{N}}(\mathbf{W}) - \frac{1}{\text{vol}(\mathcal{B}(\mathbf{W}^*, \delta))} \right) d\mathbf{W} \right| + \left| \int_{\mathcal{B}^c(\mathbf{W}^*, \delta)} p_{\mathcal{N}}(\mathbf{W}) d\mathbf{W} \right| \\
&\leq \left| \int_{\mathcal{B}(\mathbf{W}^*, \delta)} p_{\mathcal{N}}(\mathbf{W}) d\mathbf{W} - \int_{\mathcal{B}(\mathbf{W}^*, \delta)} \frac{1}{\text{vol}(\mathcal{B}(\mathbf{W}^*, \delta))} d\mathbf{W} \right| + \left| \int_{\mathcal{B}^c(\mathbf{W}^*, \delta)} p_{\mathcal{N}}(\mathbf{W}) d\mathbf{W} \right| \\
&= 1 - \int_{\mathcal{B}(\mathbf{W}^*, \delta)} p_{\mathcal{N}}(\mathbf{W}) d\mathbf{W} + \int_{\mathcal{B}^c(\mathbf{W}^*, \delta)} p_{\mathcal{N}}(\mathbf{W}) d\mathbf{W} \\
&= 2 \int_{\mathcal{B}^c(\mathbf{W}^*, \delta)} p_{\mathcal{N}}(\mathbf{W}) d\mathbf{W} \\
&\leq 6e^{-\frac{\alpha d w_{\max}^2}{8}} .
\end{aligned}
$$
$$\text{(A.45)}$$

Combining (A.43), (A.44) and (A.45), we have

$$
\begin{aligned}
\Pr\left\{|\hat{\mu}(\mathbf{x}_i) - \mu(\mathbf{x}_i)| \le \eta\right\} &\ge \Pr\left\{|\hat{\mu}(\mathbf{x}_i) - \mathbb{E}[\hat{\mu}(\mathbf{x}_i)]| + |\mathbb{E}[\hat{\mu}(\mathbf{x}_i)] - \mu(\mathbf{x}_i)| \le \eta\right\} \\
&\ge \Pr\left\{|\hat{\mu}(\mathbf{x}_i) - \mathbb{E}[\hat{\mu}(\mathbf{x}_i)]| + 6e^{-\frac{\alpha d w_{\max}^2}{8}} \le \eta\right\} \\
&\ge \Pr\left\{|\hat{\mu}(\mathbf{x}_i) - \mathbb{E}[\hat{\mu}(\mathbf{x}_i)]| \le \eta - 6e^{-\frac{\alpha d w_{\max}^2}{8}}\right\} \\
&\ge 1 - 2\exp\left(-2\left(\eta - 6e^{-\frac{\alpha d w_{\max}^2}{8}}\right)^2 T\right).
\end{aligned}
\tag{A.46}
$$

Let $\rho = 2\exp\left(-2\left(\eta - 6e^{-\frac{\alpha d w_{\max}^2}{8}}\right)^2 T\right)$, we have with probability at least $1 - \rho$, $\rho \in (0, 1]$,

$$
|\hat{\mu}(\mathbf{x}_i) - \mu(\mathbf{x}_i)| \le 6e^{-\frac{\alpha d w_{\max}^2}{8}} + \sqrt{\frac{1}{2T}\ln\frac{2}{\rho}}.
\tag{A.47}
$$

Therefore, by plugging in the deviation $|\hat{\mu}(\mathbf{x}_i) - \mu(\mathbf{x}_i)|$ in (A.42), we have with probability at least $1 - \rho$

$$
|\hat{v}(\mathbf{x}_i) - v(\mathbf{x}_i)| \le |\nu|\,|1 + \nu| \le \left(6e^{-\frac{\alpha d w_{\max}^2}{8}} + \sqrt{\frac{1}{2T}\ln\frac{2}{\rho}}\right)\left(1 + 6e^{-\frac{\alpha d w_{\max}^2}{8}} + \sqrt{\frac{1}{2T}\ln\frac{2}{\rho}}\right).
\tag{A.48}
$$

$\square$

Proposition 5 shows the sample complexity of estimating $v(\mathbf{x}_i)$ with $T$ models for a single sample. In practice, one might need to estimate the score variance for multiple samples, e.g., computing average score variance over a test dataset. Naïvely, we need $nT$ models to estimate $v(\mathbf{x}_i)$ for a dataset with $n$ samples. In contrast, we can easily generalize the results in Proposition 5 to $n$ samples by union bounds, and show that in such cases sample complexity grows only logarithmically under mild assumptions. To be precise, since the samples $\mathbf{x}_1, \mathbf{x}_2, \cdots, \mathbf{x}_n$ are i.i.d., we have the following union bound for the concentration of sample mean, i.e.,

$$
\Pr\left\{\bigcup_{i=1}^n \left\{|\hat{\mu}(\mathbf{x}_i) - \mathbb{E}[\hat{\mu}(\mathbf{x}_i)]| \ge \eta\right\}\right\} \le \prod_{i=1}^n \Pr\left\{|\hat{\mu}(\mathbf{x}_i) - \mathbb{E}[\hat{\mu}(\mathbf{x}_i)]| \ge \eta\right\} \le 2n\exp(-2\eta^2 T).
\tag{A.49}
$$

Therefore, with probability $1 - \rho$, for all $i \in [n]$, $|\hat{\mu}(\mathbf{x}_i) - \mathbb{E}[\hat{\mu}(\mathbf{x}_i)]| \le \sqrt{\frac{1}{2T}\ln\frac{2n}{\rho}}$. By following the proof of Proposition 5, with probability $1 - \rho$, for all $i \in [n]$,

$$
|\hat{v}(\mathbf{x}_i) - v(\mathbf{x}_i)| \le \left(6ne^{-\frac{\alpha d w_{\max}^2}{8}} + \sqrt{\frac{1}{2T}\ln\frac{2n}{\rho}}\right)\left(1 + 6ne^{-\frac{\alpha d w_{\max}^2}{8}} + \sqrt{\frac{1}{2T}\ln\frac{2n}{\rho}}\right).
\tag{A.50}
$$

Since the term $e^{-\frac{\alpha d w_{\max}^2}{8}}$ could be made arbitrarily small, it can compensate the linear growth of $n$.

## B   DISCUSSION ON PREDICTIVE MULTIPLICITY METRICS

We give a thorough introduction of predictive multiplicity metrics, including their mathematical formulation, operational meanings, and computational details. Predictive multiplicity metrics can be categorized into two groups: score-based and decision-based, where a decision is a thresholded score or the score vector after argmax. Precisely, consider a binary classification, if we have a score $s$, then the decision can be obtained by $\mathbb{1}[s > \tau]$, where $\tau$ is a threshold and $\mathbb{1}[\cdot]$ is the indicator function. For a $c$-class classification problem where $c > 2$, the score is a vector, say $\mathbf{s} \in \Delta_c$, and the decision can be obtained by $\operatorname*{argmax}_{i \in [c]}[\mathbf{s}]_i$. In the following, we start with the decision-based metrics.

First, consider a Rashomon set $\mathcal{R}$, the pattern Rashomon ratio $r(\mathcal{D})$ of a dataset $\mathcal{D}$ is defined as the ratio of the count of all possible binary predicted classes given by the functions in the Rashomon set to that given by the functions in the hypothesis space [Semenova et al., 2019, Defn. 12], i.e,

$$r(\mathcal{D}) \triangleq \frac{\sum_{i=0}^{2^n-1} \mathbb{1}[\exists h_{\mathbf{w}} \in \mathcal{R}, [\operatorname{argmax} h_{\mathbf{w}}(\mathbf{x}_1), \cdots, \operatorname{argmax} h_{\mathbf{w}}(\mathbf{x}_n)] = \operatorname{binary}(i)]}{\sum_{i=0}^{2^n-1} \mathbb{1}[\exists h_{\mathbf{w}} \in \mathcal{H}, [\operatorname{argmax} h_{\mathbf{w}}(\mathbf{x}_1), \cdots, \operatorname{argmax} h_{\mathbf{w}}(\mathbf{x}_n)] = \operatorname{binary}(i)]}, \quad \text{(B.51)}$$

where $\operatorname{binary}(\cdot)$ denotes the vector of the binary representation of an integer, e.g., $5 \equiv [0, \cdots, 0, 1, 0, 1]$. Computing pattern Rashomon ratio involves a summation with the number of terms grows exponentially fast with the number of samples $n$. Therefore, pattern Rashomon ratio is an "expensive" metric for predictive multiplicity when evaluated on large datasets.

There are several decision-based predictive multiplicity metrics that use the disagreement among decisions given by models in the Rashomon set under difference disguise [Black et al., 2022; D'Amour et al., 2022; Marx et al., 2020]. For example, Marx et al. [2020] propose two metrics: ambiguity and discrepancy. Ambiguity is the proportion of samples in a dataset that can be assigned conflicting predictions by competing classifiers in the Rashomon set. Discrepancy is the maximum number of predictions that could change in a dataset if we were to switch between models within the Rashomon set. More precisely, given a pre-trained model $h_{\mathbf{w}^*}$, the ambiguity $\alpha(\mathcal{D})$ and the discrepancy $\delta(\mathcal{D})$ are respectively defined as [Marx et al., 2020, Definitions 3 and 4]

$$\alpha(\mathcal{D}) \triangleq \frac{1}{|\mathcal{D}|} \sum_{\mathbf{x}_i \in \mathcal{D}} \max_{h_{\mathbf{w}} \in \mathcal{R}} \mathbb{1}\left[\operatorname{argmax} h_{\mathbf{w}}(\mathbf{x}_i) \neq \operatorname{argmax} h_{\mathbf{w}^*}(\mathbf{x}_i)\right]$$

$$\delta(\mathcal{D}) \triangleq \max_{h_{\mathbf{w}} \in \mathcal{R}} \frac{1}{|\mathcal{D}|} \sum_{\mathbf{x}_i \in \mathcal{D}} \mathbb{1}\left[\operatorname{argmax} h_{\mathbf{w}}(\mathbf{x}_i) \neq \operatorname{argmax} h_{\mathbf{w}^*}(\mathbf{x}_i)\right]$$

$$\text{(B.52)}$$

Both ambiguity and discrepancy can be estimated by a mixed integer program [Marx et al., 2020, Section 3].

Moreover, instead of computing the empirical mean of decision disagreement over a dataset $\mathcal{D}$, we may define disagreement directly using the notion of probability. Precisely, Black et al. [2022, Section A.1] and Kulynych et al. [2023, Eq. (4)] propose the following quantity to measure disagreement:

$$\mu(\mathbf{x}_i) \triangleq 2 \Pr\{\mathbb{1}[h_{\mathbf{w}}(\mathbf{x}_i) > \tau] \neq \mathbb{1}[h'_{\mathbf{w}}(\mathbf{x}_i) > \tau]; h_{\mathbf{w}}, h'_{\mathbf{w}} \in \mathcal{R}\}, \quad \text{(B.53)}$$

where $h_{\mathbf{w}}$ and $h'_{\mathbf{w}}$ are any two models in the Rashomon set and the factor 2 ensures that $\mu(\mathbf{x}_i)$ is in the $[0, 1]$ range for the ease of interpretation. Kulynych et al. [2023] further proposed a plug-in estimator to estimate disagreement for binary classification with a sample complexity bound on the number of models obtained by re-training.

On the other hand, score-based metrics focus on the spread of the output scores [Hsu & Calmon, 2022; Watson-Daniels et al., 2023; Long et al., 2023]. Borrowing from information theory, Hsu & Calmon [2022, Definition 2] measures the spread of output scores for $c$-class classification problems in the probability simplex $\Delta_c$ by an analog of channel capacity, termed the Rashomon Capacity, i.e.,

$$c(\mathbf{x}_i) \triangleq \sup_{P_{\mathcal{R}}} \inf_{\mathbf{q} \in \Delta_c} \mathbb{E}_{h_{\mathbf{w}} \sim P_{\mathcal{R}}} D_{\mathsf{KL}}(h_{\mathbf{w}}(\mathbf{x}_i) \| \mathbf{q}), \quad \text{(B.54)}$$

where $P_{\mathcal{R}}$ is the probability distribution over the models in the Rashomon set, and $D_{\mathsf{KL}}(\cdot \| \cdot)$ is the Kullback-Leibler (KL) divergence. The infimum $\inf_{\mathbf{q} \in \Delta_c} \mathbb{E}_{h_{\mathbf{w}} \sim P_{\mathcal{R}}} D_{\mathsf{KL}}(h_{\mathbf{w}}(\mathbf{x}_i) \| \mathbf{q})$ measures (in the sense of KL divergence) the spread of the scores of a sample $\mathbf{x}_i$ given a distribution $P_{\mathcal{R}}$ over

Table B.2: Predictive multiplicity metrics implemented in this paper.

| Metrics | GitHub |
|---------|--------|
| Viable Prediction Range | not included |
| Score std./var. | not included |
| Rashomon Capacity | https://github.com/HsiangHsu/rashomon-capacity |
| Disagreement | https://github.com/spring-epfl/dp_multiplicity |
| Ambiguity
Discrepancy | https://github.com/charliemarx/pmtools |

all all the models $h_{\mathbf{w}}$ in the Rashomon set, where the minimizing $q$ acts as a "centroid" for the outputs of the classifiers. The supremum picks the worst-case distribution $P_{\mathcal{R}}$ over all possible distributions in the Rashomon set. The Rashomon Capacity is closely connected to the information diameter, a metric to measure the diameter of a set of probability distributions [Kemperman, 1974]. Hsu & Calmon [2022] proved that the Rashomon Capacity can be estimated using $c$ models in the Rashomon set, obtained by the adversarial weight perturbation (AWP) algorithm, and the Blahut-Aromoto algorithm [Blahut, 1972; Arimoto, 1972].

Based on ambiguity and discrepancy, Watson-Daniels et al. [2023, Definition 2] proposed Viable Prediction Range (VPR) $v(\mathbf{x}_i)$, which is the largest score deviation of a sample that can be achieved by models in the Rashomon set:

$$v(\mathbf{x}_i) \triangleq \max_{h_{\mathbf{w}} \in \mathcal{R}} h_{\mathbf{w}}(\mathbf{x}_i) - \min_{h_{\mathbf{w}} \in \mathcal{R}} h_{\mathbf{w}}(\mathbf{x}_i). \tag{B.55}$$

The VPR can be computed using similar mixed integer programs in Marx et al. [2020] for binary classification with linear classifiers.

Instead of computing the largest score deviation of a sample, Long et al. [2023, Definition 2] measures the standard deviation (std.) $s(\mathbf{x}_i)$ (and the variance (var.)) of the scores of a sample by all models in the Rashomon set:

$$s(\mathbf{x}_i) \triangleq \sqrt{\mathbb{E}_{h_{\mathbf{w}} \sim P_{\mathcal{R}}}[(h_{\mathbf{w}}(\mathbf{x}_i) - \mathbb{E}_{h_{\mathbf{w}} \sim P_{\mathcal{R}}}[h_{\mathbf{w}}(\mathbf{x}_i)])^2]}. \tag{B.56}$$

The score standard deviation can be estimated using the re-training strategy.

**Computation.** There are some theoretical analyses of estimating predictive multiplicity metrics. For example, the exact estimation of ambiguity and discrepancy for linear classifiers are studied in Marx et al. [2020]. Similarly, the theoretical analysis of estimating VPR with logistic regression is also provided in Watson-Daniels et al. [2023]. Moreover, the estimation of disagreement is also discussed in Kulynych et al. [2023]. However, the statistical properties of estimating other metrics such as the Rashomon Capacity, or with a general hypothesis space, still remain an open challenge. Finally, we summarize the GitHub implementation for computing the predictive multiplicity metrics in Table B.2. For metrics where the implementations are not included, we implement them directly following the mathematical definitions.

**The AWP algorithm.** The Adversarial Weight Perturbation algorithm, as proposed in Hsu & Calmon [2022, Eq. 9] , will adversarially perturb the output scores of a given sample to each class. For example, if there are $c$ classes, AWP maximizes each entries of $h\_w(x)$, i.e., $[h\_w(x)]\_i$ for each $i \in [c]$, under the constraint that the model after perturbation is still inside the Rashomon set. Given a sample $\mathbf{x}_i$, we obtain models with output predictions $\mathbf{p}_k$ by approximately solving the following optimization problem which maximizes the output score for class $k$:

$$\mathbf{p}_k = h_{\hat{\theta}}(\mathbf{x}_i), \text{ where } \hat{\theta} = \underset{h_{\theta} \in \mathcal{R}(\mathcal{H}, \epsilon)}{\operatorname{argmax}} [h_{\theta}(\mathbf{x}_i)]_k, \forall k = 1, 2, \cdots, c. \tag{B.57}$$

To solve equation B.57, for each $k$, we set the objective to be $\min_{\theta \in \Theta} -[h_{\theta}(\mathbf{x}_i)]_k$, compute the gradients, and update the parameter $\theta$ until $L(h_{\theta}) > \epsilon$. Therefore, for one sample, AWP requires to perform one perturbations (adversarial training) via SGD for each class (i.e., $c$ adversarial training in total for one sample), and if there is a dataset with $n$ samples, we need $n \times c$ adversarial training to estimate multiplicity metrics for the entire dataset. We can imagine that each of these $n \times c$

perturbations leads to a model in the Rashomon set that outputs the most inconsistent score for a given class per sample. On the other hand, the dropout method does not require to iterate over all samples and does not require gradient computations and updating model weights. In this sense, AWP is very different from the proposed dropout strategy.

## C ADDITIONAL DISCUSSION ON DROPOUT AND PREDICTION UNCERTAINTY

We introduce dropout inference for prediction uncertainty proposed in Gal & Ghahramani [2016], and discuss its difference from the notion of Rashomon set and predictive multiplicity.

Gal & Ghahramani [2016] proposed a theoretical understanding of dropout at inference with Bayesian approximation. In the Bayesian framework, the conditional probability $p(y|\mathbf{x}, \mathcal{D})$ of the output $y$ for an unseen input $\mathbf{x}$ is viewed as a deep Gaussian process [Damianou & Lawrence, 2013], and can be expressed as

$$p(y|\mathbf{x}, \mathcal{D}) = \int_{\mathbf{w} \in \mathcal{W}} p(y|\mathbf{x}, \mathbf{w})p(\mathbf{w}|\mathcal{D})d\mathbf{w}. \tag{C.58}$$

Here, $p(\mathbf{w}|\mathcal{D})$ is the distribution of the weights given the training set $\mathcal{D}$, and is usually intractable in practice. By substituting the true distribution $p(\mathbf{w}|\mathcal{D})$ with an approximation $q(\mathbf{w})$, the posterior $p(y|\mathbf{x}, \mathcal{D})$ can be computed as

$$p(y|\mathbf{x}, \mathcal{D}) \approx \int_{\mathbf{w} \in \mathcal{W}} p(y|\mathbf{x}, \mathbf{w})q(\mathbf{w})d\mathbf{w}. \tag{C.59}$$

$q(\mathbf{w})$ is assumed to follow a distribution parameterized by $\theta$, and a suitable approximation $q(\mathbf{w})$ can be found by minimizing the KL divergence $\min_{\theta} D_{\mathsf{KL}}(p(\mathbf{w}|\mathcal{D})\|q(\mathbf{w}))$.

Gal & Ghahramani [2016] used dropout for $q(\mathbf{w})$ with single-hidden-layer networks, and interpret dropout inference as as a Monte Carlo sampling that is equivalent to estimate characteristics of the underlying distribution $p(y|\mathbf{x}, \mathcal{D})$ in (C.59). Therefore, the variance of the prediction $y$ can be estimated using $p(y|\mathbf{x}, \mathcal{D})$, and can be taken to indicate the *prediction uncertainty* of the model for a particular input. Later on, Lee et al. [2017] generalized the analysis in Gal & Ghahramani [2016] to multi-hidden-layer neural networks.

Note that the integral in (C.59) is computed over the set of all possible models $p(y|\mathbf{x}, \mathbf{w})$ in the hypothesis space $\mathcal{W}$; however, not all models provide satisfactory performance in terms of loss and accuracy, i.e., not all models are in the Rashomon set. In other words, the posterior we are interested in is

$$p(y|\mathbf{x}, \mathcal{D}) = \int_{\mathbf{w} \in \mathcal{R}(\mathbf{w}^*, \epsilon)} p(y|\mathbf{x}, \mathbf{w})p(\mathbf{w}|\mathcal{D})d\mathbf{w}. \tag{C.60}$$

The difference between the integrals in (C.58) and (C.60) is the main difference between the consideration of prediction uncertainty and predictive multiplicity. In Table C.3, we use the UCI Credit Approval dataset to compare the predictive uncertainty, i.e., score variance evaluated with (C.58), against score variance evaluated with (C.60), where the Rashomon parameter $\epsilon$ is the mean loss deviation under each dropout parameters $p$ for Bernoulli dropout and $\alpha$ for Gaussian dropout. For the sake of illustration, we show the value of the 80% quantile of the score variances across all samples. It is clear that without the consideration of the Rashomon set, both dropout techniques could include models that are not in the Rashomon set (i.e., models that have loss deviations larger than $\epsilon$), and hence over-estimating the score variance.

Table C.3: Comparison between the score variances computed with (C.58) and (C.60) on the Credit Approval dataset. It is clear that using (C.58), i.e., without the consideration of the Rashomon set, the score variance is over-estimated.

| | Bernoulli Dropout | | | Gaussian Dropout | | |
|---|---|---|---|---|---|---|
| Dropout Parameters | $p = 0.020$ | $p = 0.040$ | $p = 0.060$ | $\alpha = 0.120$ | $\alpha = 0.210$ | $\alpha = 0.330$ |
| Rashomon Parameters | $\epsilon = 0.003$ | $\epsilon = 0.006$ | $\epsilon = 0.007$ | $\epsilon = 0.002$ | $\epsilon = 0.005$ | $\epsilon = 0.011$ |
| Score Var. with (C.58) | 0.001131 | 0.002051 | 0.003280 | 0.000385 | 0.001211 | 0.003000 |
| Score Var. with (C.60) | 0.000010 | 0.000114 | 0.001699 | 0.000067 | 0.000285 | 0.000858 |

# D    DETAILS ON THE EXPERIMENTAL SETUP

We summarize the dataset descriptions, training setups and the results.

## D.1    DATASET DESCRIPTION AND PRE-PROCESSING

The datasets used in this paper are the 6 datasets from the UCI machine learning repository [Asuncion & Newman, 2007] and the Microsoft COCO dataset [Lin et al., 2014]. The UCI machine learning repository is a well-known and widely used collection of 650 datasets for machine learning research and experimentation, and contains a diverse and extensive collection of datasets across various domains. We select 6 datasets that may possess critical consequences if predictive multiplicity is not accounted for. The first three datasets are related to financial applications, including Adult Income, Bank Marketing and Credit Approval, and the other three are related to medical, including Dermatology, Mammography and Contraception. The Adult Income dataset aims to predict whether the income of an individual exceeds 50,000 per year based on 1994 census data. The Bank Marketing dataset is related with direct marketing campaigns of a Portuguese banking institution based on phone calls in order to predict if the client will subscribe a bank term deposit or not. The Credit Approval dataset concerns the prediction of successful credit card application with anonymized feature names. The Dermatology dataset aims to determine the differential diagnosis of erythemato-squamous diseases in dermatology. The Mammography dataset aims to discriminate between benign and malignant mammographic masses based on BI-RADS attributes and the patient's age. The Contraception dataset contains samples from married women who were either not pregnant or do not know if they were at the time of interview of the 1987 National Indonesia Contraceptive Prevalence Survey, and the goal is to predict the current contraceptive method choice (no use, long-term methods, or short-term methods) of a woman based on her demographic and socio-economic characteristics. For these datasets, we remove samples with missing values, one-hot encoded nominal features, re-scale numeric features, and set the target label name to be 1 and the rest to be 0. For the number of features, training/test split and the label names, see Table D.4.

The Microsoft COCO (Common Objects in Context) dataset is a benchmark dataset with 200,000 images spanning 80 object categories, comprising a vast collection of high-quality images annotated with rich and detailed object segmentation, captioning, and keypoint information. For the image datasets, we follow standard normalization procedures to normalize the values of each pixel in each color channel.

## D.2    TRAINING SETUPS AND RESULTS.

For the UCI datasets, we use a single-hidden-layer neural network with 1000 hidden neurons, and train for 100 epochs with learning rate 0.001 and batch size 100. The performance of the base models, including training and test losses and accuracy, is included in Table D.5. The re-training strategy follows the same setting but with fewer epochs. For Adult Income dataset, we re-train 100 times for each epoch $20, 25, 30, 35, 40, 45, 50, 55$. For Bank Marketing dataset, we re-train 100 times for each epoch $1, 2, 3, 4, 5, 6, 7, 8, 9$. For Credit Approval dataset, we re-train 100 times for each epoch $35, 40, 45, 50, 55, 60$. For Dermatology dataset, we re-train 100 times for each epoch $1, 2, 3, 4, 5, 6, 7, 8, 9$. For Mammography dataset, we re-train 100 times for each epoch $11, 12, 13, 14, 15, 16, 17, 18, 19$. For Contraception dataset, we re-train 100 times for each epoch $11, 12, 13, 14, 15, 16, 17, 18, 19$. Similarly,

Table D.4: UCI dataset descriptions.

| Dataset | # of features | Training set size | Test set size | Label (# of classes) |
|---|---|---|---|---|
| Adult Income | 104 | 22621 | 7541 | income >50K (2) |
| Bank Marketing | 63 | 30891 | 10297 | has deposit (2) |
| Credit Approval | 46 | 489 | 164 | approved (2) |
| Dermatology | 34 | 268 | 90 | has erythema (2) |
| Mammography | 5 | 622 | 208 | benign or malignant (2) |
| Contraception | 9 | 1104 | 369 | long or short term (2) |

Table D.5: UCI dataset base model results.

| Dataset | Training | | Test | |
|---|---|---|---|---|
| | Loss | Accuracy | Loss | Accuracy |
| Adult Income | 0.3026 | 85.96% | 0.3190 | 85.00% |
| Bank Marketing | 0.1962 | 91.27% | 0.2013 | 91.06% |
| Credit Approval | 0.3149 | 88.14% | 0.2975 | 87.80% |
| Dermatology | 0.1138 | 99.25% | 0.1248 | 98.89% |
| Mammography | 0.3604 | 84.24% | 0.4920 | 79.33% |
| Contraception | 0.5658 | 73.64% | 0.5986 | 69.38% |

for Bernoulli and Gaussian dropouts, we obtain 100 models each with 21 values of $p$ and $\alpha$ spread evenly between $0.2$ and $0.6$ respectively. Finally, we perform adversarial weight perturbation on four datasets (Credit Approval, Dermatology, Mammography and Contraception), with $\epsilon \in \{0.000, 0.004, 0.008, 0.012, 0.016, 0.020, 0.024, 0.028, 0.032, 0.036, 0.040\}$ and the perturbation learning rate is 0.001.

For MS COCO, we adopt two pre-trained models (the YoloV3 [Redmon & Farhadi, 2018] and Mask R-CNN [He et al., 2017] detectors) in MMDetection [Chen et al., 2019] to validate the Rashomon effect on the detection models with the proposed dropout inference. For the YoloV3 detector, we use $p$ ranged from 5e-5 to 1e-4 for Bernoulli dropout and use $\alpha$ ranged from 0.03 to 0.1 for Gaussian dropout; while for the Mask R-CNN detector, we use $p$ ranged from 5e-5 to 3e-4 for Bernoulli dropout and use $\alpha$ ranged from 0.01 to 0.07 for Gaussian dropout. Then, we choose the models whose average precision on person class is within $\epsilon \in \{0.005, 0.006, 0.007, 0.008, 0.009, 0.010, 0.011, 0.012\}$.

# E  ADDITIONAL RESULTS AND EXPERIMENTS

We include (i) the deviations of loss and accuracy versus different dropout parameters, $p$ for Bernoulli dropout and $\alpha$ for Gaussian dropout, (ii) the complete values of the estimated predictive multiplicity metrics, (iii)the comparison of the efficiency between re-training and dropout-based Rashomon set exploration, (iv) addition experiments on CIFAR-10/-100 datasets, and (v) ablation studies.

## E.1  ADDITIONAL RESULTS ON UCI DATASETS

In Figure E.5, we show the deviation of loss and accuracy versus different dropout parameters for both Bernoulli ($p \in [0.0, 0.2]$) and Gaussian ($\alpha \in [0.0, 0.6]$) dropouts. As we can see, the dropout parameters can stably control the loss deviation and the according accuracy changes. Note that for all dropout parameters, the accuracy change is $\approx 1\%$. Moreover, the standard deviations of the loss difference $\epsilon$ and the accuracy are determined by the number of samples in the datasets—not necessarily the accuracy of the base models. For example, the Adult Income and Bank Marketing datasets have the most number of samples $> 20000$, and hence has the smallest standard deviations. On the other hand, despite that the Dermatology dataset also has $98\%$ accuracy, it suffers from a relatively large standard deviation in terms of loss difference and accuracy.

Figure E.6, Figure E.7, Figure E.8, Figure E.9, Figure E.10, and Figure E.11 summarize the cumulative distributions of the predictive multiplicity metrics that are defined across all samples, and Figure E.12 shows the ambiguity and discrepancy, for UCI Adult Income, Bank Marketing, Credit Approval, Dermatology, Mammography, and Contraception datasets [Asuncion & Newman, 2007]. Since all estimates of predictive multiplicity metrics are under-estimate of their true values, and therefore higher values indicate a more effective method for estimation. Observing viable prediction range, score variance, and the Rashomon Capacity, it is clear that both Bernoulli and Gaussian dropouts are more effective in terms of estimating predictive multiplicity metrics compared to re-training. Moreover, Bernoulli and Gaussian dropouts are also comparable to the adversarial weight perturbation algorithm. Note that we do not perform adversarial weight perturbation algorithm on Adult Income and Bank Marketing datasets since the number of samples is too large, which causes extreme time complexity.

Table E.6 summarizes the raw runtime values of four strategies, and the speedups are reported in Table 1. The runtimes for Bernoulli and Gaussian dropouts are in a similar order, and therefore in Table 1 we only report the speedup of Gaussian dropout. The adversarial weight perturbation algorithm costs the most time since it has to scan over all samples and perturb (i.e., train) the weights.

Finally, Table E.7 shows the efficiency of obtaining models from the Rashomon set with different Rashomon parameters $\epsilon$, compared with the re-training strategy. Note that it is hard to control the eventual loss value with the re-training strategy, and therefore we perform the re-training strategy will different number of epochs, in order to obtain models corresponding to different $\epsilon$ in the Rashomon set. On the other hand, it is easier to control $\epsilon$ using dropout techniques, as supported in Figure E.5. For example, when $\epsilon = 0.004$, there are only less than $3\%$ of the models obtained by the re-training strategy that are in the Rashomon set; however, by carefully controlling dropout parameters, there are more than $70\%$ of the models obtained by Gaussian dropout that are in the Rashomon set. Moreover, Gaussian dropout is more "benign" compare to Bernoulli dropout in terms of changing the model weights, since Bernoulli dropout directly removes the weights while Gaussian dropout only scales them.

In Figure E.13, we show the model weights of logistic regression learnt with the UCI Adult Income dataset, explored by the re-training and dropout strategies. Since the model weights (features) is high-dimensional, we apply t-SNE visualization [Van der Maaten & Hinton, 2008] on the model weights for the sake of illustration. As observed, the model weights obtained by he re-training and dropout strategies largely overlapped with each other, indicating that they have similar performance in terms of exploring the Rashomon set when the hypothesis space is small. Note that there is no closed-form characterization of the Rashomon set for logistic regression.

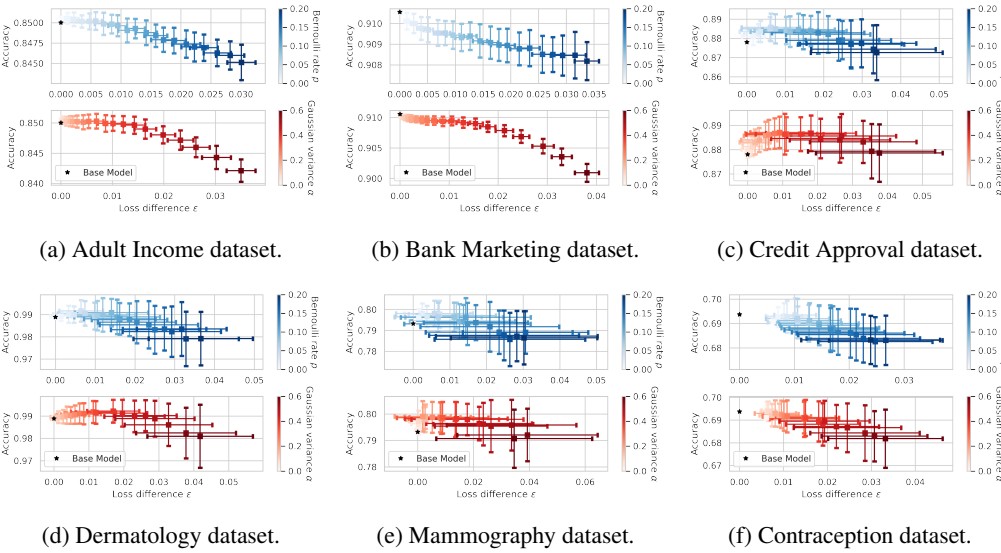

(a) Adult Income dataset.  (b) Bank Marketing dataset.  (c) Credit Approval dataset.

(d) Dermatology dataset.  (e) Mammography dataset.  (f) Contraception dataset.

Figure E.5: Loss and accuracy deviations versus Bernoulli and Gaussian dropouts on the UCI datasets.

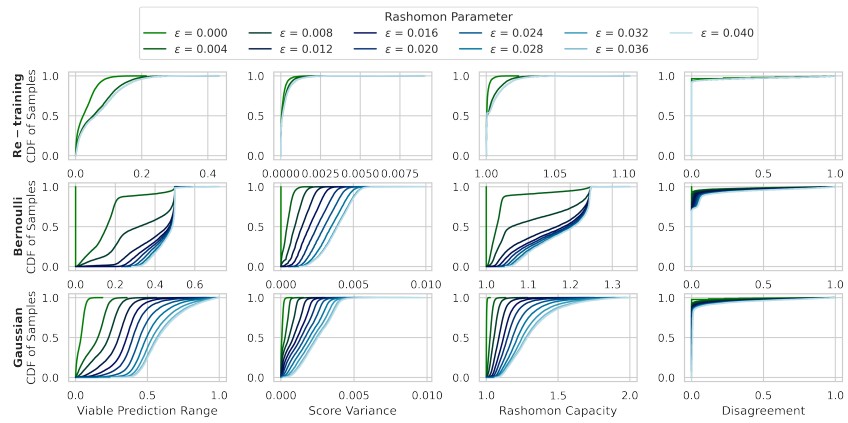

Figure E.6: Cumulative distributions of the predictive multiplicity metrics that are defined across all samples in the UCI Adult Income dataset.

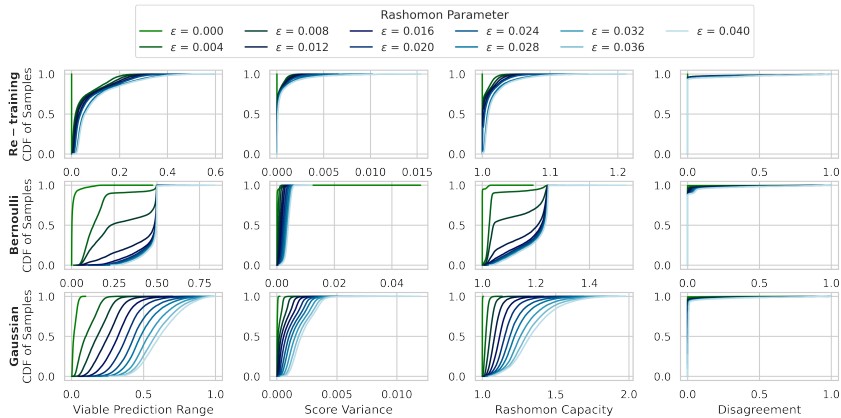

Figure E.7: Cumulative distributions of the predictive multiplicity metrics that are defined across all samples in the UCI Bank Marketing dataset.

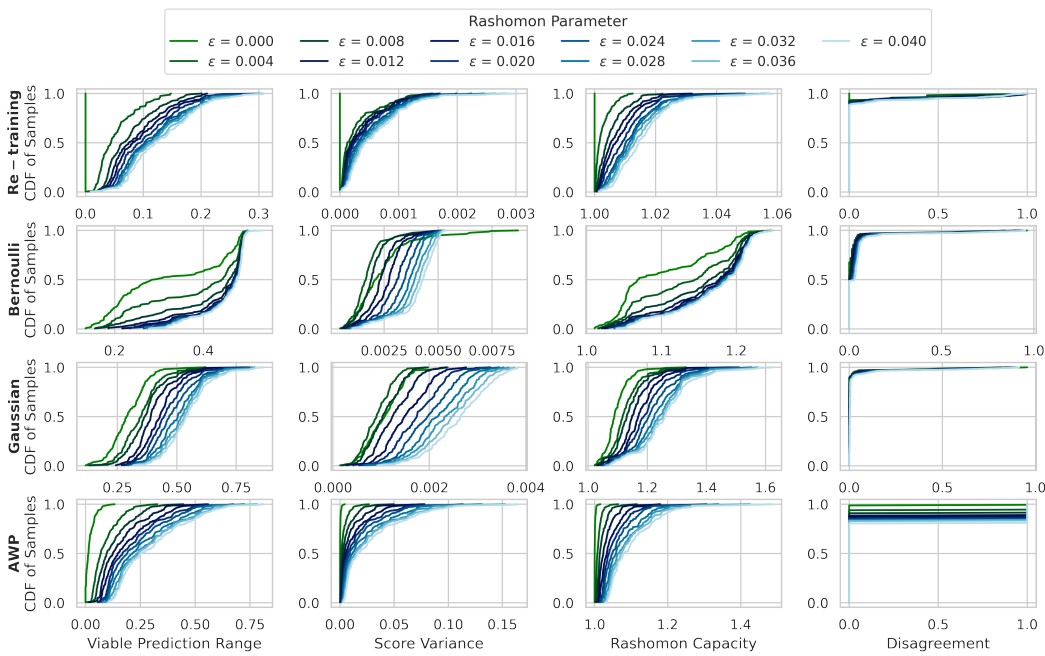

Figure E.8: Cumulative distributions of the predictive multiplicity metrics that are defined across all samples in the UCI Credit approval dataset.

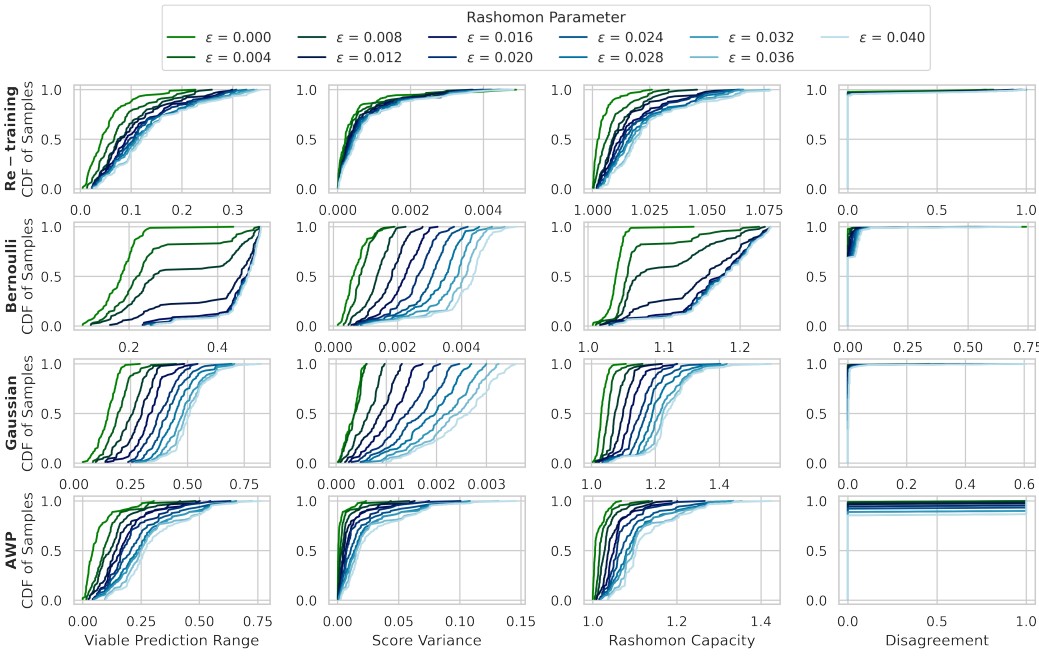

Figure E.9: Cumulative distributions of the predictive multiplicity metrics that are defined across all samples in the UCI Dermatology dataset.

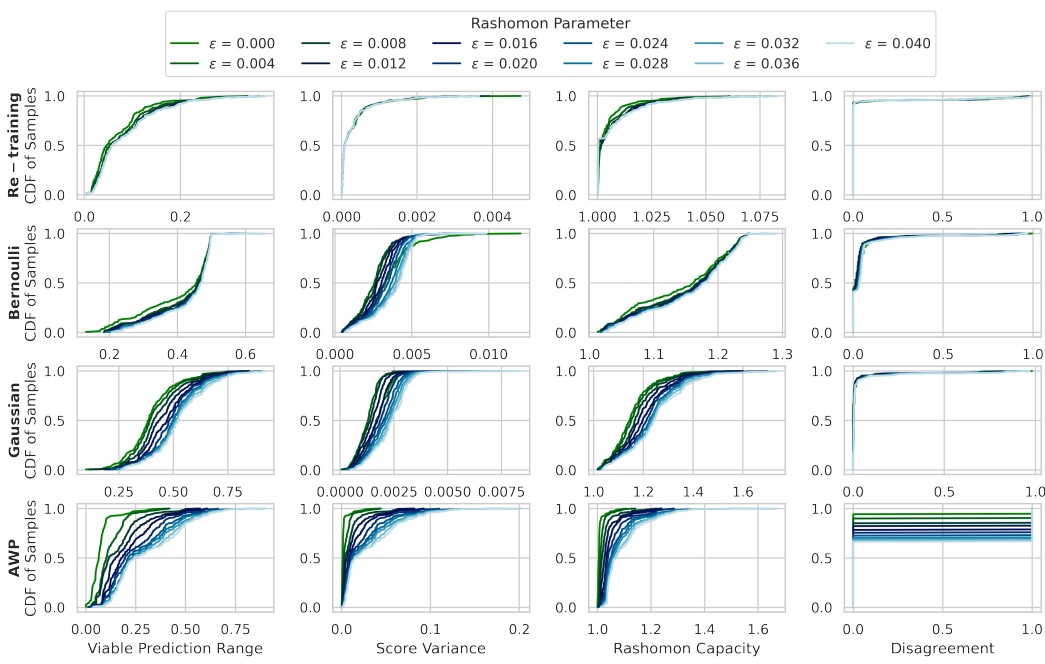

Figure E.10: Cumulative distributions of the predictive multiplicity metrics that are defined across all samples in the UCI Mammography dataset.

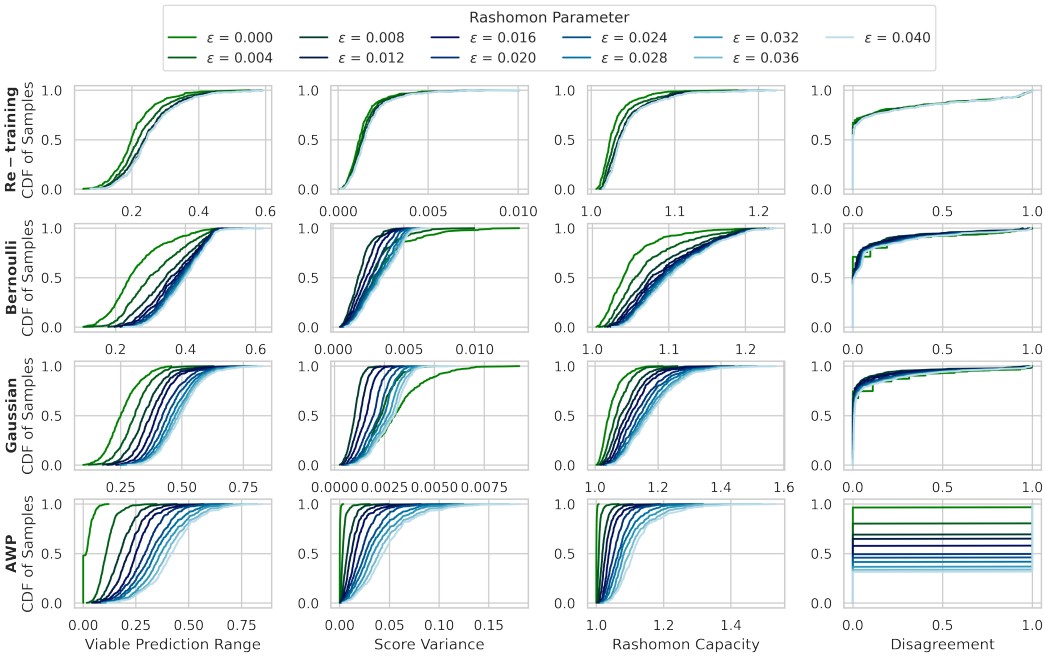

Figure E.11: Cumulative distributions of the predictive multiplicity metrics that are defined across all samples in the UCI Contraception dataset.

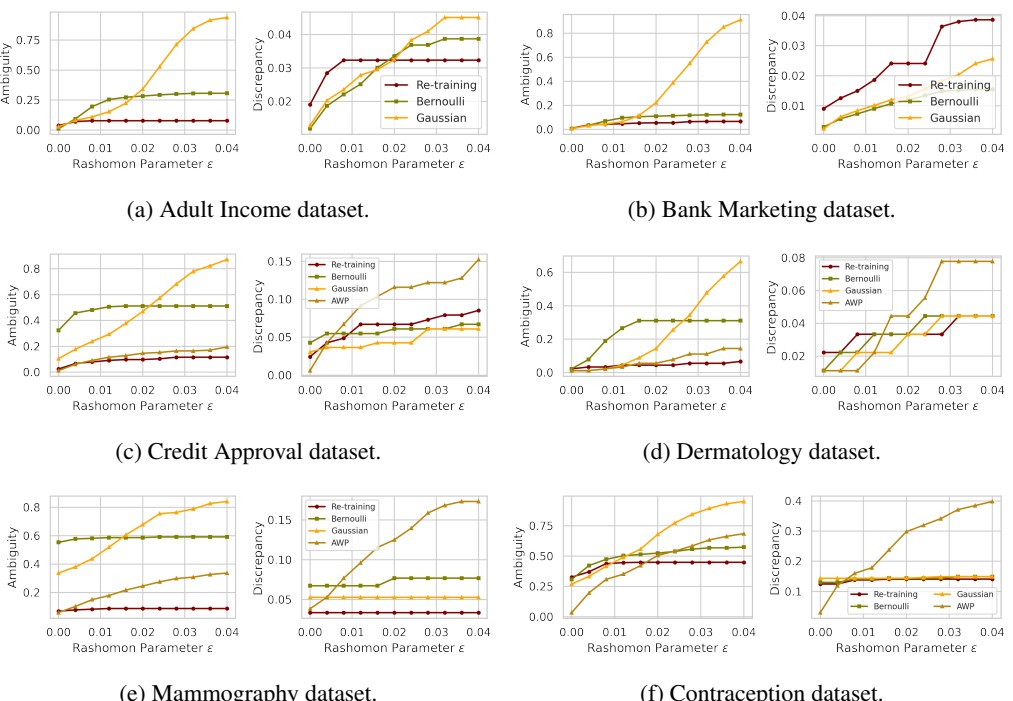

(a) Adult Income dataset.

(b) Bank Marketing dataset.

(c) Credit Approval dataset.

(d) Dermatology dataset.

(e) Mammography dataset.

(f) Contraception dataset.

Figure E.12: Ambiguity and discrepancy of the UCI datasets.

Table E.6: Raw runtime values on the UCI datasets.

| Dataset | Runtime (seconds per model) | | | |
|---|---|---|---|---|
| | Re-training | AWP | Bernoulli Dropout | Gaussian Dropout |
| Adult Income | $19.35 \pm 0.68$ | — | $0.0614 \pm 0.24$ | $0.0633 \pm 0.24$ |
| Bank Deposit | $7.78 \pm 2.94$ | — | $0.2243 \pm 0.42$ | $0.2200 \pm 0.41$ |
| Credit Approval | $0.51 \pm 0.56$ | $10.18 \pm 2.29$ | $0.0019 \pm 0.04$ | $0.0019 \pm 0.04$ |
| Dermatology | $0.11 \pm 0.34$ | $4.27 \pm 0.96$ | $0.0038 \pm 0.06$ | $0.0038 \pm 0.06$ |
| Mammography | $0.31 \pm 0.52$ | $10.09 \pm 1.62$ | $0.0024 \pm 0.05$ | $0.0024 \pm 0.05$ |
| Contraception | $0.56 \pm 0.67$ | $21.09 \pm 3.78$ | $0.0071 \pm 0.08$ | $0.0071 \pm 0.08$ |

Table E.7: The efficiency of obtaining models from the Rashomon set with different Rashomon parameters $\epsilon$ on the UCI datasets. Each number is the percentage of the obtained models that are in the Rashomon set by different methods. For most datasets and $\epsilon$, dropout has higher percentage compared to re-training.

| Dataset | Rashomon Parameter $\epsilon$ | Methods | | |
|---|---|---|---|---|
| | | Re-training | Bernoulli Dropout | Gaussian Dropout |
| Adult Income | 0.001 | 29.50% | 0.00% | 75.00% |
| | 0.002 | 55.25% | 50.00% | 82.67% |
| | 0.003 | 71.00% | 66.67% | 85.00% |
| | 0.004 | 81.25% | 75.00% | 87.38% |
| Bank Marketing | 0.001 | 2.89% | 0.00% | 75.00% |
| | 0.002 | 9.89% | 50.00% | 83.17% |
| | 0.003 | 18.44% | 66.67% | 85.43% |
| | 0.004 | 24.67% | 75.00% | 87.50% |
| Credit Approval | 0.001 | 0.33% | 0.00% | 63.00% |
| | 0.002 | 0.67% | 50.00% | 71.25% |
| | 0.003 | 2.00% | 50.00% | 71.00% |
| | 0.004 | 2.67% | 59.00% | 72.00% |
| Dermatology | 0.001 | 0.78% | 0.00% | 66.75% |
| | 0.002 | 1.33% | 50.00% | 74.80% |
| | 0.003 | 1.89% | 50.00% | 78.50% |
| | 0.004 | 2.11% | 61.33% | 79.14% |
| Mammography | 0.001 | 20.78% | 0.00% | 57.50% |
| | 0.002 | 25.22% | 50.00% | 59.50% |
| | 0.003 | 30.78% | 52.33% | 65.38% |
| | 0.004 | 38.11% | 57.75% | 69.38% |
| Contraception | 0.007 | 92.00% | 0.00% | 66.00% |
| | 0.008 | 95.56% | 59.00% | 69.43% |
| | 0.010 | 97.89% | 65.50% | 73.75% |
| | 0.011 | 98.89% | 64.43% | 75.40% |

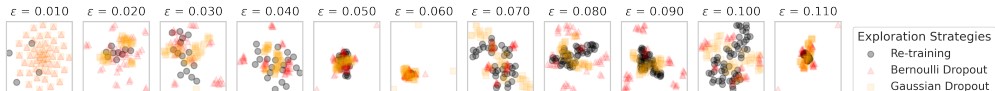

Figure E.13: Comparisons of the model weights of logistic regression obtained from the re-training and dropout strategies on UCI Adult Income dataset with t-SNE visualization [Van der Maaten & Hinton, 2008].

## E.2 ADDITIONAL RESULTS ON THE MS COCO DATASET

In Figure E.14 and Figure E.16a, we report the estimation of all predictive multiplicity metrics that corresponds to the example shown in Figure 3 for the MS COCO dataset. There are a significant portion of samples ($30\% \sim 50\%$) that disagree with each other on the human bounding boxes found by the Yolov3 detector. Due to extremely high time complexity, we do not include the re-training strategy and the adversarial weight perturbation algorithm.

In Figure E.15 and Figure E.16b, we implement another detector Mask R-CNN [He et al., 2017], which is a state-of-the-art model for instance segmentation, developed on top of a region-based convolutional neural networks called Faster R-CNN. The Mask R-CNN detector makes a larger portion of the samples have higher predictive multiplicity, in terms of viable prediction range and the Rashomon Capacity, compared against the Yolov3 detector.

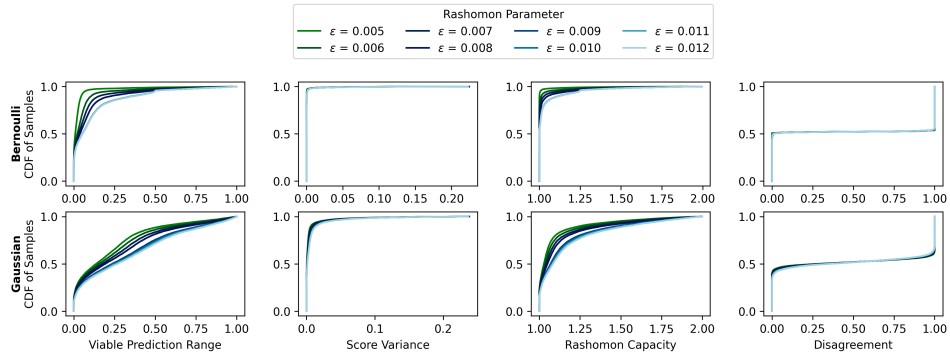

Figure E.14: Cumulative distributions of the predictive multiplicity metrics that are defined across all samples in the MS COCO dataset with the YoloV3 detector.

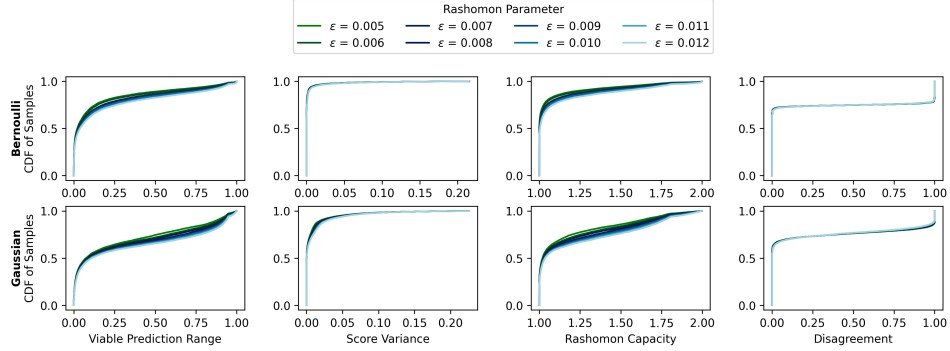

Figure E.15: Cumulative distributions of the predictive multiplicity metrics that are defined across all samples in the MS COCO dataset with the Mask R-CNN detector.

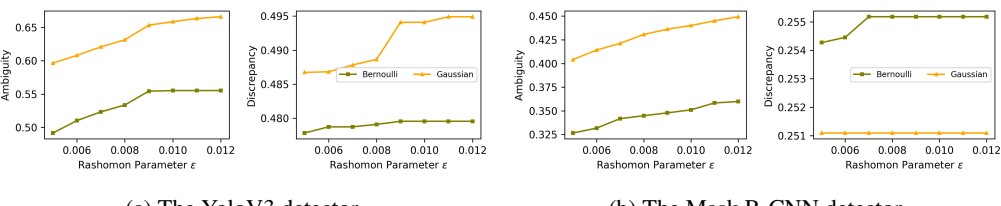

(a) The YoloV3 detector.         (b) The Mask R-CNN detector.

Figure E.16: Ambiguity and discrepancy of the MS COCO dataset with the YoloV3 and Mask R-CNN detectors.

### E.3 Additional experiments on CIFAR-10/-100 datasets

We measure the predictive multiplicity metrics on more challenging image datasets CIFAR-10/-100 [Krizhevsky & Hinton, 2009] with convolutional neural network architectures including VGG16 [Simonyan & Zisserman, 2014] and ResNet50 [He et al., 2016]. CIFAR-10 consists of 60,000 $32 \times 32$ colored tiny images divided into ten classes, with each class representing a distinct object category such as airplanes, dogs, and cars. CIFAR-100 offers a more challenging dataset containing 100 classes with 600 images each, providing a broader range of object categories, including fine-grained distinctions like various types of birds and flowers. We train CIFAR-10 with VGG16 for 7 epochs with learning rate 0.001, and CIFAR-100 with ResNet50 for 40 epochs with learning rate 0.001—the performance of the bases models is summarized in Table E.8. We obtain 50 Bernoulli and Gaussian dropout models for each 5 values $p$ and $\alpha$ spread evenly in $[0, 0.008]$ and $[0, 0.1]$ respectively for CIFAR-10, and similarly for CIFAR-100 where $p$ and $\alpha$ spread evenly in $[0, 0.002]$ and $[0, 0.05]$. For the re-training results, we re-train 20 times for each epoch 4,5,6,7,8,9 for CIFAR-10, and 20 times for each epoch 20,25,30,35 for CIFAR-100. The estimates of predictive multiplicity metrics are summarized in Figure E.18, and runtime/speedup in Table E.9. Since VGG16 and ResNet50 are large-scale models that contain many local minima, re-training can easily find different local minima that have similar performance. On the other hand, the dropout methods can only search models at the neighborhood of the given pre-trained model. Therefore, re-training outperforms dropout in terms of exploring diverse models in the Rashomon set, at the expense of a much larger runtime. Finally, Figure E.19 and Figure E.20 summarize the cumulative distributions of the predictive multiplicity metrics that are defined across all samples for CIFAR-10 and CIFAR-100 respectively, and Figure E.21 shows the ambiguity and discrepancy.

Table E.8: CIFAR-10/-100 dataset base model performance.

| Dataset | Architecture | Training | | Test | |
| --- | --- | --- | --- | --- | --- |
| | | Loss | Accuracy | Loss | Accuracy |
| CIFAR-10 | VGG-16 | 0.2692 | 90.85% | 0.5743 | 81.63% |
| CIFAR-100 | ResNet-50 | 0.0014 | 99.96% | 2.1591 | 59.53% |

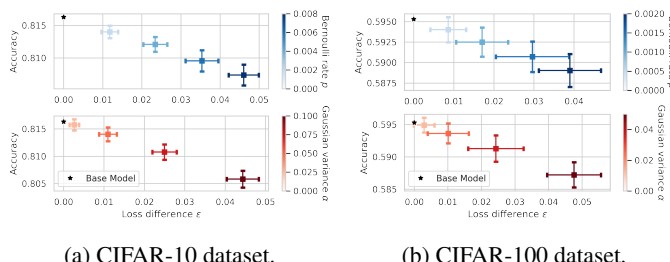

(a) CIFAR-10 dataset.  (b) CIFAR-100 dataset.

Figure E.17: Loss and accuracy deviations versus Bernoulli and Gaussian dropouts on CIFAR-10/-100 datasets.

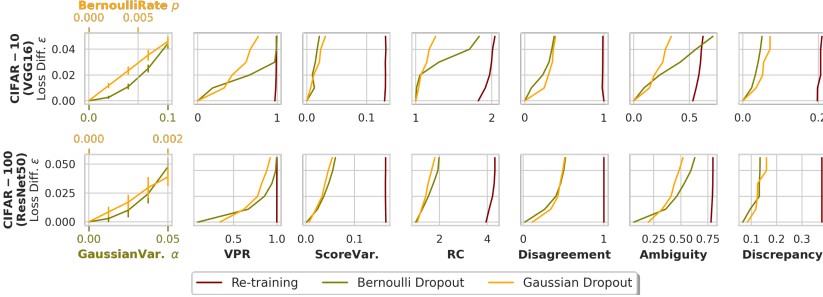

Figure E.18: Loss vs. dropout parameters and the corresponding predictive multiplicity metrics of the baselines with CIFAR-10/-100 datasets.

Table E.9: Raw runtime values and speedup on CIFAR-10/-100 datasets. All runtimes are evaluated with the same computational platform.

| Dataset | Runtime (seconds per model) | | | Speedup |
|---|---|---|---|---|
| | Re-training | Bernoulli Dropout | Gaussian Dropout | Gaussian Dropout over Re-training |
| CIFAR-10 (VGG16) | $485.10 \pm 10.29$ | $1.8720 \pm 0.33$ | $4.1040 \pm 0.93$ | $118.20\times$ |
| CIFAR-100 (ResNet50) | $3985.45 \pm 1626.27$ | $1.0880 \pm 0.28$ | $1.3200 \pm 0.47$ | $3019.28\times$ |

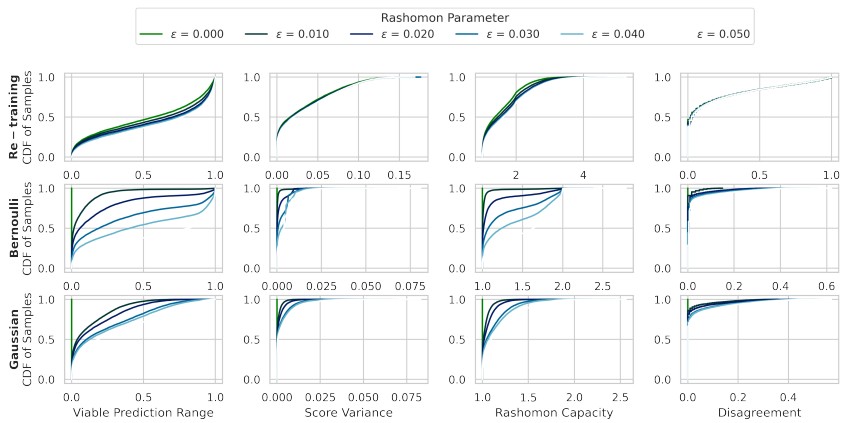

Figure E.19: Cumulative distributions of the predictive multiplicity metrics that are defined across all samples in CIFAR-10 dataset.

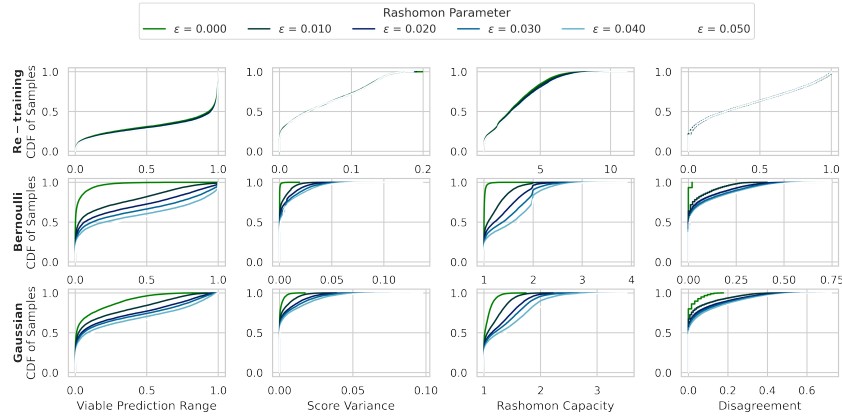

Figure E.20: Cumulative distributions of the predictive multiplicity metrics that are defined across all samples in CIFAR-100 dataset.

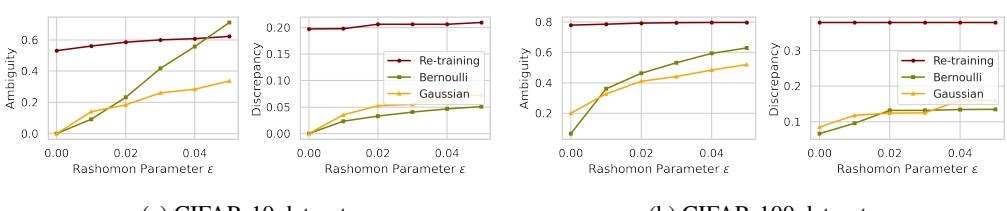

(a) CIFAR-10 dataset.

(b) CIFAR-100 dataset.

Figure E.21: Ambuguity and dsicrepancy of CIFAR-10/-100 datasets.

Finally, we implement the re-training + dropout strategy, as mentioned in Section 6, to better explore the Rashomon set. In Figure E.22 and E.23, the predictive multiplicity metrics presented in each row are all estimated from 100 models in the Rashomon set. The first row is 100 dropout models obtained from 1 model via re-training. The second row is 50 dropout models obtained separately from 2 different model via re-training (again 100 models in total). The third row is 25 dropout models obtained separately from 4 different model via re-training, and the fourth is 20 dropout models obtained separately from 5 different model via re-training. Observing multiplicity metrics such as the viable prediction range, the score variance and the Rashomon Capacity, it is clear that with more diverse model via re-training, the estimates of the multiplicity metrics have a higher cumulant. Take the viable prediction range as an example, around 20% of the samples has up to value 0.5 if we apply dropout on only one re-training model (first row). However, there are 50% of the samples has up to value 0.5 if we apply dropout on 5 diverse re-training models (last row).

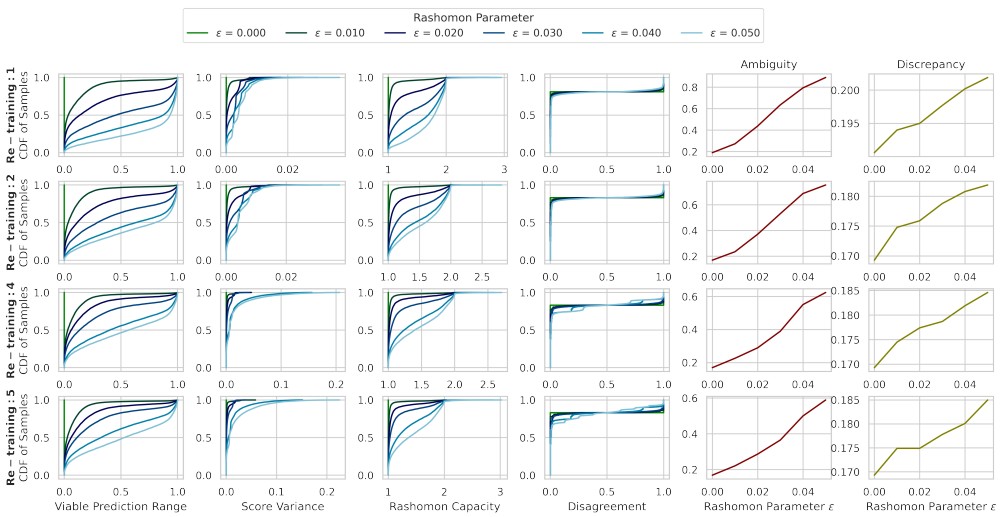

Figure E.22: Re-training + Bernoulli dropout to explore the Rashomon set on the CIFAR-10 dataset with VGG-16.

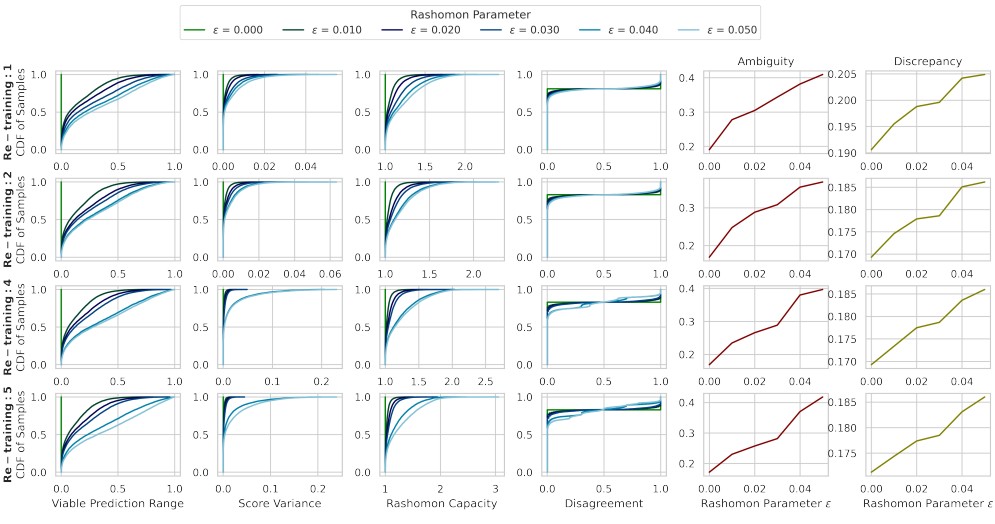

Figure E.23: Re-training + Gaussian dropout to explore the Rashomon set on the CIFAR-10 dataset with VGG-16..

### E.4 ABLATION STUDY

We use the UCI Adult Income dataset to perform ablation studies on the width and depth of the feed-forward neural networks, and use CIFAR-10/-100 for different neural network architectures.

We train the UCI Adult Income dataset on a neural network with different numbers of layers $K \in \{1, 2, 3, 4, 5\}$, and the number of neurons in each layers is 200. Figure E.24 shows the loss and accuracy deviations caused by Bernoulli and Gaussian dropouts. As the number of layers increases, the loss deviation also increases, which is consistent with the theoretical bound derived in (9). On the other hand, Figure E.25 shows that for a neural network with only one hidden layer and different number of neurons $\{200, 400, 600, 800, 1000\}$, it does not have a significant effect on the loss deviation under the same dropout parameter. The reason is in (9), the effect of number of layers $K$ is polynomial while the effect of the number of neurons $m$ is linear.

Finally, we discuss the Rashomon effect with different neural network architectures on CIFAR-10/-100 datasets. Note that VGG-16, ResNet-18 and ResNet-50 have 138M, 11M and 25.6M parameters. We compare the loss deviation using Figure E.17 for VGG-16 and Figure E.26 for ResNet-18/-50. For CIFAR-10, Bernoulli dropout on different architectures has similar performance, while Gaussian dropout leads to the largest loss deviation on ResNet-18. It is because ResNet-18 has the least number of parameters. Moreover, ResNet-18 and ResNet-50 have similar loss deviations on CIFAR-100.

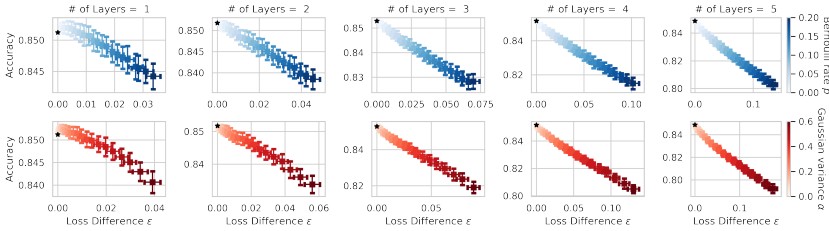

Figure E.24: Loss and accuracy deviations versus Bernoulli and Gaussian dropouts on on Adult Income dataset with different numbers of layers.

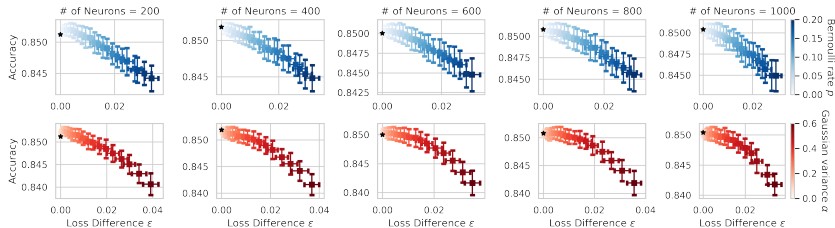

Figure E.25: Loss and accuracy deviations versus Bernoulli and Gaussian dropouts on on Adult Income dataset with different numbers of neurons.

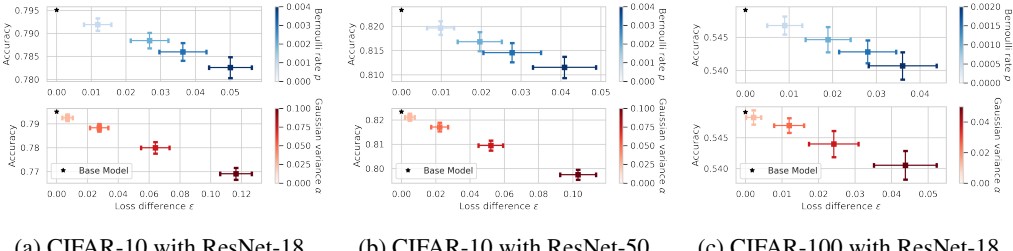

(a) CIFAR-10 with ResNet-18.    (b) CIFAR-10 with ResNet-50.    (c) CIFAR-100 with ResNet-18.

Figure E.26: Loss and accuracy deviations versus Bernoulli and Gaussian dropouts on CIFAR-10/-100 datasets with different neural network architectures.

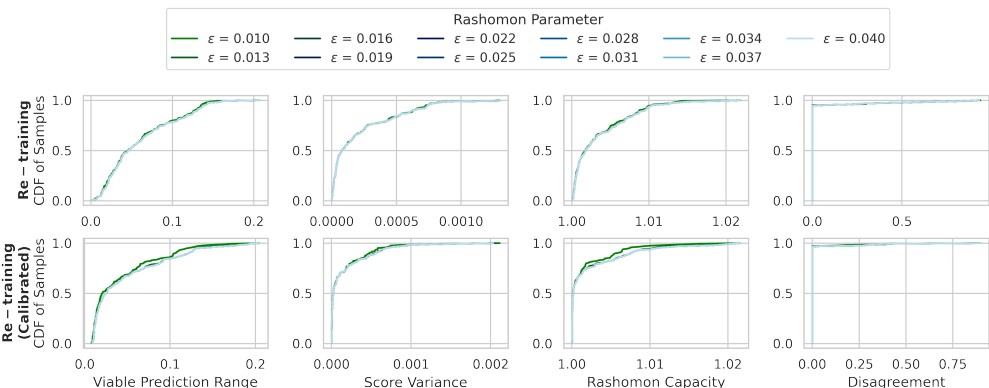

Figure E.27: Score-based predictive multiplicity metrics of the Credit Approval dataset with and without model calibration. Calibration, as an score manipulation on average performance, will not affect predictive multiplicity.

Moreover, we discuss whether model calibration could either reduce or exacerbate predictive multiplicity. If a perfectly calibrated classifier assigns a 50% score to a sample (e.g., in binary classification), it does not necessarily mean that this sample has high multiplicity. A perfectly calibrated classifier is one whose predicted classes matches the true classes on average across samples (e.g., samples predicted to be 50% of one class have true outcomes matching that class 50% of the time). However, this does not necessarily translate to a (in)consistent set of predictions for a single target sample across equally calibrated classifiers. It may be the case that all calibrated models drawn from the Rashomon set assign the same 50% probability for that sample (no multiplicity). Conversely, some models may assign higher and lower confidence for that sample (high multiplicity) yet, on average, still be well-calibrated. Again, this happens because calibration (like accuracy and loss) is an average metric across all samples.

We implement Platt scaling [Platt et al., 1999] to calibrate the scores produced by the neural networks for the Credit Approval dataset, and compare the score-based predictive multiplicity metrics with and without calibration in Figure E.27. We observe that the score-based predictive multiplicity metrics are not impacted with calibration, for the exact reasons we have provided above.

