# OpenReview forum: "Dropout-Based Rashomon Set Exploration for Efficient Predictive Multiplicity Estimation"
_ICLR.cc/2024/Conference — ICLR 2024 poster_

### Official Review · Reviewer_BakJ · 2023-10-26

**Soundness:** 3 good
**Presentation:** 4 excellent
**Contribution:** 3 good
**Rating:** 6
**Confidence:** 4

**Summary:**

This paper’s main goal is to explore Rashomon sets of feed-forward neural networks with the help of dropout. Both Gaussian and Bernoulli dropouts are considered, the former involving the addition of noise to the weights of the networks. The approach is used to estimate the Rashomon set, and proof of the consistency of the approach is provided. Empirical results show the effectiveness of the approach on various datasets.

**Strengths:**

The idea is intuitive. It leads to impressive computation time saving compared to other approaches from the literature. The article is well-written and clear. The experiments are directly in line with the motivations of the work (ethical concerns).

**Weaknesses:**

**Major**

1.1 – The biggest weakness of the approach concerns its limitations. As honestly discussed by the authors in Section 6 – Limitations, the fact that when the hypothesis space to explore is huge (that is when the predictor has many parameters to tune), the exploration that is done by the dropout approach is fairly limited. This is clearly related to Proposition 4, where an important value of $M$ is necessary for layers having hundreds of neurons and an important value of $k$ is necessary with complex neural networks.

1.2 – Two scenarios could occur: the first one is that the hypothesis space to explore is relatively small. That is, it is fairly explored by the dropout method. But, even though a 30x to 5000x speedup over other approaches is seen, it is never defended that those other approaches are not scalable with small hypothesis space. Even though there is a huge speedup gain, if the other approaches are relatively fast (do not take hours to compute), then why favourising dropout? The second scenario is that the hypothesis space to explore is large. The time gain is then undermined by the limitation in the exploration. Plus, when it comes to large models, it is common to retrain only the classification head of the predictor, or to fix many layers; doing so could really fasten the retraining scheme, thus undermining the potential advantage of the dropout approach.

2 – It seems to me that both the depiction in Figure 2 and the speedup reported in Table 1 are lacking important details. For example: How many different models are sought? How many reruns were done for retraining VS how many dropouts were computed? What was the total time for each individual method? To my understanding, many reruns are already needed in the first place, no matter the approach, in order to ensure that the reference model is an « empirical minimizer »; was that taken into account when comparing the time for building the empirical Rashomon sets in Table 1? What was the size of the predictor used on these different UCI tasks (this kind of information is necessary in the main article, not the supplementary material)?

3 – I feel like something is conceptually wrong with the comparison between retraining and the current dropout scheme. Retraining makes it such that the validation loss is the highest possible. Therefore, it makes sense that many runs are needed in order to find models close to the « empirical minimizer». With the dropout scheme, an empirical minimizer is found, and then dropout is applied while making sure the training loss does not diminish too much. The two approaches do not have the same objectives.

4.1 - The dropout leads to a scheme where each new model depends on the initial model. All of the models are thus dependent. Therefore, the estimation of the Rashomon metrics is biased. And while « not all estimators of predictive multiplicity metrics carry a theoretical analysis of its statistical properties such as consistency and sample complex », I feel like it is a property of interest. Indeed, one of the motivations of the work is the need for ethics and, more specifically, fairness. I see the goal in exploring the Rashomon sets to find many predictors giving different predictions to people for different reasons. But having all of the models interconnected makes it such that the reasons for the predictions are all linked and just a few are explored with the dropout scheme.

4.2 – Proposition 5 aims at proving that the approach is not biased, but relies on the assumption that «  the models around W∗ are uniformly distributed in a d-dimensional ball with center $\mathbf{W}^*$ and radius $\delta$, i.e., $B(\mathbf{W}^*, \delta)$. Accordingly, we may assume that the population means $\mu$ for a sample can be expressed as [...] ». The method explicitly does that (especially the Gaussian dropout), exploring around the « population mean », that is, the empirical minimizer. Therefore, assuming the uniform distribution of the Rashomon set around a center trivially leads to the unbiasedness of the dropout scheme, but is unreasonable.

**Minor**

1 – Typo: « Moreover, as lone as »

**Questions:**

1 – It is said that « not all estimators of predictive multiplicity metrics carry a theoretical analysis of its statistical properties such as consistency and sample complex » Could you provide some citation supporting this claim?

2 – What justifies fixing a single Bernoulli or a Gaussian dropout parameter for all layers simultaneously? Shouldn’t the layers be treated independently?

3 – Concerning the quantification of predictive multiplicity, it is said that « [f]or example, Long et al. [2023], Cooper et al. [2023] and Watson-Daniels et al. [2023] quantify predictive multiplicity by the standard deviation, variance and the largest possible difference of the scores (termed viable prediction range (VPR) therein) respectively » So, what definition between those three is retained in the article?

**Details Of Ethics Concerns:**

This might not be of an "ethical" concern, but the 9-page limit is exceeded by  1/4th ~ 1/3rd of a page.

---

> ### Author Response · Authors · 2023-11-16
> **Response to Reviewer BakJ**
>
> 1. **The biggest weakness of the approach concerns its limitations. As honestly discussed by the authors in Section 6 – Limitations, the fact that when the hypothesis space to explore is huge (that is when the predictor has many parameters to tune), the exploration that is done by the dropout approach is fairly limited. This is clearly related to Proposition 4, where an important value of M is necessary for layers having hundreds of neurons and an important value of k is necessary with complex neural networks.**
>
>     Indeed, when the hypothesis space is huge, exploring models in the Rashomon set, i.e., obtaining almost-equally-optimal models from the hypothesis space, is in general computationally infeasible. Despite dropout-based exploration is limited, it makes the exploration possible compared to other strategies such as re-training and AWP, which are much harder computational tasks as they requires repeated re-training and perturbation. We thank the reviewer for bringing this up, and will add a more thorough comparisons of the pros and cons among the three strategies, re-training, AWP, and the proposed one in Section 3 and 4.
>
> 2. **Two scenarios could occur: the first one is that the hypothesis space to explore is relatively small. That is, it is fairly explored by the dropout method. But, even though a 30x to 5000x speedup over other approaches is seen, it is never defended that those other approaches are not scalable with small hypothesis space. Even though there is a huge speedup gain, if the other approaches are relatively fast (do not take hours to compute), then why favourising dropout? The second scenario is that the hypothesis space to explore is large. The time gain is then undermined by the limitation in the exploration. Plus, when it comes to large models, it is common to retrain only the classification head of the predictor, or to fix many layers; doing so could really fasten the retraining scheme, thus undermining the potential advantage of the dropout approach.**
>
>     We thank the reviewer for bringing up this practical concern. When the hypothesis space is relatively small, the re-training and AWP strategies could be fast; however, the dropout method could still be faster (despite that the speedup could be smaller) than re-training and AWP for the same reason—it does not require gradient computation and weight updating. Moreover, whether the proposed dropout method could explore the Rashomon set as effective as re-training or AWP when the hypothesis space is mall, is an interesting theoretical question and also requires numerical evidence.
>
>     The second scenario is quite a common practice when fine-tuning a pre-trained model. Indeed, re-training and AWP could be significantly faster when only applied on part of the weights (e.g., the classification head). In this sense, the dropout method could also be applied on part of the weights in order to accelerate. The comparison of runtime performance among re-training, AWP, and the dropout method when applied on partial weights (also briefly discussed in Section 6 Future Directions.) is an interesting and practical research direction. Due to needlessness of updating weights, the proposed dropout strategy could still be more computationally efficient with the speedup varying depending on different complexity of the hypothesis space.
>     Finally, as we have discussed in Section 6 Future Direction, the proposed dropout method could also be applied as a compliment to re-training, in order to leverage between efficiency and effectiveness of Rashomon set exploration. We will address this concern by adding a paragraph in Section 4 or 5 to further discuss the impact of the cardinality of the hypothesis space with some numerical examples.
>
> **Continued...**

---

> > ### Author Response · Authors · 2023-11-16
> > **Response to Reviewer BakJ - 2**
> >
> > 3. **It seems to me that both the depiction in Figure 2 and the speedup reported in Table 1 are lacking important details. For example: How many different models are sought? How many reruns were done for retraining VS how many dropouts were computed? What was the total time for each individual method? To my understanding, many reruns are already needed in the first place, no matter the approach, in order to ensure that the reference model is an empirical minimizer; was that taken into account when comparing the time for building the empirical Rashomon sets in Table 1? What was the size of the predictor used on these different UCI tasks (this kind of information is necessary in the main article, not the supplementary material)?**
> >
> >     We appreciate the reviewer for pointing it out, all this information about implementation details are included in Section D and E in the Appendix, and are pointed to the readers in Section 4 and 5. We will move these information from the Appendix to a paragraph in the main text, and re-emphasize the pointers to the readers in the main text. In Figure 2, we first train an empirical minimizer $\mathbf{w}^\*$, and note that the runtime for obtaining this $\mathbf{w}^\*$ is not counted in Table 1. For each loss diff. $\epsilon$, we obtain 100 models for both re-training and the dropout method. For the re-training approach, we train a lot of models with different weight initializations, and collect the models that has test loss deviation against the empirical minimizer $\mathbf{w}^\*$ smaller than $\epsilon$. Similar the dropout approach we also collect 100 models. The predictors all have the same architecture, a neural network with a single hidden layer composed of 1k neurons.
> >
> >     We would like to emphasize that the empirical minimizer $\mathbf{w}^\*$ are given to all three strategies, therefore the time to rerun to obtain this empirical minimizer $\mathbf{w}^\*$ is not accounted in the speedup in Table 1. Moreover, the speedup in Table 1 is calculated per model, not the total time for each individual methods. We will improve this point by clarifying the speedup is compared per model in the caption.
> >
> > 4. **I feel like something is conceptually wrong with the comparison between retraining and the current dropout scheme. Retraining makes it such that the validation loss is the highest possible. Therefore, it makes sense that many runs are needed in order to find models close to the "empirical minimizer". With the dropout scheme, an empirical minimizer is found, and then dropout is applied while making sure the training loss does not diminish too much. The two approaches do not have the same objectives.**
> >
> >     We would like to follow our answers to the previous question and clarify that an empirical minimizer $\mathbf{w}^\*$ is given to all the three strategies, re-training, AWP and dropout to explore the Rashomon set defined according to $\mathbf{w}^\*$. Indeed re-training and dropout are two very different strategies, with the same goal of exploring the Rashomon set. Re-training has been the most common method in existing literature such as [Long et al., 2023] and [Kulynych et al., 2023]. However, it also limits the scale of the experiments that could be presented in those papers due to time complexity.
> >
> >     Here we provide an additional numerical results on why re-training is time consuming. In Table E.7 in the Appendix, we obtain 500 models with re-training and the dropout methods, and report the percentage of the models from each strategies that are in the Rashomon set with a given empirical minimizer and varying $\epsilon$. It is shown that for most cases, the percentage of models in the Rashomon set obtained by re-training is significantly smaller than that by dropout. The reason is that it is very hard to control the loss fluctuations from randomized algorithms such as stochastic gradient descent. We will add a paragraph in Section 4 to summarize the discussions above in the final revision.
> >
> > **Continued...**

---

> > > ### Author Response · Authors · 2023-11-16
> > > **Response to Reviewer BakJ - 3**
> > >
> > > 5. **The dropout leads to a scheme where each new model depends on the initial model. All of the models are thus dependent. Therefore, the estimation of the Rashomon metrics is biased. And while "not all estimators of predictive multiplicity metrics carry a theoretical analysis of its statistical properties such as consistency and sample complex", I feel like it is a property of interest. Indeed, one of the motivations of the work is the need for ethics and, more specifically, fairness. I see the goal in exploring the Rashomon sets to find many predictors giving different predictions to people for different reasons. But having all of the models interconnected makes it such that the reasons for the predictions are all linked and just a few are explored with the dropout scheme.**
> > >
> > >     We agree with the reviewer on the limitation of the dropout methods, and the need to study theoretical analysis of predictive multiplicity metric estimation for more general scenarios. We would like to emphasize that the goal of this paper is not using the dropout method to find models in the Rashomon set for other ethical needs, but provides an alternative method to estimate predictive multiplicity metrics more efficiently. How to explore in the Rashomon set to find models that can comply with other ethical needs (using the dropout method proposed here or not) is an interesting and open future direction.
> > >
> > >     Moreover, since the entire (true) Rashomon set is not computationally feasible, all estimations based on exploring the Rashomon set (re-training or the proposed dropout method) lead to under-estimation of multiplicity metrics. The best practice we could study in the future is to improve the under-estimation. Finally, despite that the proposed dropout method may only explore a small subset in the Rashomon set, it give an lower bound of the multiplicity metrics in a much more efficient manner.
> > >
> > > 6. **Proposition 5 aims at proving that the approach is not biased, but relies on the assumption that "the models around $W\*$ are uniformly distributed in a d-dimensional ball with center $W\*$ and radius $\delta$, i.e., $B(W\*, \delta)$. Accordingly, we may assume that the population means $\mu$ for a sample can be expressed as [...]". The method explicitly does that (especially the Gaussian dropout), exploring around the "population mean", that is, the empirical minimizer. Therefore, assuming the uniform distribution of the Rashomon set around a center trivially leads to the unbiasedness of the dropout scheme, but is unreasonable.**
> > >
> > >     We thank the reviewer for reading Proposition 5 in the Appendix. Indeed, the assumption of the uniform distribution around $W\*$ may not reflect the true underlying distributions of models in the Rashomon set (e.g., when there are exponentially many local minima), but it is an ideal assumption that facilitates mathematical analysis that was also adopted in existing literature, e.g., [Kulynych et al., 2023]. How to characterize the distribution of models in the Rashomon set still remains an open and active challenge in the field.
> > >
> > >     Additionally, please note that Gaussian distribution in a high dimensional space does not look like a bell (in our vanilla understanding, concentrated in the middle), but looks like a bubble where the mass is concentrated around $\sqrt{d}$, where $d$ is the dimension. This property contradicting to our intuition is stated in the Gaussian annulus theorem (Lemma A.3 in the Appendix). Therefore, we would like to argue that the derivation of Proposition 5 to bound the gap is not trivial. We will include a more detailed discussion on the uniform assumption in the Appendix, and add pointers in the main text.
> > >
> > > 7. **It is said that "not all estimators of predictive multiplicity metrics carry a theoretical analysis of its statistical properties such as consistency and sample complex" Could you provide some citation supporting this claim?**
> > >
> > >     We thank the reviewer for pointing this out and will add supporting discussions accordingly in the beginning of Section 4. There are some theoretical analyses of estimating predictive multiplicity metrics. For example, the exact estimation of ambiguity and discrepancy for linear classifiers are studied in [Marx et al., 2020]. Similarly, the theoretical analysis of estimating VPR with logistic regression is also provided in [Watson-Daniels et al., 2023]. Moreover, the estimation of disagreement is also discussed in [Kulynych et al., 2023]. However, the statistical properties of estimating other metrics such as the Rashomon Capacity, or with a general hypothesis space, still remain an open challenge.
> > >
> > > **Continued...**

---

> > > > ### Author Response · Authors · 2023-11-16
> > > > **Response to Reviewer BakJ - 4**
> > > >
> > > > 8. **What justifies fixing a single Bernoulli or a Gaussian dropout parameter for all layers simultaneously? Shouldn’t the layers be treated independently?**
> > > >
> > > >     We thank the reviewer for raising this interesting direction. As an early study and for theoretical convenience, in this paper we fix the dropout parameters for all layers. However, as we have discussed in Section 6, different layers in a neural network may carry different levels of information, and therefore the dropout parameters could vary across the layers. We will re-emphasize this in the Future Direction in Section 6, and our theoretical analysis could serves as a first step in the “adaptive” dropout parameters to explore the Rashomon set.
> > > >
> > > > 9. **Concerning the quantification of predictive multiplicity, it is said that "[f]or example, Long et al. [2023], Cooper et al. [2023] and Watson-Daniels et al. [2023] quantify predictive multiplicity by the standard deviation, variance and the largest possible difference of the scores (termed viable prediction range (VPR) therein) respectively" So, what definition between those three is retained in the article?**
> > > >
> > > >     In this paper, we adopt the score variance from [Cooper et al., 2023] and the VPR from [Watson-Daniels et at., 2023] for score-based predictive multiplicity metrics, and report them in Section 4. We retain the definitions and implementations of them in the original article. Due to space limit, we summarize all the mathematical formulations and implementations in Appendix B. Please note that in this paper we did not propose a new predictive multiplicity metric, but a new strategy to efficiently estimate all existing predictive multiplicity metrics. We will re-emphasize this point by adding a sentence that summarizes the discussion above in Section 2.
> > > >
> > > > 10. **This might not be of an "ethical" concern, but the 9-page limit is exceeded by 1/4th \~ 1/3rd of a page.**
> > > >
> > > >     We thank the reviewer for pointing this out. As instructed from the Author Guide for ICLR 2024, Ethics Statement and Reproducibility Statement are optional and *will not count toward the 9 page limit*, but should not be more than 1 page, as we have complied with.
> > > >
> > > >
> > > > Thanks again and we would be happy to provide more clarifications and answer any follow-up questions.

---

> ### Comment · Reviewer_BakJ · 2023-11-20
>
> I thank the authors for their exhaustive response.
>
> I don't feel like the responses (#1, #2, #4, #5, #6) actually answer my concern. For example, having literature comparing their approach to retraining (#4) might not be relevant here, for it depends on how these approaches resemble or not retraining on the point I raised in Weakness - Major - 3. Another example would be, response #5: sure, "[the goal of this paper is to] provides an alternative method to estimate predictive multiplicity metrics more efficiently", but in the article, what motivates its estimation is that "when predictive multiplicity is left out of account, an arbitrary choice of a single model in the Rashomon set may lead to systemic exclusion from critical opportunities, unexplainable discrimination, and unfairness, to individuals". Finally, response 6: even though the mass is concentrated around $\sqrt{d}$, it is uniformely distributed on each point at a given distance of the mean.
>
> Response #3 did help me understand how a single empirical minimizer was shared by the approaches. I do feel like some of the details in Apprendix (some of the points I raised) deserve to be in the main part of the paper and not the appendices and am glad to know that some of that information will be transfered to the main part.
>
> I thank again the authors for their honest responses; I will keep my score.

---

> > ### Author Response · Authors · 2023-11-21
> > **Second Response to Reviewer BakJ**
> >
> > We thank the reviewer for the further comments and we understand the concerns.
> >
> > We would like to emphasize that when the hypothesis is huge and general, exploring models in the Rashomon set is a computationally infeasible task. Indeed, re-training is the strategy that could potentially explore the most "diverse" models in the Rashomon set, with an cost of high time complexity. On the other hand, the proposed dropout method may only search "local" models around a given pre-trained model; however, the time complexity to obtain a model is significantly smaller, with the probability that a dropout model is in a Rashomon set guaranteed by the proposed propositions. The effectiveness and the efficiency to explore the Rashomon set then form a trade-off---re-training is effective but time-consuming, whereas dropout is less effective but more efficient. How to leverage these two strategies is up to the practitioners' and stakeholders' choice. Note that both re-training and the dropout strategies provide a lower bound for multiplicity metrics, and dropout offers a fast way for a lower bound (despite that could be smaller than the bound obtained by re-training). If the lower bounds of multiplicity metrics from the dropout method are already large, we could conclude that the model suffers from predictive multiplicity.
> >
> > In Figure 2 (hypothesis space is single-hidden-layer (i.e., small) neural networks), we have shown that the proposed dropout method has comparable performance in terms of estimating predictive multiplicity metrics. When the hypothesis space is large (e.g., VGG-16), we implement the re-training + dropout method mentioned in Section 6 (added in Appendix E.3 in the revision), showing a potentially better mixed strategy that leverages the pros of both methods.
> >
> > We admit that the uniform assumption of the distribution of models in the Rashomon set might not reflect their true underlying distributions. However, it is a compromise in theoretical analysis, just like we have almost always assume that a noise follows Gaussian distribution. How to characterize the distributions of models in the Rashomon set is our research direction in the near future.
> > Also, the comparison of the runtime between re-training and dropout might not be perfectly aligned on an equal basis, as re-training models to approach a given pre-trained model is different from re-training models to the possibly best performance. In fact, a procedure such as a teacher (pre-trained model) and student (re-training model) architecture could be more reasonable to compare the runtime. However, the way we perform the re-training strategies is inherited from existing works such as [Kulynch et al., 2023] and [Semenova et al., 2019]. On the other hand, the comparison of the runtime speedup between AWP and dropout has less concern, as they share the same objective of perturbing an given pre-trained model to explore the Rashomon set.
> >
> > We hope that the response above could (partially) address your concerns, and we appreciate your comments to help us improve this manuscript, thanks again!

---

> > > ### Comment · Reviewer_BakJ · 2023-11-22
> > >
> > > I thank the reviewers for their response. I stand by my score and will be defending acceptance.

---

> > > > ### Author Response · Authors · 2023-11-22
> > > >
> > > > Thank you again for your engagements in the author-reviewer discussion period, and for defending acceptance of this manuscript. Your questions and suggestions have help us improve the manuscript!

---

### Official Review · Reviewer_kvbj · 2023-10-26

**Soundness:** 3 good
**Presentation:** 2 fair
**Contribution:** 2 fair
**Rating:** 5
**Confidence:** 3

**Summary:**

This paper studies the possibility of using Dropout to explore the Rashomon set. It proves that for a FFNN, we could bound the probability that a Dropout realization is in a certain Rashomon set. In experiments it shows that the proposed the method does not explore the Rashomon set as effectively as AWP (as measured by several predictive multiplicity metrics), but is much faster as it does not retrain any model.

**Strengths:**

1. This paper establishes some theoretical bounds (although seemingly loose) on the probability that FFNNs with Dropout are in the corresponding Rashomon Set.
2. The proposes method is easy to implement.

**Weaknesses:**

1. It is unclear what's the practical use of the propose method. It is fast, but it does not explore the Rashomon set well. For this reason, we can only mitigate predictive multiplicity *as estimated by Dropout* but not in general.
2. Following 1, it seems like additional experiments on whether the mitigation via Dropout also transfers to, say, AWP, is interesting.
3. The bounds in Proposition 2 and 3 only converge to 1 when $d\to\infty$, which does not seem like useful. See Q3 as well.
4. It is not clear why a concentration bound helps. Notably, in applications, we want the models that are in the Rashomon set but closer to the boundary. In fact, it seems like in practice we need to sample a few weights and empirically verify that they have low loss (?). If so, a concentrated distribution, especially one that's more concentrated when the dim of the model increases, seems like a bad feature. A method that samples very diverse model that potentially has a higher probability of falling outside the Rashomon set seems more desirable.

**Questions:**

1. AWP is slower due to re-training, but the models are trained only once. Therefore, doesn't it run *faster* than Dropout (because it uses fewer samples/models to explore the Rashomon set) with a reasonably large test data?
2. What does "5 models" mean in Figure 4b? 5 different base weights, or 5 different architectures?
3. Is $\epsilon$ and the $L$ in Eq.(5), (7) and (10) related to the "sum" of loss or the "mean" of loss? It seems like it's the sum? If so, by changing $\epsilon$ to some offset on the mean loss, we can probably get a convergence basing on the sample size, which is much more meaningful than dimension of the model's hidden layers.

---

> ### Author Response · Authors · 2023-11-16
> **Response to Reviewer kvbj**
>
> 1. **It is unclear what's the practical use of the propose method. It is fast, but it does not explore the Rashomon set well. For this reason, we can only mitigate predictive multiplicity *as estimated by Dropout* but not in general.**
>
> 2. **Following 1, it seems like additional experiments on whether the mitigation via Dropout also transfers to, say, AWP, is interesting.**
>
>     We address the reviewer's concerns in Weaknesses 1 and Weaknesses 2 together here. The *good* exploration of the Rashomon set, i.e., obtaining very *diverse* models with almost-equally-optimal performance, comes with the cost of efficiency (cf. e.g. [Xin et al., 2023], [Watson-Daniels et al., 2023]).
>     With this in mind, the practical use of the proposed dropout method is to provide an alternative to the current re-training strategy that is not only more efficient but also bears with theoretical analysis (Proposition 1 to Proposition 4).
>     Numerical results, e.g., in Figure 2, also show that the proposed method is comparable or outperforms the re-training strategy in terms of exploring the Rashomon set.
>     Note that both re-training and the dropout methods give a subset (defined in (2)) of the true Rashomon set, and will all lead to an *under-estimate* of predictive multiplicity metrics. Therefore, higher estimates of the multiplicity metrics indicate a better exploration of the Rashomon set.
>     To the best of our knowledge, it is the first time when the dropout method is connected and analyzed with notions of the Rashomon set and predictive multiplicity, making reporting predictive multiplicity for large-scale models possible.
>
>     Moreover, the methods to mitigate predictive multiplicity discussed in Section 5 are not limited to the dropout methods. In fact, the ensemble method shown in Figure 4(a) work for any strategies that could obtain models in the Rashomon set. Since more models in the the ensemble leads to smaller multiplicity metrics, the proposed dropout method serves as an efficient way to obtain a huge amount of models.
>
>     Mitigating predictive multiplicity, especially when selecting a model with the least predictive multiplicity from several given models, as shown in Figure 4(b), could also be achieved by the AWP. We will include an additional experiment in the Appendix of future revision.
>     Lastly, please note that the re-training strategy can be combined with the dropout method to order to leverage the efficiency and efficacy of Rashomon set exploration, as already discussed in Section 6 Future Directions. We will summarize the discussion above and add a paragraph accordingly in Section 2.
>
> 3. **The bounds in Proposition 2 and 3 only converge to 1 when $d\to\infty$, which does not seem like useful. See Q3 as well.**
>
>     We thank the reviewer for bring this up, and will add explanations and citations to support this assumption.
>     The condition $d\to\infty$ is a regime commonly discussed in the asymptotic analysis of learning models such as over-parameterization [1], and the universal approximator theorem [2]. We would like to emphasize that the asymptotic behavior when $d\to\infty$ is based on the bound we derived in Eq. (5).
>
>     [1] Cao, Y. and Gu, Q., 2020, April. Generalization error bounds of gradient descent for learning over-parameterized deep relu networks. In Proceedings of the AAAI Conference on Artificial Intelligence (Vol. 34, No. 04, pp. 3349-3356).
>
>     [2] Hornik, K., Stinchcombe, M. and White, H., 1989. Multilayer feedforward networks are universal approximators. Neural networks, 2(5), pp.359-366.
>
> **Continued...**

---

> > ### Author Response · Authors · 2023-11-16
> > **Response to Reviewer kvbj - 2**
> >
> > 4. **It is not clear why a concentration bound helps. Notably, in applications, we want the models that are in the Rashomon set but closer to the boundary. In fact, it seems like in practice we need to sample a few weights and empirically verify that they have low loss (?). If so, a concentrated distribution, especially one that's more concentrated when the dim of the model increases, seems like a bad feature. A method that samples very diverse model that potentially has a higher probability of falling outside the Rashomon set seems more desirable.**
> >
> >     We thank the reviewer for asking why concentration bound helps in our analysis. The goal to explore the Rashomon set is to find models with diverse outputs while satisfying the loss deviation constraint. If we are allowed to find models outside of the Rashomon set, the performance of the models could be much worse and are less likely to be selected in practice. Note that the phenomenon of predictive multiplicity is only discussed within almost-equally optimal models (say models that all have 99\% accuracy). In this sense, the concentration bound helps to characterize the probability that a model after dropout will still be inside the Rashomon set, and is a vital mathematical tool used in our proofs.
> >
> >     Like the reviewer suggested, ideally it is more efficient to directly select models at the boundaries of the Rashomon set. It is possible when the Rashomon set could be explicitly expressed, e.g., for ridge regression. However, for general hypothesis space, how to search at the boundaries of a Rashomon set is still an open challenge, and re-training (and AWP) was the only strategy. Note that the models obtained from either re-training or the dropout method require an empirical validation of the loss with the test data to make sure the models are in the Rashomon set. In Table E.7 in the Appendix, we report the percentage of the 500 models from each strategies that are in the Rashomon set with a given empirical minimizer and varying $\epsilon$. For most cases, the percentage of models in the Rashomon set obtained by the dropout method is significantly higher than that by the re-training strategy. In other words, the dropout method is more controllable (cf. the leftmost column in Figure 2) in terms of the loss deviation and Table E.7 serves as an support on why the concentration bound could be useful. We will summarize the discussions above and emphasize it in Section 4 in future revision.
> >
> > 5. **AWP is slower due to re-training, but the models are trained only once. Therefore, doesn't it run *faster* than Dropout (because it uses fewer samples/models to explore the Rashomon set) with a reasonably large test data?**
> >
> >     We thank the reviewer for raising this confusion, and will provide more details of the AWP algorithm to help future readers to better understand. Suppose we have a dataset with n samples and each sample could be classified into c candidate classes. AWP maximizes each entries of a **given** model $h\_w(x)$, i.e., $[h\_w(x)]\_i$ for each $i \in [c]$, under the constraint that the model after perturbation is still inside the Rashomon set. Therefore, the AWP requires $n\times c$ perturbations/re-training to estimate multiplicity metrics for an entire dataset. We can imagine that each of these $n\times c$ perturbations leads to a model in the Rashomon set that outputs the most inconsistent score for a given class per sample. On the other hand, the dropout procedure applies dropout on the given model $h\_w(x)$ without the need to re-train (gradient computation and weight updating) the model. We will add the discussion above in Section 2.
> >
> > 6. **What does "5 models" mean in Figure 4b? 5 different base weights, or 5 different architectures?**
> >
> >     The 5 models are obtained with different weight initializations and the same architectures. We will emphasize it by adding more description in the caption of Figure 4(b) according and thank you for clarification!
> >
> > **Continued...**

---

> > > ### Author Response · Authors · 2023-11-16
> > > **Response to Reviewer kvbj - 3**
> > >
> > > 7. **Is $\epsilon$ and $L$ the in Eq.(5), (7) and (10) related to the "sum" of loss or the "mean" of loss? It seems like it's the sum? If so, by changing $\epsilon$ to some offset on the mean loss, we can probably get a convergence basing on the sample size, which is much more meaningful than dimension of the model's hidden layers.**
> > >
> > >     We thank the reviewer for bringing up this point that might cause potential confusion, and will add a paragraph in Section 3 to clarify. The Rashomon set in Eq. (2) is defined regarding the mean of the loss for a given dataset of a **fixed size $n$**.  In other words, the dataset is a given parameter, not a source of randomness in the definition of a Rashomon set.
> > >     Note that the randomness in the convergence in Eq. (7) and (10) is with respect to the dropout matrix, not like the vanilla concentration bound that consider drawing samples from a data distribution, it makes sense that those convergence results does not rely on the number of samples.
> > >     In Eq. (5), the loss we used is the sum of the loss for each samples as we w.l.o.g. assume the data matrix $X$ is whitened. In Eq. (7) and (10) the losses are defined with the mean of the loss for each sample. We will make this point clear in future revision.
> > >
> > > Thanks again and we would be happy to provide more clarifications and answer any follow-up questions.

---

### Official Review · Reviewer_pYox · 2023-10-31

**Soundness:** 3 good
**Presentation:** 4 excellent
**Contribution:** 3 good
**Rating:** 6
**Confidence:** 2

**Summary:**

The paper described a Rashomon set exploration method through Drop-out with probabilistic bound. The paper starts with fairly well-covered literature of Rashomon set research and motivates its proposal by pointing out the computation cost of existing empirical solution (re-training, AWP). The solution is fairly simple by adopting Drop-out where Rashomon set likely rests. Probably the most significant part of this paper (theoretically) would be pointing out the probabilistic bound of Rashomon set under Drop-out. Empirical results show the proposed method is computationally efficient than previous solutions and even showing better divergence metric than retraining.

**Strengths:**

1. Predictive multiplicity itself is an interesting topic that worths more investigation. The method proposed in this work is a great complement of existing literature in this field.
2. The paper is well written and motivated. It comes with sufficient background knowledge to understand the gap in the literature.
3. Potential application of this approach is covered in Section 5, which is good since I was concerning where people can use this innovation in their work.

**Weaknesses:**

1. Model augmented by Dropout could result in a fairly small search space of Rashomon set. I am not very convinced that this is a good idea in practice if our goal is to look for a better model that can address various reliability problem of predictive model. e.g. fairness etc. It maybe inspirational to see the movement of predictive multiplicity measurement, but I am wondering what is the practical meaning of it.
2. The paper demonstrates the effectiveness of the proposed method on toy datasets that were used for decades. As the paper concerns the efficiency of existing methods, I am wondering if the authors can introduce more realistic tasks to show the effectiveness of the proposed method quantitatively.  While COCO is good example, it is very qualitative without much statistic support.
3. There is a descriptive gap in section 3.2 where transforming deviation between $L_{SSE}(\mathbf{w}_D^*)$ and $L_{SSE}(\mathbf{w})$ suddenly become  deviation between  $L_{SSE}(\mathbf{w}_D^*)$ and $L_{SSE}(\mathbf{w}')$. I don't quite see why they are aligned or if the model works correctly under $L_{SSE}(\mathbf{w}')$ if it is not trained with such dropout rate.

**Questions:**

The proposition 1 uses deviation between  $L_{SSE}(\mathbf{w}_D^*)$ and $L_{SSE}(\mathbf{w}')$ but not original model parameter $L_{SSE}(\mathbf{w}')$. How to make the connection ?

---

> ### Author Response · Authors · 2023-11-16
> **Response to Reviewer pYox**
>
> We thank the reviewer for the feedback and encouragement! We address the questions point-by-point below.
>
> 1. **Model augmented by Dropout could result in a fairly small search space of Rashomon set. I am not very convinced that this is a good idea in practice if our goal is to look for a better model that can address various reliability problem of predictive model. e.g. fairness etc. It maybe inspirational to see the movement of predictive multiplicity measurement, but I am wondering what is the practical meaning of it.**
>
>     We thank the reviewer to point out this confusion. Our goal is not finding models in the Rashomon set that comply with additional reliability problems such as fairness, but to efficiently estimate predictive multiplicity, despite that both involve exploring the Rashomon set, i.e., obtaining almost-equally-optimal models from the hypothesis space. As stated in the second paragraph in Section 1, our focus is estimating predictive multiplicity, a problem induced from the Rashomon effect, not the Rashomon effect stated in the first paragraph of Section 1. We agree with the reviewer that studying if the dropout models in the Rashomon set could lead to other reliability properties such as fairness is an interesting future direction.
>
>     There is a trade-off between the efficiency and efficacy of exploring the Rashomon set and prediction multiplicity estimation. Note that re-training and our proposed dropout method all aim to approximate a subset of the entire (true) Rashomon set, and thus all estimates based on re-training or our method are under-estimation of the true multiplicity metrics. Despite that our method may only explore a small subset in the Rashomon set, it give an lower bound of the multiplicity metrics in an efficient manner, as clearly shown in Table 1. Moreover, the estimation using the dropout method also gives results comparable to re-training and AWP, as shown in the empirical evidence in Figure 2. Therefore, the proposed dropout strategy serves as yet another option (besides AWP and re-training) to explore the Rashomon set depending on the need of practitioners.
>
>     We will summarize the discussion above and add it to the final revision.
>
> 2. **The paper demonstrates the effectiveness of the proposed method on toy datasets that were used for decades. As the paper concerns the efficiency of existing methods, I am wondering if the authors can introduce more realistic tasks to show the effectiveness of the proposed method quantitatively. While COCO is good example, it is very qualitative without much statistic support.**
>
>     The datasets we used for evaluation throughout this paper includes 6 UCI datasets, CIFAR-10/-100, and the MS COCO dataset, ranging from small tabular data to large-scale image datasets. Indeed, the UCI datasets are small and have been used for decades; however, they are still widely used in predictive multiplicity literature, especially in the early stage of this new field. For example, the UCI Adult dataset is used in [Long et al., 2023] and [Kulynch et al., 2023]; the mammography dataset is used in [Watson-Daniels et al., 2023]; and other UCI datasets are also adopted in [Xin et al., 2022]. Note that all these literatures are very recent.
>
>     Aside from the UCI datasets, we also evaluate the proposed method on image datasets such as CIFAR-10/-100 and the MS COCO, where the corresponding quantitative metrics including loss/accuracy changes and predictive multiplicity metrics are all reported in Sections E.2 and E.3, and also referred in the main text. We believe the datasets we used here are on par or more abundant than existing literature. We will re-emphasize the reasons (as above) why we select these datasets in the final version of the paper.
>
>
> **Continued...**

---

> > ### Author Response · Authors · 2023-11-16
> > **Response to Reviewer pYox - 2**
> >
> > 3. **There is a descriptive gap in section 3.2 where transforming deviation between $L\_{SSE}(\mathbf{w}\_D^\*)$ and $L\_{SSE}(\mathbf{w}^\*)$ suddenly become deviation between $L\_{SSE}(\mathbf{w}\_D^\*)$ and $L\_{SSE}(\mathbf{w}'^\*)$. I don’t quite seen why they are aligned or if the model works correctly under $L\_{SSE}(\mathbf{w}'^\*)$ if it is not trained with such dropout rate.**
> > 4. **The proposition 1 uses deviation between $L\_{SSE}(\mathbf{w}\_D^\*)$ and $L\_{SSE}(\mathbf{w}'^\*)$ but no original model parameter $L\_{SSE}(\mathbf{w}'^\*)$. How to make the connection ?**
> >
> >     Thank you for raising this point and we will address these two related questions together here. Proposition 1 proves that the expected deviation between the losses evaluated on dropout weights $\mathbf{w}_D^*$ and on the **diluted** weights $\mathbf{w}' = (1-p)\mathbf{w}^\*$.  The dilution factor $(1-p)$ comes from the expectation of the dropout matrix $D$, and in fact, dropout has been known as weak dilution in past literature [Hertz et al., 1991]. That’s why we did not introduce the notion of the Rashomon set in Proposition 1. Despite that in Proposition 1, the loss deviation does not directly work with the original optimal weight $\mathbf{w}^\*$, Proposition 1 serves as the inspiration to control loss deviation via dropout for the rest of the propositions presented in this paper. In the rest of the propositions, we use different proving techniques to make the loss deviation directly work with the original weight $\mathbf{w}^\*$ instead of the diluted weight $\mathbf{w}'^\*$. Note that the dropout here is applied in the inference/ test time, and therefore it does not depend how the model is trained (e.g., with or without dropout), i.e., the proposed dropout-based Rashomon set exploration is model-agnostic. We thank the reviewer again for helping us be aware of the potential confusion in the description of Proposition 1, and will clarify it by adding more detailed discussion accordingly.
> >
> >
> > Thanks again and we would be happy to provide more clarifications and answer any follow-up questions.

---

> > > ### Comment · Reviewer_pYox · 2023-11-22
> > >
> > > Thank you for the response. I will keep my current score.

---

> > > > ### Author Response · Authors · 2023-11-22
> > > >
> > > > Thank you again for the response. Your comments and suggestions have helped us improve this manuscript!

---

### Official Review · Reviewer_7265 · 2023-11-02

**Soundness:** 3 good
**Presentation:** 4 excellent
**Contribution:** 3 good
**Rating:** 6
**Confidence:** 3

**Summary:**

The paper studies the problem of how to measure and mitigate predictive multiplicity.
To achieve them, the authors utilize the dropout technique to explore the models in the Rashomon set.
Rigorous theoretical analysis is provided to connect dropout and Rashomon set.
Numerical results demonstrate the effectiveness of the proposed method.

**Strengths:**

1. The proposed method is simple, straightforward, and well-motivated. Utilizing the dropout technique to explore the models in the Rashomon set is interesting.
2. Rigorous theoretical analysis is provided to connect dropout and Rashomon set.
3. The paper is well-written and well-organized. The authors first show the implementations on linear models and extend them to feedforward neural networks.
4. The limitations and potential solutions are also discussed in the paper.

**Weaknesses:**

1. In the experiments, the authors mentioned that "On the other hand, AWP outperforms both dropouts and re-training, since it adversarially searches the models that mostly flip the decisions toward all possible classes for each sample." I may miss some details of the method part, how can the proposed method to adversarially search the models since the dropout is random?
2. As mentioned by the authors, good performance comes at the cost of efficiency.

**Questions:**

1. It seems the proposed method is only evaluated on in-distribution scenarios. Can it be applied to out-of-distribution data?
2. Are the uncertainty scores calibrated? In other words, are the confidence scores reliable?

---

> ### Author Response · Authors · 2023-11-16
> **Response to Reviewer 7265**
>
> We thank the reviewer for the feedback and encouragement! We address the questions point-by-point below.
>
> 1. **In the experiments, the authors mentioned that "On the other hand, AWP outperforms both dropouts and re-training, since it adversarially searches the models that mostly flip the decisions toward all possible classes for each sample." I may miss some details of the method part, how can the proposed method to adversarially search the models since the dropout is random?**
>
>     We thank the reviewer for pointing out this confusion. We would like to further clarify that AWP and re-training  are two existing baselines we compared against with our proposed dropout method. The AWP, re-training and ours are three different strategies to explore the Rashomon set, i.e., obtaining almost-equally-optimal models from the hypothesis space, and ours and AWP will not be applied at the same time. The AWP algorithm, as proposed in [Eq. 9, Hsu \& Calmon, 2022], will adversarially perturb the output scores of a given sample to each class. For example, if there are $c$ classes, AWP maximizes each entries of $h\_w(x)$, i.e., $[h\_w(x)]\_i$ for each $i \in [c]$, under the constraint that the model after perturbation is still inside the Rashomon set. Therefore, for one sample, AWP requires to perform one perturbations (adversarial training) via SGD for each class (i.e., $c$ adversarial training in total for one sample), and if there is a dataset with $n$ samples, we need $n\times c$ adversarial training to estimate multiplicity metrics for the entire dataset. We can imagine that each of these $n\times c$ perturbations leads to a model in the Rashomon set that outputs the most inconsistent score for a given class per sample. On the other hand, the dropout method does not require to iterate over all samples and does not require gradient computations and updating model weights. In this sense, AWP is very different from the proposed dropout strategy.
>
>     Due to space limit, we refer the readers to [Section 4, Hsu \& Calmon, 2022] for more details of the AWP algorithm in the initial draft. We will add pseudo codes for the AWP in Appendix B, and a brief discussion in Section 2 to clarify the difference of the three strategies in the final version of the paper.
>
> 2. **It seems the proposed method is only evaluated on in-distribution scenarios. Can it be applied to out-of-distribution data?**
>
>     We thank the reviewer for bringing this up. Indeed, the loss, accuracy and predictive multiplicity metrics reported in this paper are all evaluated on test data, i.e., unseen but in-distribution samples. The proposed method can also be applied for out-of-distribution (OOD) data. However, without special treatment (e.g., OOD generalization), the pre-trained model would have much higher loss $L(h_{w*}, \mathcal{D}\_{OOD})$ when fed with OOD data, leading to a larger Rashomon set (with the same $\epsilon$) and potentially much higher values of predictive multiplicity metrics (since multiplicity metrics are optimized over the Rashomon set; please also refer to [Defn 1, Hsu et al., 2022]). We will add a brief paragraph to include the discussion above in Section 3.
>
> 3. **Are the uncertainty scores calibrated? In other words, are the confidence scores reliable?**
>
>     Thanks for the great question! We would like to further clarify that if a perfectly calibrated classifier assigns a 50\% score to a sample (e.g., in binary classification), it does not necessarily mean that this sample has high multiplicity. A perfectly calibrated classifier is one whose predicted classes matches the true classes **on average** across samples (e.g., samples predicted to be 50\% of one class have true outcomes matching that class 50\% of the time). However, this does not necessarily translate to a (in)consistent set of predictions for a **single target sample** across equally calibrated classifiers. It may be the case that **all** calibrated models drawn from the Rashomon set assign the same 50\% probability for that sample (no multiplicity). Conversely, some models may assign higher and lower confidence for that sample (high multiplicity) yet, on average, still be well-calibrated. Again, this happens because calibration (like accuracy and loss) is an average metric across all samples. We will add a brief discussion to include the response above in Section 4 in the final revision.
>
> Thanks again for the review! We would be happy to provide more clarifications and answer any follow-up questions.

---

> > ### Comment · Reviewer_7265 · 2023-11-23
> > **Thanks for the rebuttal**
> >
> > Thank the authors for the detailed response. The answer to Q1 helps me correct my understanding of AWP. I admit that the question about OOD generalization is a little bit beyond the scope of this paper. I prefer to keep my score and vote for acceptance.

---

### Author Response · Authors · 2023-11-21
**Common Response to Reviewers**

We would like to thank the reviewers for their time and effort in reading and commenting on the manuscript. We appreciate that the reviewers found the paper **"well-motivated and written"** (Reviewer 7265, pYox and BakJ), **“a great complement of existing literature in this field”** (Reviewer pYox), and **“theoretical analysis is provided”** (Reviewer 7265 and kvbj). We have integrated all of the reviewers' major and minor comments (including typos) and outlined changes that we have addressed in the revised version; the changes to the original main text and appendices are highlighted **in green** therein.

 - We clarify the experimental settings and the computational details regarding the runtime comparison of the baselines re-training and AWP, against the proposed dropout strategy, and provide a more detailed introduction of the AWP algorithm in the Appendix and in the main text.

 - We clarify and add more discussions on the theoretical results, including the asymptotic analysis $d \to\infty$, the uniform assumption of the distribution of models in the Rashomon set, the current status of theoretical analysis of predictive multiplicity estimators, and questions about proposition 2, 4 and 5.

 - We provide **three additional experiments**:
    - We compare the predictive multiplicity metrics estimated from classifiers **with and without model calibration** using Platt scaling in Appendix E.4 (Figure E.27). It is shown that calibration will not reduce predictive multiplicity, as the reason we provided at the end of Appendix E.4.
    - We show that the proposed dropout method could explore similar parameter spaces as the re-training strategy **when the hypothesis space is small** (e.g., logistic regression) in Appendix E.1 (Figure E.13). We would like the emphasize that either baselines such as re-training and AWP, or the proposed dropout method, can only explore partially of the entire (true) Rashomon set when the hypothesis space is large, and therefore all estimates obtained from the three strategies are under-estimates of the multiplicity metrics.
    - We implement the **re-training + dropout** strategy, as mentioned in Section 6 to  leverage between the re-training and dropout strategies in Appendix E.3 (Figure E.22 and E.23). When the hypothesis space is large (VGG-16 trained with CIFAR-10), the re-training + dropout strategy is more effective in terms of exploring models in the Rashomon set to estimate predictive multiplicity.

Finally, we would like to point up that exploring the Rashomon set for general hypothesis space has been an open challenge, and the proposed dropout strategy serves as yet another option (besides existing AWP and re-training algorithms) to explore the Rashomon set. Despite that the dropout strategy may only search locally around an given pre-trained model, it provides lower bounds of predictive multiplicity metrics in an efficient manner. This paper is also one of the first kind that systematically compare the efficiency of Rashomon set exploration for general hypothesis spaces. Please feel free to follow up! We very much welcome further discussions.

---

### Author Response · Authors · 2023-11-22
**Thank you again for your time and feedback!**

Thank you once more for your comments and time---your suggestions have helped us improve this manuscript! We're writing since we are now nearing the end of the author-reviewer discussion period. If you have additional comments, please let us know and we would be happy to try to answer before the deadline. If you think we have addressed your questions, we would appreciate it if you would kindly consider raising your score.

---

### Meta-Review · Area_Chair_L9UN · 2023-12-12

**Metareview:**

In most settings, optimal classifier is not uniquely defined and can still provide different predictions. To better understand this phenomenon, the concept of Rashomon set ie set of models that achieves similar accuracy, is introduced. Unfortunately the computation of such sets is not usually feasible and some approximation via discretization is used instead. The previous approaches are based on retraining the models after stochastic perturbation of the data or training procedure and then select the models with at most epsilon error.
This paper explore a different approach based on dropout and show that huge speed up can be easily obtained. Moreover, additional theoretical insights are provided to explains the behavior of the methods in some restricted settings.

The majority of the reviewers agree that the approach is interesting, the paper is well written and are leaning toward acceptance. Several limitations are pointed and the authors quite agree on them. Despite the limitations, I think this paper provides an important step toward understanding predictive multiplicity and provides a tool for it.

> Minor additional limitation

Even the empirical approximation is based on the knowledge the exact loss of the optimal solution of ERM $w^\star$. This is *often* not available and usually the consequence of this is ignored, included in this paper.
I appreciated that the authors first restrict to simple model such as Ridge regression to gain some insights and more of these analyses are actually needed. Even in linear model class where for example $X^\top X$ is not invertible, the description (and even approximation) of Rashomon set seems unclear.

$\epsilon$ controls the size of the set. But this might not be a usefull notion since the size is infinite anyway.

**Justification For Why Not Higher Score:**

Overall the reviewers agree that the contributions are interesting and the topics is quite highly important. However several explicit limitations prevented from voting for higher score.

**Justification For Why Not Lower Score:**

This paper is an important step toward understanding a highly difficult problem and currently very little effective approach are available. As such, I think this paper is above acceptance threshold.

---

### Decision · Program_Chairs · 2024-01-16

Accept (poster)